# GUI-World: A Video Benchmark and Dataset for Multimodal GUI-oriented Understanding

**Dongping Chen**[1][*]**, Yue Huang**[2][*]**, Siyuan Wu**[1]**, Jingyu Tang**[1]**, Liuyi Chen**[1]**,**
**Yilin Bai**[1]**, Zhigang He**[1]**, Chenlong Wang**[1]**, Huichi Zhou**[3]**, Yiqiang Li**[1]**,**
**Tianshuo Zhou**[1]**, Yue Yu**[1]**, Chujie Gao**[1]**, Qihui Zhang**[4]**, Yi Gui**[1]**, Zhen Li**[1]**,**
**Yao Wan**[1][†]**, Pan Zhou**[1][†]**, Jianfeng Gao**[5]**, Lichao Sun**[6]

[1]Huazhong University of Science and Technology, [2]University of Notre Dame
[3]Imperial College London, [4]Peking University, [5]Microsoft Research, [6]Lehigh University
{u202112313, wanyao, panzhou}@hust.edu.cn

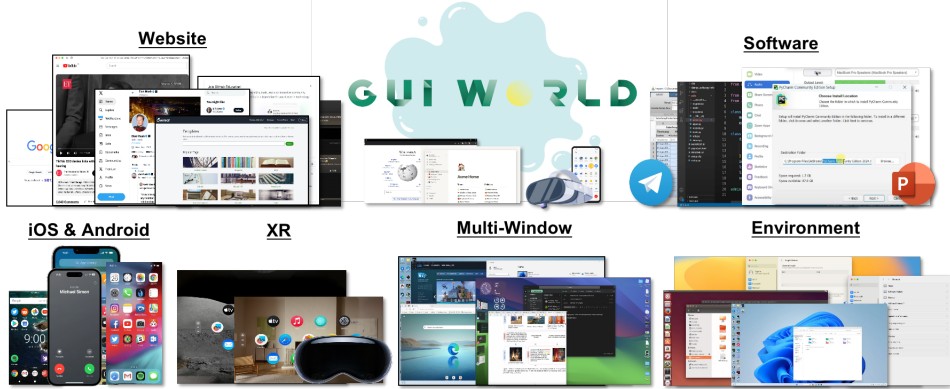

Figure 1: GUI-World: A comprehensive dataset for GUI-oriented capabilities encompasses six scenarios and diverse tasks, offering significant potential for real-world applications.

## Abstract

Recently, Multimodal Large Language Models (MLLMs) have been used as agents to control keyboard and mouse inputs by directly perceiving the Graphical User Interface (GUI) and generating corresponding commands. However, current agents primarily demonstrate strong understanding capabilities in static environments and are mainly applied to relatively simple domains, such as Web or mobile interfaces. We argue that a robust GUI agent should be capable of perceiving temporal information on the GUI, including dynamic Web content and multi-step tasks. Additionally, it should possess a comprehensive understanding of various GUI scenarios, including desktop software and multi-window interactions. To this end, this paper introduces a new dataset, termed GUI-World, which features meticulously crafted Human-MLLM annotations, extensively covering six GUI scenarios and eight types of GUI-oriented questions in three formats. We evaluate the capabilities of current state-of-the-art MLLMs, including Image LLMs and Video LLMs, in understanding various types of GUI content, especially dynamic and sequential content. Our findings reveal that current models struggle with dynamic GUI content without manually annotated keyframes or operation history. On the other hand, Video LLMs fall short in all GUI-oriented tasks given the sparse GUI video dataset. Therefore, we take the initial step of leveraging a fine-tuned Video LLM, GUI-Vid, as a GUI-oriented assistant, demonstrating an improved understanding of various GUI tasks. However, due to the limitations in the performance of base LLMs, we conclude that using video LLMs as GUI agents remains a significant challenge. We believe our work provides valuable insights for future research in dynamic GUI content understanding. All the dataset and code are publicly available at: https://gui-world.github.io.

---

[*]Equal Contribution, [†]Corresponding Authors

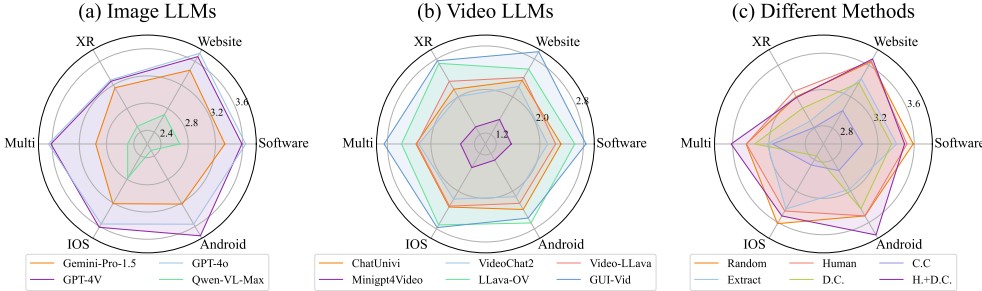

Figure 2: Comparative performance of different MLLMs in six scenarios of GUI-WORLD. (a) Performance of four mainstream Image LLMs. (b) Performance of five Video LLMs and our GUI-VID. (c) Performance among six methods. See Section 3.2 for more details.

# 1 INTRODUCTION

Multimodal Large Language Models (MLLMs), such as GPT-4V(ision) (OpenAI, 2023) and LLaVA (Liu et al., 2023b), have significantly contributed to the development of both vision and language domains (Yin et al., 2024). These models bring forth innovative solutions and paradigms for traditional visual tasks, including visual reasoning (Yang et al., 2023b), medical image interpretation (Li et al., 2024b), and applications in embodied agents (Huang et al., 2023). One particularly promising area is Graphical User Interface (GUI) understanding, which holds significant potential for real-world applications, such as webpage comprehension (Hong et al., 2024; Lai et al., 2024) and navigation by GUI agents (Yang et al., 2023a; Niu et al., 2024; Wang et al., 2024). The key challenges of GUI understanding are twofold: effective GUI agents are expected to (1) possess a deep understanding of GUI elements, including webpage icons, text identified through Optical Character Recognition (OCR), and page layouts, and (2) exhibit an exceptional ability to follow instructions within GUI contexts, such as conducting searches through search engines.

Despite significant progress, as illustrated in Table 1, prior studies on GUI-related datasets and benchmarks suffer the following limitations: (1) *Inability to Handle Dynamic Environments.* Most studies primarily emphasize the static features of GUI scenarios, often overlooking the importance of enabling MLLMs to effectively handle dynamic information and sequential operations. For instance, an agent's task performance can be disrupted by unexpected elements such as pop-up advertisements, underscoring a gap in handling dynamic sequential tasks. (2) *Limited Scenarios.* Current research is typically restricted to Web-based or mobile environments, which limits to assess their generalization ability and robustness across scenarios. For instance, GUI-oriented models may need to operate across diverse platforms such as Windows, macOS, Linux, iOS, Android, and eXtended Reality (XR) environments. Operations may sometimes involve multiple windows. Therefore, expanding research to encompass these varied environments will enhance the adaptability and effectiveness.

To mitigate these gaps, this paper introduces GUI-WORLD, a comprehensive dataset containing 12,379 GUI videos, specifically designed to evaluate and enhance the capabilities of GUI agents. This dataset encompasses a wide range of GUI scenarios, including popular websites, desktop and mobile applications across various operating systems, multi-window interactions, as well as XR environments. The data collection process involves sourcing GUI videos from screen recordings and instructional videos on YouTube. Subsequently, we utilize a Human-MLLM collaborative approach to generate a diverse set of captions, complex queries, and multi-round conversation, ultimately constructing GUI-WORLD.

Likewise, we also establish a comprehensive benchmark for GUI understanding, which encompasses nine mainstream MLLMs (*e.g.*, GPT-4o (OpenAI, 2024) and Gemini-1.5-Pro (Team et al., 2023)), five keyframe selection strategies (*e.g.*, UVD (Zhang et al., 2024d)), and six GUI scenarios, aiming to provide a thorough evaluation of the GUI-oriented understanding capabilities of MLLMs. As shown in Section 2, the assessment results indicate that most MLLMs struggle with GUI-WORLD, highlighting their limited dynamic understanding of graphical interfaces and underscoring the need for further enhancement.

Using this dataset, we take the first step to fine-tune a GUI-oriented Video LLM that excels at handling dynamic and sequential GUI tasks, leading to substantial enhancements in the general capabilities and

Table 1: Comparison of GUI datasets and benchmarks. **Sem.**: semantic instruction level, **VL**: Vision-Language, **Seq.**: Tasks for sequential images, **Cro.**: Cross-app or multi-window tasks, **Dyn.**: Tasks for dynamic GUI content.

| Dataset | Size | Sem. | VL | Video | Env Type | | | | Task Coverage | | | Task |
| | | | | | Web. | Mob. | Desk. | XR | Seq. | Cro. | Dyn. | |
| --- | --- | --- | --- | --- | --- | --- | --- | --- | --- | --- | --- | --- |
| Rico (Deka et al., 2017) | 72,219 | Low | ✔ | ✔ | ✗ | ✔ | ✗ | ✗ | ✔ | ✔ | ✗ | UI Code/Layout Generation |
| MiniWoB++ (Liu et al., 2018) | 100 | Low | ✔ | ✗ | ✔ | ✗ | ✗ | ✗ | ✗ | ✗ | ✗ | Web Navigation |
| Screen2Words (Wang et al., 2021) | 22,417 | High | ✔ | ✗ | ✗ | ✔ | ✗ | ✗ | ✗ | ✗ | ✗ | UI Summarization |
| MetaGUI (Sun et al., 2022) | 1,125 | Low | ✔ | ✗ | ✗ | ✔ | ✗ | ✗ | ✔ | ✗ | ✗ | Mobile Navigation |
| UGIF (Venkatesh et al., 2022) | 523 | High | ✔ | ✗ | ✗ | ✔ | ✗ | ✗ | ✔ | ✗ | ✗ | Instruction Following |
| AITW (Rawles et al., 2023) | 715,142 | High | ✔ | ✗ | ✗ | ✔ | ✗ | ✗ | ✔ | ✔ | ✗ | GUI Understanding |
| Ferret-UI (You et al., 2024) | 123,702 | Low | ✔ | ✗ | ✗ | ✔ | ✗ | ✗ | ✗ | ✗ | ✗ | UI Grounding & Understanding |
| Spotlight (Li & Li, 2022) | 2.5M | Low | ✔ | ✗ | ✗ | ✔ | ✗ | ✗ | ✗ | ✗ | ✗ | GUI Understanding |
| WebArena (Zhou et al., 2023) | 812 | Low | ✔ | ✗ | ✔ | ✗ | ✗ | ✗ | ✔ | ✗ | ✗ | Web Navigation |
| Mind2Web (Deng et al., 2024) | 2,350 | Both | ✔ | ✗ | ✔ | ✗ | ✗ | ✗ | ✔ | ✗ | ✗ | Web Navigation |
| OmniAct (Kapoor et al., 2024) | 9,802 | Low | ✔ | ✗ | ✔ | ✗ | ✔ | ✗ | ✔ | ✗ | ✗ | Code Generation |
| GUICourse (Chen et al., 2024c) | 10.7M | Both | ✔ | ✗ | ✔ | ✔ | ✗ | ✗ | ✔ | ✔ | ✗ | GUI Understanding |
| MMINA (Zhang et al., 2024e) | 1,050 | Low | ✔ | ✗ | ✔ | ✗ | ✗ | ✗ | ✔ | ✔ | ✗ | Web Navigation |
| AgentStudio (Zheng et al., 2024b) | 304 | High | ✔ | ✗ | ✔ | ✗ | ✔ | ✗ | ✔ | ✔ | ✗ | General Control |
| OSWorld (Xie et al., 2024) | 369 | High | ✔ | ✗ | ✔ | ✗ | ✔ | ✗ | ✔ | ✔ | ✗ | General Control |
| **GUI-WORLD (Ours)** | 12,379 | Both | ✔ | ✔ | ✔ | ✔ | ✔ | ✔ | ✔ | ✔ | ✔ | GUI Understanding Instruction Following |

showcasing the utility and effectiveness of GUI-WORLD. Additionally, we delve into discussing various factors critical to GUI understanding, including the integration of textual information, the number of keyframes, image resolutions, and vision perception, providing a pioneering and comprehensive study of the GUI domain.

Overall, the key contributions of this paper are threefold:

- **A New Dataset.** We propose GUI-WORLD, a comprehensive GUI dataset comprising 12,379 videos specifically designed to assess and improve GUI-oriented capabilities of MLLMs, spanning a range of categories and scenarios, including desktop, mobile, and XR environments. It stands as the first GUI-oriented instruction-tuning dataset in video domain.
- **Comprehensive Experiments and Valuable Insights.** Our experiments indicate that most existing MLLMs continue to face challenges with GUI-oriented tasks, particularly in sequential and dynamic GUI content. Empirical results suggest that enhancing vision perception, such as more keyframes and higher resolution, can lead to substantial performance improvements in GUI tasks.
- **A Explorative GUI-oriented Video LLM.** Based on GUI-WORLD, we propose GUI-VID, a GUI-oriented video LLM with enhanced capabilities to handle various and complex GUI tasks. GUI-VID shows a significant improvement on the benchmark and achieves results comparable to the top-performing models, thereby paving ways for the future of GUI models.

## 2 GUI-WORLD: A DATASET FOR GUI UNDERSTANDING

### 2.1 OVERVIEW

We introduce GUI-WORLD, a comprehensive dataset covering six GUI scenarios including video, human-annotated keyframes, as well as detailed captions and diverse types of QA produced by our data curation framework, aiming at benchmarking and enhancing the general GUI-oriented

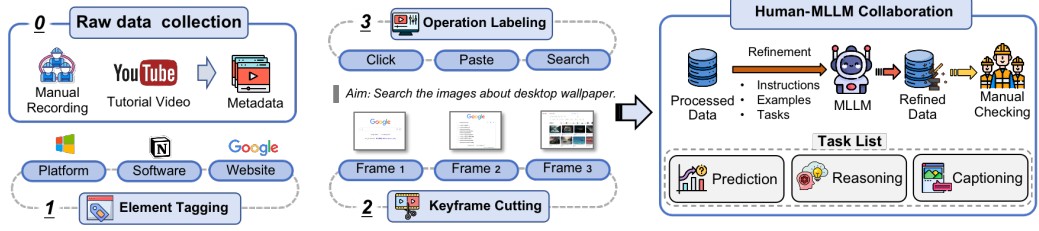

Figure 3: An overview construction pipeline of GUI-WORLD.

Table 2: The statistics of GUI-WORLD. For Android, we select videos from Rico (Deka et al., 2017) and randomly sample 10 frames. **Avg. Frame** refers to the average number of frames in each clip, and **Avg. Anno.** refers to the average number of manually annotated GUI actions.

| Category | Total Videos | Free-form | MCQA | Conversation | Total Frame. (Avg.) | Avg. Anno. |
|---|---|---|---|---|---|---|
| Software | 4,720 | 27,840 | 9,440 | 9,440 | 23,520 (4.983) | 7.558 |
| Website | 2,499 | 14,994 | 4,998 | 4,998 | 15,371 (6.151) | 6.862 |
| IOS | 492 | 2,952 | 984 | 984 | 2,194 (4.459) | 7.067 |
| Multi | 475 | 2,850 | 950 | 950 | 2,507 (5.277) | 7.197 |
| XR | 393 | 2,358 | 786 | 786 | 1,584 (4.030) | 10.970 |
| Android | 3,800 | 15,199 | 7,600 | 7,600 | 38,000 (10.000) | - |
| Summary | 12,379 | 76,673 | 24,758 | 24,758 | 83,176 (6.719) | 7.463 |

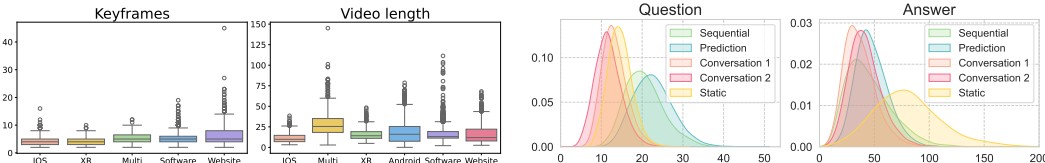

Figure 4: **Left:** Distribution of the number of keyframes and video lengths. **Right:** Length distribution for each type of question and its golden answer.

capabilities. These GUI scenarios encompass desktop operating systems (*e.g.*, macOS, Windows) and mobile platforms (*e.g.*, Android and iOS), websites, software, and even XR (*e.g.*, GUI in Apple Vision Pro (Apple, 2024)). We divide the dataset into a train-test split, each containing 10,702 and 1,677 samples. Discussion for each scenario is referred to Appendix A.1.

As illustrated in Figure 3, the development of GUI-WORLD follows a two-stage process. Further details about video and query statistics are outlined in Table 2, including distributions of keyframe counts, video durations, query lengths, and their corresponding golden answers. Figures 4 and 5 further illustrate these data distributions.

## 2.2 GUI VIDEO COLLECTION AND KEYFRAME ANNOTATION PROCESS

We describe the pipeline for collecting screen recordings from student workers and GUI-related instructional videos from YouTube for GUI-WORLD and the procedures followed to convert these videos into keyframe sequences.

A significant portion of our video data is derived from screen recordings executed by student workers, which can directly reflect real-life GUI usage scenarios. A typical video collection scenario involves assigning a student worker a specific software task. The student begins by familiarizing themselves with the software, followed by recording a series of operations in a short video clip, such as "Sign up", "Sign in", "Create a New Page", and "Invite Other Collaborators" in the software "Notion[1]".

Despite the high fidelity of these manually recorded videos, we encounter several challenges: (1) Student workers often require substantial time to acquaint themselves with professional software (*e.g.*, MATLAB, Adobe After Effects (Ae)), which can hinder the progress of data collection. (2) The videos may lack comprehensiveness, typically capturing only commonly used operations and overlooking rarer functions crucial for dataset completeness. To address these issues, we also source videos from social media platforms that host a diverse array of GUI videos. Specifically, we download tutorial videos from YouTube—given its prevalence as a video-sharing platform—because they richly detail various GUI operations. These videos are then segmented into shorter clips, each representing a distinct sequence of operations.

The subsequent step involves annotating these video clips with keyframes and textual descriptions of each keyframe using custom-designed annotation software. Although several algorithms exist for keyframe extraction (Zhu et al., 2016; Yan et al., 2018; Mahasseni et al., 2017; OpenCV), they typically underperform with GUI videos where changes between frames might be minimal (*e.g.*, a slight movement in the mouse cursor). To ensure high-quality datasets, we therefore perform manual

---
[1] https://www.notion.so/

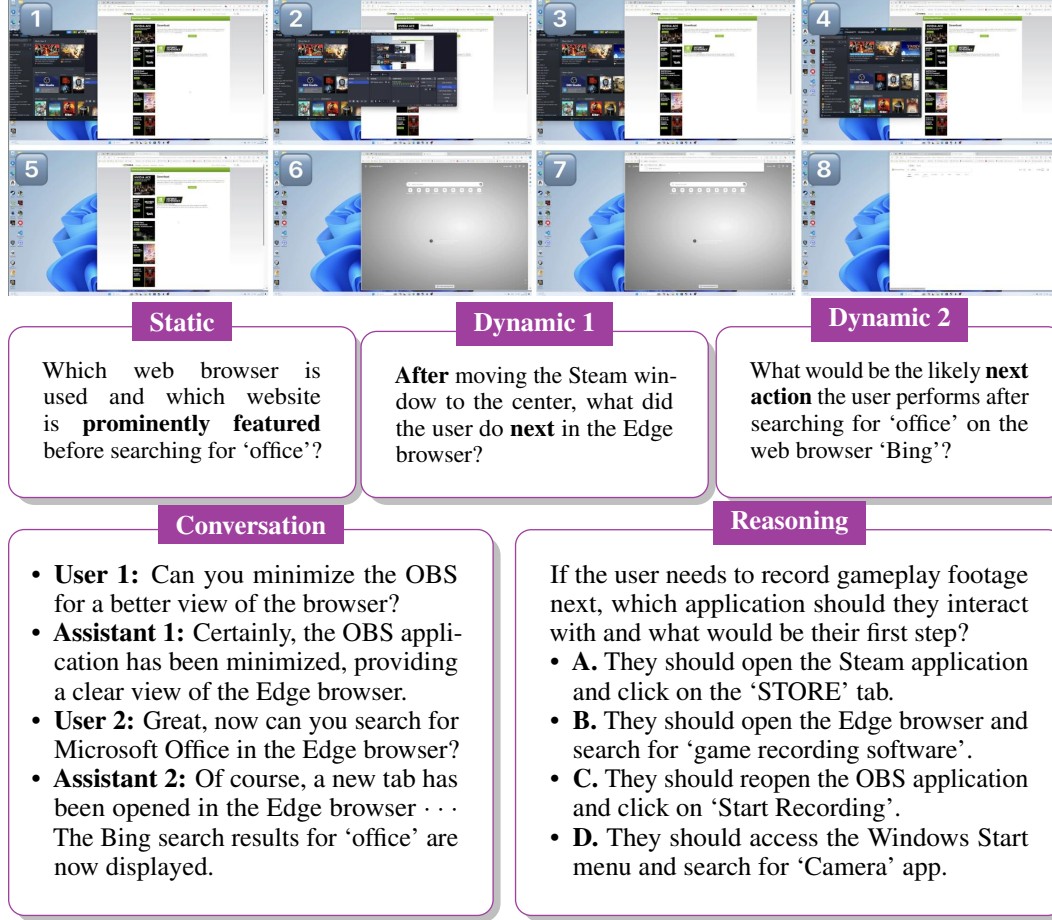

**Static**

Which web browser is used and which website is **prominently featured** before searching for 'office'?

**Dynamic 1**

**After** moving the Steam window to the center, what did the user do **next** in the Edge browser?

**Dynamic 2**

What would be the likely **next action** the user performs after searching for 'office' on the web browser 'Bing'?

**Conversation**

- **User 1:** Can you minimize the OBS for a better view of the browser?
- **Assistant 1:** Certainly, the OBS application has been minimized, providing a clear view of the Edge browser.
- **User 2:** Great, now can you search for Microsoft Office in the Edge browser?
- **Assistant 2:** Of course, a new tab has been opened in the Edge browser · · · The Bing search results for 'office' are now displayed.

**Reasoning**

If the user needs to record gameplay footage next, which application should they interact with and what would be their first step?
- **A.** They should open the Steam application and click on the 'STORE' tab.
- **B.** They should open the Edge browser and search for 'game recording software'.
- **C.** They should reopen the OBS application and click on 'Start Recording'.
- **D.** They should access the Windows Start menu and search for 'Camera' app.

Figure 5: An example in multi-window GUI scene as a case study.

extraction of these keyframes. Each keyframe is meticulously annotated to include details such as the operation performed, the purpose between two keyframes, the software or website used, mouse actions (*e.g.*, scroll, click), and keyboard inputs (*e.g.*, copy (Ctrl + C), paste (Ctrl + V), specific input). We detail our annotation process in Appendix A.3.

## 2.3 GUI TASKS GENERATION FROM HUMAN-MLLM COLLABORATION

Drawing insights from prior research (Dekoninck et al., 2024), we develop a Human-MLLM collaboration pipeline to annotate captions and diverse types of QA specifically tailored for GUI comprehension. The process involves inputting an instructional prompt, a comprehensive description, key information (*e.g.*, system or application), and a sequence of human-annotated keyframes into GPT-4V. As depicted in Table 11, GUI-WORLD features various question types, detailed as follows:

▷ **Detailed and Summarized Captioning:** This task challenges basic GUI knowledge and multimodal perception, also addressing the deficiency of detailed GUI content in video-caption pairs. Initially, GPT-4V generates two distinct descriptions for each video: one concentrating on fine-grained details and the other on overall information. Furthermore, GPT-4V provides a succinct summary, highlighting core operations and overarching objectives in the video.

▷ **Static GUI Content:** This task challenges MLLM with textual, layout, and iconographic analysis of static GUI content. We instruct GPT-4V to generate free-form queries with a golden answer concerning static GUI elements or specific scenes that recur in more than two keyframes, ensuring their consistent presence in the video. Additionally, GPT-4V also crafts QA pairs that evaluate inferential skills in static content, focusing on interrelations among icons or textual information.

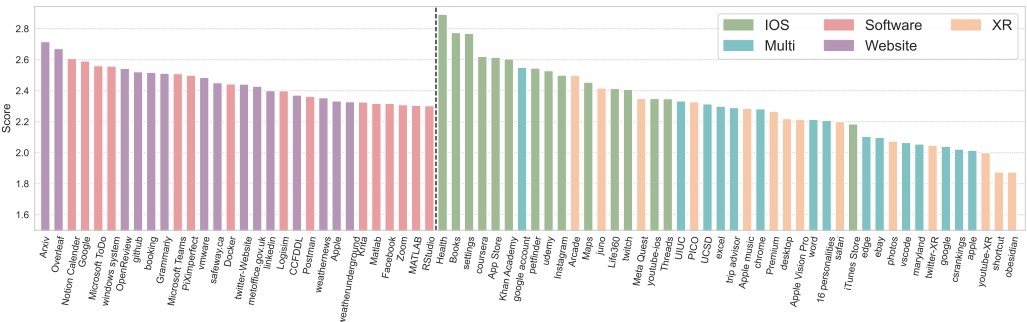

Figure 6: Fine-grained performance of GPT-4V in each GUI scenario (w.o. Android).

▷ **Dynamic and Sequential GUI Content:** This task concentrates on temporal content in GUI video, such as dynamically changing interfaces, and aims to elucidate the sequential information and reasoning chains within GUI content. We direct GPT-4V to identify consistently changing elements to create queries for dynamic content. Moreover, predictive tasks are formulated on order and temporal relation in provided sequential images, challenging agents to anticipate future events or states.

In the last stage, human annotators will follow the guideline in Appendix A.3 and carefully review the entire video and MLLM-generated QA pairs to correct inaccuracies and hallucinations, as well as supplement information for both questions and answers to make these tasks more challenging.

# 3 EXPERIMENTS AND ANALYSIS

## 3.1 EXPERIMENTAL SETUPS

**Models.**[2]   We conduct evaluations on five of the most popular vision LLMs: GPT-4V(ision) (OpenAI, 2023), GPT-4o (OpenAI, 2024), Qwen-VL-Max (Bai et al., 2023), LLaVA-OV-7B (Li et al., 2024a), and Gemini-Pro-1.5 (Team et al., 2023). Additionally, we test the effect of different vision inputs on GPT-4o, using no input, low and high-resolution settings, as well as without providing images, to further assess how resolution influences performance. Each model's responses employ a three-step Chain-of-Thought (CoT) (Wei et al., 2022) process, *i.e.*, *"Describe-Analyze-Answer"*, to evaluate their peak performance. Additionally, we assessed four advanced video LLMs—ChatUnivi (Jin et al., 2023), Minigpt4-video (Ataallah et al., 2024), Videochat2 (Li et al., 2023c), VideoLLaVA (Lin et al., 2023a) —for their performance on GUI content. See Appendix C for detailed setups.

**Evaluation Metrics.**   To assess free-form questions and multiple-round conversations, we utilize the LLM-as-a-Judge methodology, which assigns a similarity score ranging from 1 to 5 between MLLM's response and a predefined golden answer, already validated by previous studies (Zheng et al., 2023; Liu et al., 2023d; Chen et al., 2024a; Ye et al., 2024). For multiple-choice questions, we measure performance using accuracy as the primary evaluation metric.

**Keyframe Extraction.**   We benchmark on three keyframe selection settings: (1) *Linspave*, where frames are evenly sampled at fixed time intervals within a video; (2) *Program*, with programmatic method Katna (KeplerLab, 2023); (3) *Model-based*, which leverages pre-trained vision representation from VIP (Ma et al., 2022) and R3M (Nair et al., 2022) to form UVD (Zhang et al., 2024d); and (4) *Human*, where humans select keyframes during the annotation process. **We use *Human* setting for Image LLMs in our main experiment with an average of 6.719 frames (Table 2). For all other settings, we input 10 frames into each MLLM.**

**Additional Information Integration.**   To investigate the effectiveness of integrating image-caption models for LLMs—typically employed in natural videos—and the helpfulness of textual GUI content in accomplishing GUI-oriented tasks, we implement three experimental settings: Detailed Caption, Concise Caption, and Vision + Detailed Caption. GPT-4V is utilized to provide captions of these keyframes, integrating human annotators' operational intents to more accurately describe each frame, being validated in Appendix A.3.

---

[2]Given that GPT-4V is announced to be deprecated, we use GPT-4o for certain ablation studies to ensure that our results provide longer-term reference value.

Table 3: The overall performance in six GUI scenarios for MCQA and Free-form queries. **MC** - Multiple-Choice QA. **Free** - average score of all free-form and conversational queries.

| Models | Software MC | Free | Website MC | Free | XR MC | Free | Multi MC | Free | IOS MC | Free | Android MC | Free | Avg. MC | Free |
|---|---|---|---|---|---|---|---|---|---|---|---|---|---|---|
| **Image LLMs** | | | | | | | | | | | | | | |
| LLaVA-OV-7B | 56.9% | 2.641 | 48.4% | 2.588 | 59.6% | 2.709 | 52.9% | 2.306 | 58.1% | 2.717 | 24.3% | 2.675 | 50.0% | 2.606 |
| Gemini-Pro-1.5 | 82.9% | 3.385 | 79.2% | 3.412 | 83.3% | 3.108 | **83.4%** | 3.246 | 80.3% | 3.467 | 78.5% | 3.168 | 81.3% | 3.298 |
| Qwen-VL-Max | 75.8% | 2.651 | 75.5% | 2.698 | 77.6% | 2.373 | 66.9% | 2.490 | 74.3% | 2.633 | 74.2% | 2.559 | 74.0% | 2.568 |
| GPT-4V | 86.0% | 3.520 | 79.8% | 3.655 | 83.4% | 3.265 | 76.9% | 3.449 | 79.9% | 3.453 | 81.3% | 3.466 | 81.2% | 3.469 |
| GPT-4o | **86.5%** | **3.644** | **83.3%** | **3.740** | **84.3%** | **3.285** | 81.1% | **3.654** | **83.3%** | **3.558** | **90.0%** | **3.561** | **84.8%** | **3.573** |
| **Video LLMs** | | | | | | | | | | | | | | |
| ChatUnivi | 28.4% | 2.389 | 22.2% | 2.349 | 20.6% | 2.161 | 17.5% | 2.275 | 22.6% | 2.337 | 23.0% | 2.390 | 22.4% | 2.317 |
| Minigpt4Video | 18.9% | 1.475 | 15.3% | 1.520 | 16.3% | 1.362 | 15.4% | 1.457 | 20.1% | 1.501 | 14.6% | 1.342 | 16.8% | 1.443 |
| VideoChat2 | 45.5% | 2.144 | 42.6% | 2.221 | 44.0% | 2.005 | 40.4% | 2.222 | 40.2% | 2.169 | 44.7% | 2.119 | 42.9% | 2.147 |
| Video-LLaVA | 52.9% | 2.290 | 52.4% | 2.410 | 44.2% | 2.258 | 45.9% | 2.329 | 49.7% | 2.319 | 51.3% | 2.259 | 49.4% | 2.311 |
| GUI-VID | **59.9%** | **2.847** | **54.1%** | **2.957** | **55.6%** | **2.764** | **52.9%** | **2.861** | **51.8%** | **2.773** | **53.4%** | **2.572** | **54.6%** | **2.796** |

Table 4: Overall performance in six GUI scenarios for MCQA and Free-form queries. **D.C.** means detailed caption, and **C.C.** means concise caption, and ✗ means no vision input.

| Models | Setting Vision | Text | Software MC | Free | Website MC | Free | XR MC | Free | Multi MC | Free | IOS MC | Free | Android MC | Free | Avg. MC | Free |
|---|---|---|---|---|---|---|---|---|---|---|---|---|---|---|---|---|
| | ✗ | D.C. | **85.0%** | 3.350 | 83.1% | 3.380 | 82.3% | 3.056 | **84.2%** | 3.358 | **81.6%** | 2.751 | 81.7% | 3.427 | 83.0% | 3.316 |
| GPT-4V | ✗ | C.C. | 80.7% | 3.028 | 72.2% | 3.025 | 82.8% | 2.809 | 81.3% | 3.160 | 76.5% | 2.868 | 76.4% | 2.939 | 78.3% | 2.971 |
| | ✔ | D.C. | 82.5% | **3.494** | **83.2%** | **3.682** | **85.9%** | **3.191** | 83.9% | **3.617** | 80.9% | **3.516** | **84.9%** | **3.758** | **83.5%** | **3.543** |

Table 5: Detailed scores for free-form tasks in the software-related scenarios. **Dyn.** refers to queries on dynamic GUI content.

| Models | Caption Concise | Detailed | Complex Tasks Static | Dyn. | Conversation Round 1 | Round 2 | Average |
|---|---|---|---|---|---|---|---|
| **Image LLMs** | | | | | | | |
| LLaVA-OV-7B | 2.149 | 1.762 | 1.868 | 2.448 | 2.947 | 3.492 | 2.641 |
| Gemini-Pro-1.5 | 3.306 | **3.035** | 2.945 | 3.093 | 3.573 | 3.790 | 3.298 |
| Qwen-VL-Max | 2.474 | 1.711 | 2.137 | 2.433 | 3.223 | 3.257 | 2.651 |
| GPT-4V | 3.352 | 2.509 | 3.053 | 3.229 | 3.928 | 4.163 | 3.520 |
| GPT-4o | **4.048** | 3.028 | **3.125** | **3.340** | **4.129** | **4.318** | **3.644** |
| **Video LLMs** | | | | | | | |
| ChatUnivi | 1.587 | 1.240 | 1.705 | 2.090 | 2.698 | **3.366** | 2.389 |
| Minigpt4Video | 1.246 | 1.073 | 1.249 | 1.455 | 1.494 | 1.719 | 1.475 |
| VideoChat2 | 1.992 | 1.312 | 1.812 | 1.920 | 2.342 | 2.720 | 2.144 |
| Video-LLaVA | 1.519 | 1.241 | 1.657 | 1.959 | 2.587 | 3.293 | 2.290 |
| GUI-VID | **3.562** | **2.058** | **2.376** | **2.763** | **3.080** | 3.260 | **2.847** |

## 3.2 EMPIRICAL RESULTS

**Commercial Vision LLMs outperform Open-source Video LLMs in Zero-shot Settings.** Commercial vision LLMs, notably GPT-4V and GPT-4o, consistently outperform open-source video LLMs in zero-shot settings. As detailed in Table 3, GPT-4o exhibits superior performance across all GUI scenarios in complex tasks, reflected in its high scores in both multiple-choice and free-form queries, with an average of 84.8% and 3.573. Similarly, Gemini demonstrates strong capabilities in captioning and descriptive tasks within software and iOS environments, scoring 2.836 and 2.936, respectively, as shown in Table 26. Further analysis (Figure 6) reveals that GPT-4V excels in applications with minimal textual content and simple layouts, such as TikTok, health apps, and GitHub. In contrast, its performance drops in more intricate applications like Microsoft ToDo and XR software. As for video LLMs, their significantly poorer performance is attributed to two main factors: their inability to accurately interpret GUI content from user inputs and a lack of sufficient GUI-oriented pre-training, which is evident from their inadequate performance in basic captioning and description tasks. See Appendix D for other metrics and detailed fine-grained performance.

**Dynamic GUI Tasks Continue to Challenge MLLMs.** In the fine-grained tasks depicted in Table 5, GPT-4V and GPT-4o excel with dynamic GUI content and conversational tasks but struggle with providing detailed descriptions for entire videos and static content. This discrepancy is attributed to minor variations in GUI that significantly impact its semantic meaning. Enhancing the number of keyframes and the granularity of perception might mitigate these issues. Among video LLMs,

Table 6: Performance comparison of keyframe selection methods for GPT-4o: *Model-based* keyframe identifiers from embodied AI demonstrate comparable performance to *human-selected* keyframes.

| Settings | Caption | | Complex Tasks | | Conversation | | Average |
|---|---|---|---|---|---|---|---|
| | Concise | Detailed | Static | Dyn. | Round 1 | Round 2 | |
| Human | 3.911 | 3.031 | 3.131 | **3.318** | **3.981** | **4.132** | 3.573 |
| Program | 3.643 | 2.764 | 2.872 | 3.052 | 3.702 | 3.837 | 3.300 |
| Linspace | 3.749 | 2.941 | 3.000 | 3.077 | 3.687 | 3.843 | 3.440 |
| UVD+vip | 3.954 | 3.105 | 3.321 | 3.219 | 3.944 | 4.107 | 3.581 |
| UVD+r3m | **3.972** | **3.121** | **3.352** | 3.243 | 3.975 | 4.119 | **3.612** |

Table 7: The improved performance with higher resolution inputs demonstrates the critical role of vision input in GUI-related tasks.

| Setting | Caption | | Complex Tasks | | Conversation | | Average |
|---|---|---|---|---|---|---|---|
| | Concise | Detailed | Static | Dyn. | Round 1 | Round 2 | |
| w/o Vision | 2.187 | 1.872 | 2.486 | 2.979 | 3.760 | 4.059 | 2.891 |
| Low Resolution | 3.672 | 2.794 | 2.869 | 3.150 | 3.783 | 4.041 | 3.394 |
| High Resolution | **3.911** | **3.031** | **3.131** | **3.318** | **3.981** | **4.132** | **3.574** |

ChatUnivi excels in conversational tasks by effectively leveraging contextual nuances, particularly in subsequent rounds, yet it underperforms in caption tasks. In contrast, GUI-VID demonstrates proficiency in dynamic tasks but falls short in both captioning and static content. This gap is linked to deficiencies in backbone pretraining, which lacked comprehensive GUI content crucial for effective vision-text alignment, as evidenced by its poor performance in simple caption task shown in Table 26 and an instruction tuning process failed to fully address these shortcomings.

**Vision Perception is Important to Dynamic GUI Content.** As demonstrated in Table 7, integrating detailed textual information slightly outperforms purely vision-based inputs or detailed captions, akin to a Chain of Thought (CoT) (Wei et al., 2022) setting. Surprisingly, GPT-4V excels in caption tasks with just detailed captions, providing insights on enhancing specific GUI-oriented tasks through additional textual information. However, it still falls short in more challenging tasks, such as retrieving static or dynamic content. This underscores the critical role of visual perception in GUI environments, where even minor changes can significantly impact outcomes.

**Keyframe Selection is Important for GUI-oriented Tasks.** Our experiments demonstrate that *model-based* keyframe identifiers, originally developed for embodied AI applications, perform competitively with *human-selected* across both basic tasks (*e.g.*, caption) and complex tasks (static and dynamic analysis). As shown in Table 6, GPT-4o exhibits significant performance improvements when utilizing these robotics-inspired model-based keyframe identifiers, with the UVD+VIP approach achieving optimal results. These findings suggest the potential to replace manual keyframe selection with automated approaches. Further analysis reveals that embodied AI keyframe identifiers successfully capture semantic transitions in GUI content, while *linspace* and *program*-based selection methods fail to do so, highlighting their particular suitability for GUI-oriented tasks. The substantial performance gaps observed between different selection methods underscore the critical importance of keyframe selection in this domain.

# 4 EXPLORING AND IMPROVING GUI-ORIENTED VIDEO LLMS

## 4.1 METHOD: PROGRESSIVE ENHANCEMENT

We introduce our strategy to enhance the GUI-oriented capabilities of current MLLMs on both static and dynamic GUI content. Inspired by previous studies (Lai et al., 2024; Li et al., 2023b), we structure our methodology into two distinct fine-tuning stages, as illustrated in Figure 7. Initially, we fine-tune the MLLM on simpler tasks, such as description queries and captioning exercises, to instill a basic understanding of GUI elements. Subsequently, building on this foundation, the second stage aims to augment the MLLM's proficiency with more complex and challenging tasks. Our fine-tuning is all based on the Supervised Fine-Tuning (SFT): $\mathcal{L}_{\text{SFT}}(\pi_\theta) = -\mathbb{E}_{(x,y)\sim\mathcal{D}}[\log \pi_\theta(y \mid x)]$, where $x$ is the input, $y$ is LLMs' output, and $\pi_\theta$ denotes the model parameters that need to be optimized.

**Stage-1: Learning Preliminary for GUI Content.** The initial phase focuses on aligning GUI content with a pre-trained vision encoder and a base LLM, utilizing GUI videos accompanied by detailed descriptions and captions. This phase aims to embed a robust understanding of fundamental

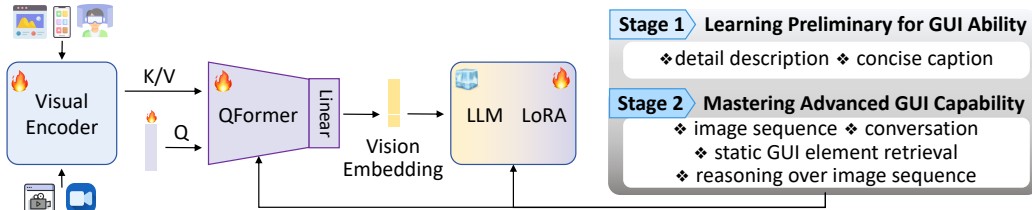

Figure 7: An overview of our fine-tuning architecture, focusing on *1)* GUI content alignment and *2)* GUI-oriented tasks instruction tuning.

GUI concepts and terminology within the MLLM. By engaging the model in basically captioning various GUI components, the model learns to recognize and articulate the functionalities and visual characteristics of these elements, thereby laying a solid groundwork for GUI knowledge.

**Stage-2: Mastering Advanced GUI Capability.** Building on the foundational knowledge established in Stage 1, the second stage focuses on advancing the MLLM's proficiency in interacting with GUI elements through more complex tasks. These tasks are designed to simulate real-world scenarios that the MLLM might encounter in GUI environments, which include predicting based on image sequences, engaging in conversations, retrieving both static and dynamic GUI elements, and performing reasoning tasks.

As illustrated in Figure 7, We employ the two-stage training architecture utilizing VideoChat2 (Li et al., 2023b) as our foundational model. Initially, videos and images are encoded using the UMT-L visual encoder (Li et al., 2023d). Subsequently, a QFormer compresses visual tokens into a smaller set of query tokens. Drawing inspiration from (Dai et al., 2023), we enhance the QFormer (Zhang et al., 2024c) by integrating instructions to enable it to extract visual representations pertinent to the given instructions. Additionally, we apply low-rank adaptation (LoRA (Hu et al., 2021)) to base LLM. This model is concurrently fine-tuned with the visual encoder and QFormer using a Vision-grounded Text Generation (VTG) loss: $\mathcal{L}_{\text{VTG}}(\theta) = -\mathbb{E}\left[\log p(y|v;\theta)\right]$, where $v$ represents the visual tokens derived from the QFormer, and $y$ represents the text output grounded in the visual context.

## 4.2 EXPERIMENTS

**Experiment Setups.** We use two dataset settings to fine-tune GUI-VID, one with video only, and the other with video and image, detailed in Appendix C. We also vary the number of keyframes (8, 16) fed into GUI-VID. All our experiments are conducted on A800 and 4090 GPUs.

**Supreme Enhancement of GUI-VID on Graphic-based Interface After Fine-tuning on GUI-WORLD.** As a pioneering study in training video LLMs as screen agents, GUI-VID significantly outperforms the baseline model, showing an average improvement of 30% across various tasks and GUI scenarios, even surpassing the commercial vision LLM, Qwen-VL-Max. This enhancement is particularly notable in captioning and dynamic task that reason over image sequences, where GUI-VID matches the performance of GPT-4V and Gemini-Pro. As depicted in Table 8, our two ablation studies during the fine-tuning phase demonstrate that utilizing GUI image-text captioning data significantly enhances the model's preliminary understanding of GUI elements, outperforming

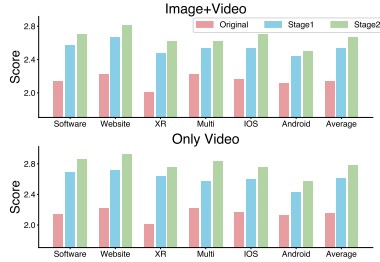

Figure 8: Two stages of progressive training enhance GUI ability.

training that relies solely on videos. Additionally, an increased number of keyframes correlates with improved performance across various scenarios, notably in environments featuring multiple windows and software applications. As shown in Figure 8, our two-stage progressive fine-tuning significantly enhances the performance in all GUI scenarios.

**Correlation between GUI Understanding and Other Mainstream GUI Tasks.** In our explorative experiments, GUI-VID still fails in some GUI operating tasks via code generation, which is due to the baseline LLM's weak performance and the challenges of code generation instruction fine-tuning. To further demonstrate how GUI understanding capability enhances mainstream GUI-related tasks,

Table 8: The overall results for ablation study on GUI-VID fine-tuning. F.K. and E.K. mean keyframes during the finetuning and evaluation process respectively. **I.**: Image, **V.**: Video.

| Baseline | F.K. | E.K. | Data I. | V. | Software MC | Free | Website MC | Free | XR MC | Free | Multi MC | Free | IOS MC | Free | Android MC | Free | Avg. MC | Free |
|---|---|---|---|---|---|---|---|---|---|---|---|---|---|---|---|---|---|---|
| Baseline | - | 8 | - | - | 45.5% | 2.144 | 42.6% | 2.221 | 44.0% | 2.005 | 40.4% | 2.222 | 40.2% | 2.169 | 44.7% | 2.119 | 42.9% | 2.147 |
|  | - | 16 | - | - | 45.1% | 2.144 | 41.8% | 2.240 | 41.0% | 2.007 | 40.7% | 2.238 | 39.9% | 2.138 | 44.7% | 2.147 | 42.2% | 2.154 |
| GUI-VID | 8 | 8 | ✗ | ✔ | 58.3% | 2.709 | 53.6% | 2.817 | 62.2% | 2.626 | **54.2%** | 2.627 | 53.1% | 2.708 | 54.9% | 2.501 | 56.0% | 2.665 |
|  |  |  | ✔ | ✔ | **59.9%** | **2.856** | 54.1% | 2.925 | 59.0% | 2.751 | 52.1% | 2.837 | 50.0% | 2.756 | 54.0% | 2.571 | 54.8% | 2.782 |
|  |  | 16 | ✗ | ✔ | 59.0% | 2.709 | **55.1%** | 2.821 | **62.8%** | 2.645 | 53.3% | 2.624 | **55.5%** | 2.727 | **55.7%** | 2.501 | **56.9%** | 2.671 |
|  |  |  | ✔ | ✔ | **59.9%** | 2.847 | 54.1% | **2.957** | 55.6% | **2.764** | 52.9% | **2.861** | 51.8% | **2.772** | 53.4% | **2.572** | 54.6% | **2.796** |

including generating operational code (Cheng et al., 2024) and providing chat assistance (Hong et al., 2024), we conduct experiments as follows (detailed in Appendix D):

- We evaluate our model on Mind2Web-Multimodal (Deng et al., 2024) in both zero-shot and fine-tuning setting. See Appendix D for further details.
- We conduct human study across 180 videos across 6 scenarios, where annotators will choose preferred responses from two response of different models when acting as GUI assistant (Table 17).

## 5 RELATED WORK

**MLLM-based Agents for GUI.** Building upon the significant advancements in LLMs (Achiam et al., 2023; Meta, 2023a;b; ai, 2024; Huang et al., 2025) and advanced modality-mixing technologies (Li et al., 2023a; Alayrac et al., 2022), groundbreaking MLLMs such as GPT-4V (OpenAI, 2023) and Gemini-Pro (Team et al., 2023), along with open-source MLLMs like the LLaVA-1.6 series (Liu et al., 2023b;a), CogVLM (Wang et al., 2023b), and Qwen-VL series (Bai et al., 2023), have shown outstanding performance across various tasks (Yu et al.; Liu et al., 2023e; Chen et al., 2024b; Wu et al., 2023; Wake et al., 2023; Zhang et al., 2024b; Zhao et al., 2024; Gui et al., 2024). Venturing beyond text and single image, several studies are now exploring the integration of video modalities for tasks requiring dynamic or sequential visual content (Jin et al., 2023; Li et al., 2023b; Maaz et al., 2023; Lin et al., 2023a). In the GUI domain, leveraging the robust vision perception capabilities of MLLMs, applications such as WebAgents (Hong et al., 2024; Zhang et al., 2024a; Zheng et al., 2024b) and Mobile Agents (Wang et al., 2023a; You et al., 2024; Wang et al., 2021; Li et al., 2020b) have gained popularity for handling everyday tasks like navigation and VQA. Frontier research is also investigating the use of MLLMs as general control agents, such as in playing computer games (Tan et al., 2024; Lin et al., 2023b) and serving as OS copilots (Song et al., 2024; Xie et al., 2024), paving the way for more complex GUI operations.

**GUI Benchmark & Dataset.** Building upon the foundational work of Rico (Deka et al., 2017), the first mobile GUI video dataset, and AitW (Rawles et al., 2023), which features 715k episodes of sequential images, research has extensively covered mobile (Sun et al., 2022; Li et al., 2020a; Zhang et al., 2023) and web GUI environments (Lù et al., 2024; Zhou et al., 2023; Yao et al., 2022; Koh et al., 2024; Liu et al., 2024b). Mind2Web (Deng et al., 2024) stands out in web-based datasets with over 2,000 tasks from 137 websites across 31 domains. Advances continue into desktop GUIs with new toolkits (Zheng et al., 2024b), benchmarks (Kapoor et al., 2024; Mialon et al., 2023), and frameworks (Zheng et al., 2024a; Liu et al., 2023c; Niu et al., 2024). Research on GUI also transfers from comprehending single images in a static workspace (Hong et al., 2024) to sequential operations or multi-hop scenarios (Xie et al., 2024; Zhang et al., 2024e), challenging the understanding and operation capability of these powerful models.

## 6 CONCLUSION

In this paper, we have introduced GUI-WORLD, a comprehensive GUI-oriented video dataset designed to benchmark and enhance understanding of virtual interfaces, especially sequential and dynamic tasks. This dataset extensively covers six scenarios and various tasks, addressing the previous research gap in comprehensively evaluating models' capabilities in graphic-based understanding. We conduct extensive benchmarks on leading MLLMs and the first video-LLM-based assistant GUI-VID fine-tuned on GUI-WORLD specifically for GUI-oriented content, achieving results comparable to top-performing models, providing detailed insights into enhancing GUI-related capabilities. We believe our work offers valuable insights for future research in dynamic GUI content understanding.

**Acknowledgements.** Pan Zhou is partially supported by National Natural Science Foundation of China (NSFC) under grant No. 62476107. Dongping Chen and Yao Wan are supported by the Fundamental Research Funds for the Central Universities (HUST: 62400001). We would like to thank Yinuo Liu, Zhengyan Fu, Shilin Zhang, Yu, Tianhe Gu, Haokuan Yuan, and Junqi Wang for their insightful feedback and support.

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

# Part I

# Appendix

## Table of Contents

Table 9: **Summary of main experiments and results.** *Task* and *Scenario* are two primary axes that we consider most important to show the evaluation results. The *task-specific* analysis shows performance across different capabilities such as image captioning, complex QA, and conversation; *scenario-specific* analysis evaluates performance across various application domains such as XR, iOS, etc.

| Table | Objective | Category |
|-------|-----------|----------|
| Table 3 | Comparative analysis of model performance across six GUI scenarios | Scenario-specific |
| Table 4 | Impact of textual information incorporation on GUI understanding | Scenario-specific |
| Table 5 | Fine-grained evaluation of free-form responses in software tasks | Task-specific |
| Table 6 | Assessment of different keyframe selection strategies | Task-specific |
| Table 7 | Analysis of vision input modalities and quality effects | Task-specific |
| Table 8 | Comprehensive evaluation of GUI-VID and its components | Scenario-specific |

## A  DETAILS OF DATASET CONSTRUCTION

### A.1  SIX MAIN GUI CATEGORIES

In earlier endeavors pertaining to GUI, such as those involving GUI testing (Kousar et al., 2023; Jorge et al., 2014; Kulesovs, 2015), the focus was segmented into GUIs for Website, Software, IOS and Android platforms. However, as a comprehensive GUI dataset, we include all potential GUI scenarios in our dataset to ensure that our data is the most comprehensive knowledge that the GUI agent needs to learn; we divide these scenarios into six categories:

- **Android.** This category focuses on the GUI scenarios that occur within the Android operating system, which is predominantly used on smartphones. Android's ubiquity in the mobile market has led to a wide variety of GUI designs and interaction patterns, making it a rich field for study. This category has been the subject of extensive scrutiny in scholarly works such as (Deka et al., 2017; Li et al., 2020a; Rawles et al., 2023; Cheng et al., 2024).

- **Software.** This category encapsulates the GUI scenarios arising within software applications, whether they are standalone programs or components of a larger suite. The diversity of software applications, from productivity tools to creative suites, offers a wide range of GUI scenarios for exploration. The literature is rich with research in this area, such as (Zhan et al., 2024).

- **Website.** This category is concerned with the GUI scenarios that manifest within a web browser. Given the ubiquity of web browsing in modern digital life, this category holds significant relevance. It holds a substantial representation in academic literature, with pioneering papers such as (Deng et al., 2024; Kapoor et al., 2024) proposing excellent GUI datasets for websites.

- **IOS.** This category zeroes in on the GUI scenarios that transpire within the iOS operating system, the proprietary system for Apple devices like the iPhone and iPad. The iOS platform is known for its distinct design aesthetics and interaction patterns, providing a unique context for GUI research. A number of studies, such as (Beltramelli, 2018; Yan et al., 2023) make use of GUI information in IOS.

- **Multi-Windows.** This category is dedicated to GUI scenarios that necessitate simultaneous interaction with multiple windows, a common occurrence in desktop environments where users often juggle between several applications or documents. Despite the common use of multi-window interaction in everyday GUI usage, there has been relatively little research into this area (Nakajima et al., 2013). The need for efficient multitasking in such scenarios presents unique challenges and opportunities for GUI design and interaction research. As of our knowledge, there are no specific datasets catering to these multi-window GUI scenarios.

- **XR.** XR encompasses Virtual Reality (VR), Augmented Reality (AR), and Mixed Reality (MR) (Rauschnabel et al., 2022). Given the advancements in XR technology and the growing accessibility of commercial-grade head-mounted displays (Apple, 2024; Met), XR has emerged as a novel medium for human-computer interaction. This necessitates the exploration of GUI within XR environments. In these scenarios, the GUI takes on a 3D, immersive form (Sanders et al., 2019), demanding the agent to comprehend and navigate a 3D space. The emerging field of XR presents a new frontier for GUI research, with unique challenges and opportunities due to its immersive and interactive nature. To date, as far as we are aware, there are no datasets that specifically address GUI in the realm of XR.

```
1  {
2      "system": "Windows",
3      "app": [
4          "edge, bing, steam"
5      ],
6      "region": "partial",
7      "goal": "View the submission interface for the dataset and benchmark
            track of nips 2024.",
8      "keyframes": [
9          {
10             "frame": 32,
11             "sub_goal": "Click to start downloading, restart downloading
                   lethal company.",
12             "mouse": "click",
13             "keyboard": "none",
14             "keyboardOperation": ""
15         },
16         {
17             "frame": 176,
18             "sub_goal": "Click on edge, edge returns to the top of the
                   screen.",
19             "mouse": "click",
20             "keyboard": "none",
21             "keyboardOperation": ""
22         },
23         {
24             "frame": 781,
25             "sub_goal": "Click on the hyperlink for dataset and benchmark
                   , preparing to jump.",
26             "mouse": "click",
27             "keyboard": "none",
28             "keyboardOperation": ""
29         },
30         {
31             "frame": 839,
32             "sub_goal": "Jump to openreview, loading.",
33             "mouse": "click",
34             "keyboard": "none",
35             "keyboardOperation": ""
36         },
37         {
38             "frame": 1079,
39             "sub_goal": "The webpage loaded the submission interface for
                   dataset and benchmark track.",
40             "mouse": "none",
41             "keyboard": "none",
42             "keyboardOperation": ""
43         },
44         {
45             "frame": 1131,
46             "sub_goal": "Place the mouse on \"add a submission\"",
47             "mouse": "hover",
48             "keyboard": "none",
49             "keyboardOperation": ""
50         }
51     ]
52 }
```

Figure 9: Metadata of annotation.

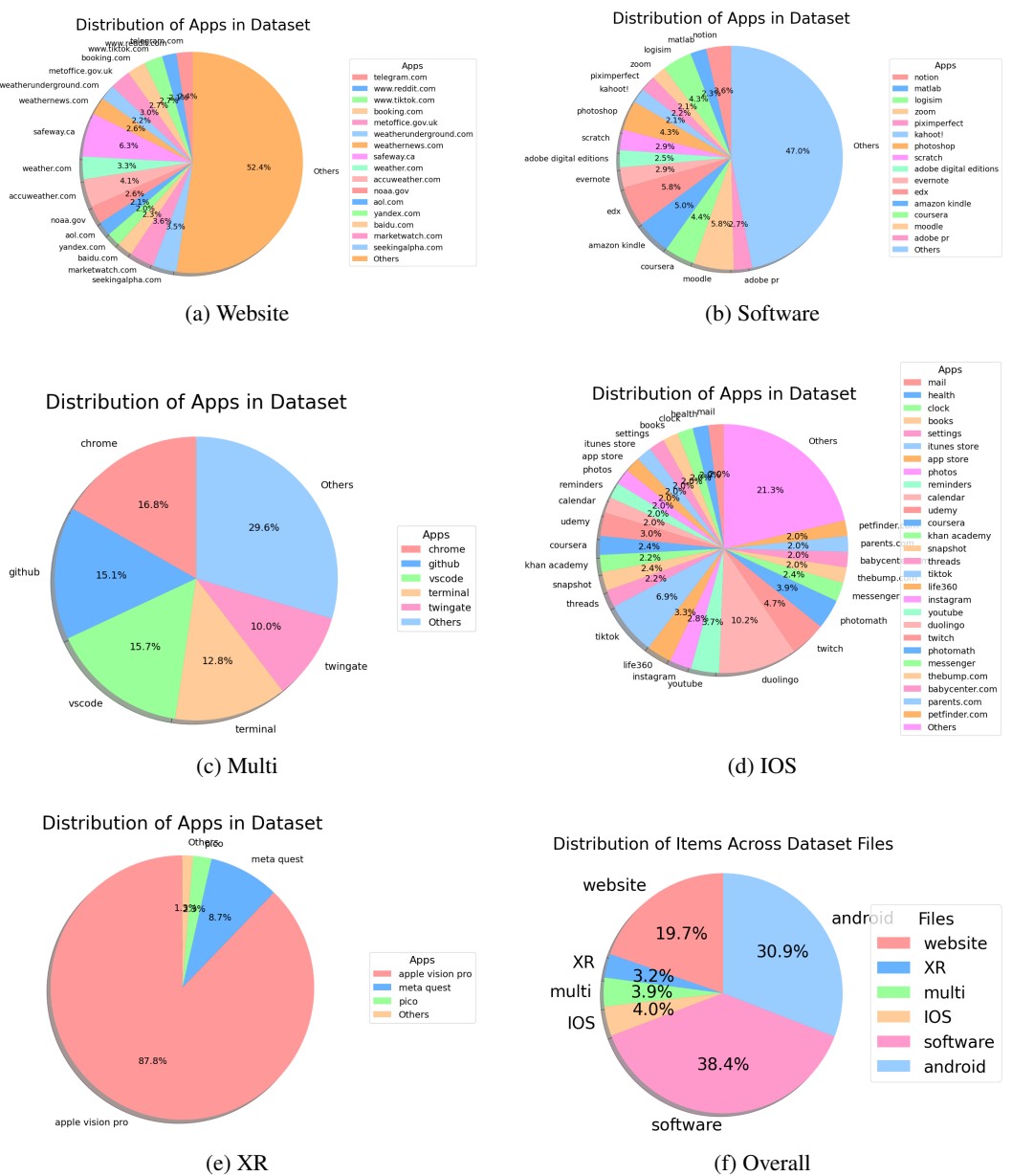

Figure 10: Detailed breakdown of each app, software, website in GUI-WORLD.

## A.2 SELECTED WEBSITE/SOFTWARE

In our study, we select a diverse range of websites and software to comprehensively evaluate GUI understanding capabilities across various user scenarios. These selections cover essential categories such as social media, productivity tools, online shopping, and educational platforms, providing a broad spectrum of GUI environments.

Figure 10 shows an overall distribution and our selected app, software, and website in GUI-WORLD.

These selections ensure that our study encompasses a wide array of user interactions and GUI complexities, thereby providing a robust evaluation of the current state-of-the-art methods in GUI understanding by MLLMs and comprehensively constructing a high-quality dataset.

Table 10: Comparison of raw answer from GPT-4V and golden answer from humans.

| Raw response from GPT-4V | Golden answer from humans |
|---|---|
| "Q": "What subcategories are visible under the 'Lifestyle' section in the static GUI content of the video?" "A": "In the 'Lifestyle' section of the telegram.com website, the visible subcategories are 'Worcester Living', 'Faith', and 'Food'. These are clearly displayed in a horizontal list just below the main 'Lifestyle' category header, indicating different areas of content that users can explore within the lifestyle section." | "Q": "What subcategories are visible under the 'Lifestyle' section in the static GUI content of the video?" "A": "The 'Lifestyle' section of the telegram.com website features visible subcategories such as 'Worcester Living', 'Faith', and 'Food'. These are prominently shown in a horizontal list beneath the main 'Lifestyle' header, offering users various content areas to explore within the section." |

### A.3 HUMAN KEYFRAMES ANNOTATION PROCESS

**Annotator's Information**    The annotation is conducted by 16 authors of this paper and 8 volunteers independently. As acknowledged, the diversity of annotators plays a crucial role in reducing bias and enhancing the reliability of the benchmark. These annotators have knowledge in the GUI domain, with different genders, ages, and educational backgrounds. The education backgrounds of annotators are above undergraduate. To ensure the annotators can proficiently mark the data, we provide them with detailed tutorials, teaching them how to use software to record videos or edit video clips. We also provide them with detailed criteria and task requirements in each annotation process.

**Recording Video.**    For self-recording videos, we employ OBS[3] on the Windows system for screen capturing and the official screen recording toolkit on the Mac/IOS system. This process necessitates human annotators to execute a series of targeted actions within specific websites or applications, which are subsequently captured as raw video footage. **We provide a list of software and website for each annotator to first get familiar with then record video that operating on them. For some popular software or website such as *chrome*, we ask several annotator to record video of it.** These actions, commonplace in everyday usage, enhance the reliability of our dataset. Subsequently, the raw videos are segmented into sub-videos, each encapsulating multiple actions (e.g., clicking a button) to achieve a specific objective (e.g., image search). The videos are then processed to extract keyframes annotated with detailed descriptions.

**Edition Based on YouTube Videos.**    For sourcing videos from YouTube, we utilize a search protocol formatted as "`[website name/application name] + tutorial`" to compile relevant video lists. Human annotators first review these videos to understand the primary operations they depict. These videos are then divided into sub-videos, each containing several actions directed towards a single goal (e.g., image search). Like the self-recorded footage, these segments are processed to isolate keyframes and furnish them with descriptive annotations.

**Keyframes Annotation.**    After obtaining the GUI video clips, human annotators will filter out the keyframes of the operations based on the video content and the mouse and keyboard actions at that time. They will also label the sub-operations or targets between the two keyframes. Once the annotation is complete, the annotators will provide an overall description of the entire video, summarizing the main goal of the human operations in the video. After all the information is annotated, we will use an LLM to refine the text content, reducing any errors made by human annotators and adjusting the sentence structure. The prompt we use for the LLM to polish the human annotations is shown in Figure 11 and Figure 12.

**Human-LLM Cooperated Instruction Generation.**    To curate and refine the golden answer of each video-instruction pair generated by GPT-4V, given that the raw response from GPT-4V may contain harmful content or hallucinations. The role of humans in the golden answer generation process is to enhance the difficulty of the questions and remove harmful and incorrect content, as shown in Table 10.

**Human Verifying GPT-4V Annotated Captions.**    We evaluate the quality of annotations from GPT-4V by selecting 1,000 detailed descriptions and captions generated by GPT-4V, which are

---

[3]https://obsproject.com/

then assessed by human annotators. The high satisfaction rate of 98% underscores the quality and relevance of the GPT-4V annotations.

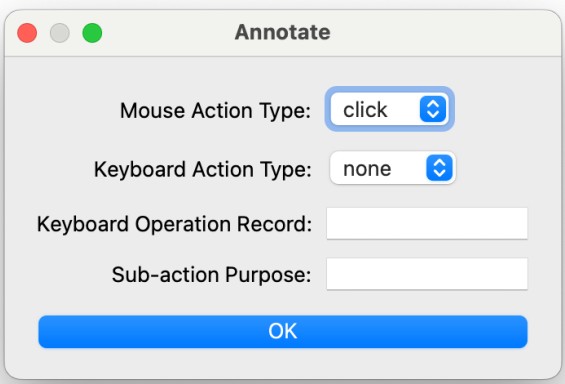

Figure 11: The overall preview of our annotating software.

Figure 12: The interface for annotating a keyframe, consists of mouse action, keyboard action, and a short sub-action purpose.

## B   DATASET ANALYSIS

In this section, we provide an analysis of the length distribution of QA in each GUI scenario, as illustrated in Figure 13 and Figure 14. Questions focused on sequential and predictive tasks are slightly longer than other types, while the golden answer of static tasks tends to be longer. Length of

Table 11: Examples of diverse question types in GUI-WORLD. (T. - Type)

| T. | Question | Examples |
|---|---|---|
| **Caption** | Detailed Description | **Q:** *Please provide a detailed description of what occurs throughout these sequential GUI images.*
**A:** The video shows a user taking the 16 Personalities test on a Windows desktop using the Edge browser... |
| | Summarized Caption | **Q:** *Write a clear description of the video, make sure the key features are well covered.*
**A:** Creating a new IT team in Todoist by selecting industry, job function, role, team size, and inviting members. |
| **Static** | Layout, Icon Retrieval | **Q:** *What related searches are suggested on the right side of the Bing results for 'emnlp 2024'?*
**A:** The suggested related searches shown include 'emnlp 2024 miami', 'eacl 2024 call for papers'... |
| | Textual Retrieval | **Q:** *What is the estimated time to complete the content for Week 2 of the course?*
**A:** The estimated time to complete the content for Week 2 of the course is 1 hour... |
| | Interrelations in GUI Content | **Q:** *What is the name of the browser and the tab where the user performs the product search?*
**A:** The browser is Microsoft Edge, and the user performs the product search in the eBay tab. |
| **Dynamic** | Content Retrieval | **Q:** *What specific action does the user take after turning their head to the left to view the left side of the page?*
**A:** After turning their head to the left to view the left side of the page, the user performs... |
| | Prediction | **Q:** *Given the mouse is over 'Add NeurIPS 2024 DB Track Submission,' what's the likely next step?*
**A:** It would be to click on the 'Add NeurIPS 2024 Datasets and Benchmarks Track Submission' button... |
| | Sequential Reasoning | **Q:** *Scrolls down from the 'Moon Gravity', which of the following cheats? A. Change Weather B. Skyfall . . .*
**A:** [[B]] |

Question-answer pair in various GUI scenarios is similarly distributed, with questions in Android environment being slightly shorter, and answers in XR environment being longer.

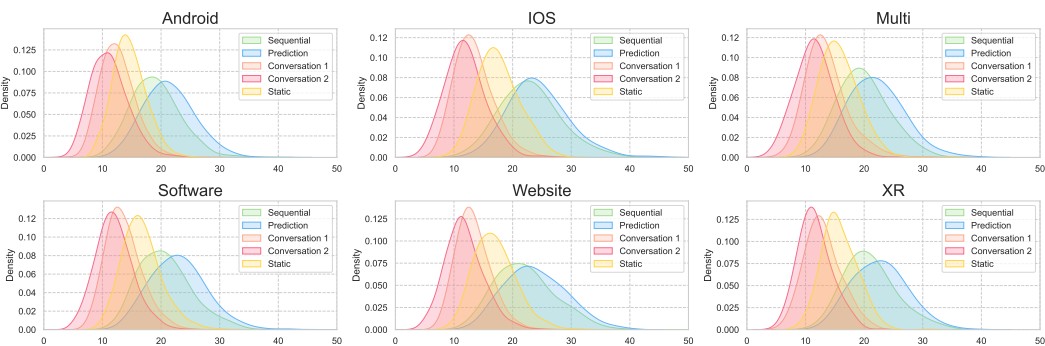

Figure 13: Length distribution of free-form questions.

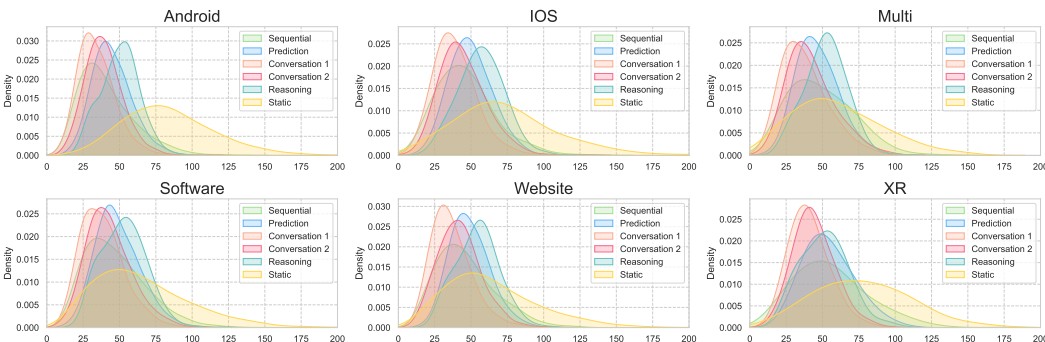

Figure 14: Length distribution of answers to free-form questions.

## C DETAILS OF EXPERIMENTS SETUPS

### C.1 FINE-TUNE DATASET CONSTRUCTION

We use two settings to fine-tune GUI-VID, one with video-text pairs only, and the other with video-text and image-text pairs, which are all GUI content:

- **Video Only.** In this setting, we only train GUI-VID with video-text pairs in GUI-WORLD, as shown in Table 12.
- **Video-Image.** Inspired by the pre-trained process of Videochat2, we include image-text pairs to help the visual encoder align GUI knowledge. These images are selected from our GUI-WORLD,

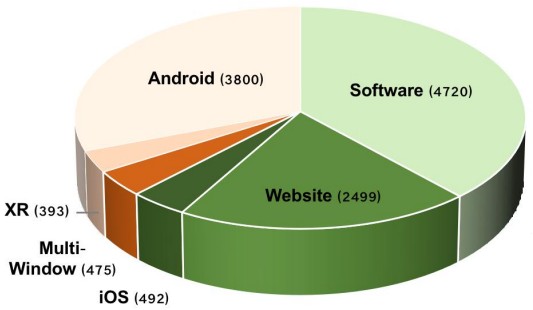

Figure 15: Statistic of different GUI scenarios in GUI-WORLD.

MetaGUI (Sun et al., 2022), and OmniAct (Kapoor et al., 2024) for high-quality GUI content. Subsequently, we use GPT-4V to generate a detailed description and a concise caption for each image. Finally, we construct a dataset consisting of video-text and image-text pairs for gaining comprehensive GUI-oriented capabilities.

Table 12: Video-only fine-tune dataset.

| Stage | Data types | Amount |
|---|---|---|
| 1 | Detailed Description | 14,276 |
| | Concise Caption | 7,138 |
| 2 | GUI VQA | 21,414 |
| | Multiple-Choice QA | 14,276 |
| | Conversation | 7,138 |

Table 13: Video-image fine-tune dataset.

| Stage | Data types | Source | Type | Amount |
|---|---|---|---|---|
| 1 | GUI-WORLD | Video | Detailed Description | 14,276 |
| | | | Concise Caption | 7,138 |
| | | Image | Detailed Description | 5,555 |
| | | | Concise Caption | 5,555 |
| | METAGUI | Image | Detailed Description | 19,626 |
| | | | Concise Caption | 19,626 |
| | OmniAct | | Detailed Description | 260 |
| | | | Concise Caption | 260 |
| 2 | GUI-WORLD | Video | GUI VQA | 21,414 |
| | | | Multiple-Choice QA | 14,276 |
| | | | Conversation | 7,138 |

## C.2 HYPERPARAMETER SETTINGS

In this section, we will introduce the hyperparameters of MLLMs to facilitate experiment reproducibility and transparency. We divide them into three parts: the inference phase during benchmark and dataset construction, the LLM-as-a-Judge phase, and the fine-tuning phase. All our experiments were conducted on a server equipped with dual A800 and dual 4090 GPUs.

**Inference.** We empirically study 7 MLLMs, involving 4 Image LLMs and 3 Video LLMs, with their hyperparameters detailed as follows:

- **GPT-4V (OpenAI, 2023) & GPT-4o (OpenAI, 2024):** We set the temperature and top-p as 0.9, max-token as 2048, and both all images input are set as high quality in *Instruction Dataset Construction* and benchmarking.
- **Gemini-Pro-1.5 (Team et al., 2023):** We use the default settings, which set temperature as 0.4, top-p as 1, and max-token as 2048. It should be noted that during our project, Gemini-Pro-1.5 is

still under the user request limit, which only provides 100 requests per day, making our benchmark difficult. Given that Gemini hasn't launched Pay-as-you-go[4], we will include benchmark results on 'Human' setting as soon as possible.

- **Qwen-VL-Max (Bai et al., 2023):** We use the default settings for Qwen-VL-Max, with top-p as 0.8 and max-token as 2048. Given that the input context window is merely 6,000 for Qwen, we scale the resolution for all images to 0.3.
- **ChatUnivi (Jin et al., 2023):** We use ChatUnivi-7B built upon Vicuna-v0-7B and set the max frame as 100, temperature as 0.2, and max-token as 1024.
- **Minigpt4video (Ataallah et al., 2024):** We use the suggested settings[5] for this model and the max-frame are set as 45, with only the max-token being modified to 1024.
- **VideoChat2 & GUI-VID (Li et al., 2023c):** For a fair comparison, we set the same hyperparameters for VideoChat2 & GUI-VID. We set the max-token as 1024, top-$p$ as 0.9, temperature as 1.0, max-frame as 8/16, repetition penalty as 1.2, and length penalty as 1.2.

**LLM-as-a-Judge.** We investigate four LLM-as-a-Judge in giving a similarity score for the MLLM's response and ground truth, namely GPT-4 (Achiam et al., 2023), ChatGPT (OpenAI, 2023), LLaMA-3-70b-instruct (Meta, 2023b), and Mixtral-8x22b-instruct-v0.1 (ai, 2024). Hyperparameter settings are detailed as follows:

- **GPT-4 & ChatGPT.** We set the temperature as 0.6 and others as default.
- **LLaMA-3-70b-instruct.** We set the temperature as 0.6, top-p as 0.9, top-k as 50.
- **Mixtral-8x22b-instruct-v0.1.** We set top-p as 0.7, top-k as 50, and temperature as 0.7.

**Fine-tune.** We include several hyperparameter settings in experiment settings and ablation studies, as shown in Table 14.

Table 14: Configuration settings for fine-tuning.

| Config | Setting |
|---|---|
| input frame | 8 |
| input resolution | 224 |
| max text length | 512 |
| input modal | I. + V. |
| optimizer | AdamW |
| optimizer momentum | $\beta_1, \beta_2 = 0.9, 0.999$ |
| weight decay | 0.02 |
| learning rate schedule | cosine decay |
| learning rate | 2e-5 |
| batch size | 4 |
| warmup epochs | 0.6 |
| total epochs | 3 |
| backbone drop path | 0 |
| QFormer drop path | 0.1 |
| QFormer dropout | 0.1 |
| QFormer token | 96 |
| flip augmentation | yes |
| augmentation | MultiScaleCrop [0.5, 1] |

Table 15: Evaluating LLM-as-a-Judge as a replacement for human judging in the scoring setting.

| Models | Pearson(↑) | Spearman(↑) | Kendall(↑) | $ per Benchmark(↓) |
|---|---|---|---|---|
| GPT-4 | **0.856** | **0.853** | **0.793** | 120$ |
| ChatGPT | 0.706 | 0.714 | 0.627 | **12$** |
| Llama-3-70b-instruct | 0.774 | 0.772 | 0.684 | **12$** |
| Mixtral-8x22b-instruct-v0.1 | 0.759 | 0.760 | 0.670 | 15$ |

---

[4] https://ai.google.dev/pricing
[5] https://github.com/Vision-CAIR/MiniGPT4-video

Table 16: Performance of GUI-VID when removing data from MetaGUI and OmniAct.

| Models | Software | | Website | | XR | | Multi | | IOS | | Android | | Avg. | |
|---|---|---|---|---|---|---|---|---|---|---|---|---|---|---|
| | MC | Free | MC | Free | MC | Free | MC | Free | MC | Free | MC | Free | MC | Free |
| VideoChat2 | 45.50% | 2.144 | 42.60% | 2.221 | 44.00% | 2.005 | 40.40% | 2.222 | 40.20% | 2.169 | 44.70% | 2.119 | 42.90% | 2.147 |
| GUI-VID | **59.90%** | **2.847** | 54.10% | **2.957** | 55.60% | 2.764 | **52.90%** | **2.861** | 51.80% | **2.773** | 53.40% | **2.572** | 54.60% | **2.796** |
| GUI-VID (w.o. other data) | 59.70% | 2.811 | **57.30%** | 2.844 | **59.62%** | **2.853** | 50.00% | 2.799 | **55.31%** | 2.769 | **57.67%** | 2.547 | **56.60%** | 2.767 |

Table 17: GUI-VID wins more user selection in GUI-related chat assistant experiment.

| Scenarios | GUI-VID | Tie | VideoChat2 |
|---|---|---|---|
| Software | **82.7%** | 13.3% | 4.0% |
| Website | **86.0%** | 12.0% | 2.0% |
| XR | **88.0%** | 8.7% | 3.3% |
| Multi | **85.3%** | 10.0% | 8.7% |
| IOS | **92.0%** | 6.0% | 2.0% |
| Android | **82.0%** | 16.0% | 2.0% |
| Average | **86.0%** | 11.0% | 3.7% |

## C.3 EVALUATION.

Given the complexity of free-form answers in GUI scenarios, the evaluation includes specific positions of GUI elements, textual content, and comparing the response to the golden answer. LLM-as-a-Judge has been widely used in previous studies for complex evaluation tasks (Zheng et al., 2023; Liu et al., 2023d). Therefore, we leverage LLM-as-a-Judge (Zheng et al., 2023) in a similar setting to MM-vet (Yu et al.), which compares the MLLM's response to the golden answer. We carefully evaluate the accessibility of leveraging LLM-as-a-Judge, selecting 1,000 samples covering 6 free-form questions mentioned in our dataset. As shown in Table 15, GPT-4 outperforms other LLMs, exhibiting a better human alignment on providing a similarity score for the response compared to the golden answer, although it is approximately 10 times more expensive than other models.

## D ADDITIONAL EXPERIMENTS RESULTS

**Removing Data from MetaGUI and OmniAct.** As shown in Table 16, after excluding OmniAct and MetaGUI from the training data, the model showed performance decreases in certain areas while demonstrating notable improvements in others. Specifically, we observed significant enhancements in MCQA tasks and XR-related capabilities. Conversely, models trained with MetaGUI and OmniAct exhibited superior performance on general free-form questions. These experimental results validate that models trained exclusively on GUI-WORLD can achieve highly competitive performance, thereby confirming the exceptional quality of our dataset.

In this section, we first provide an ablation study on keyframe selection methods. Then, we conduct statistics and human preference experiments on correlations of GUI understanding capability to other mainstream GUI-related tasks. Furthermore, we provide detailed, performance on newly released models after our submission of the first version, followed by very detailed results on each task in each GUI scenario.

**Ablation Study on Keyframe Identify Methods.** Firstly, we show performance on model-based keyframe identify methods in Table 6, with details of UVD+VIP and UVD+R3M in Tables 24 and 25.

**Correlation between GUI Understanding and Other Mainstream GUI Tasks.** Furthermore, we conduct additional analysis and experiments to show how GUI understanding capability helps mainstream GUI-related tasks, including generating code to operate GUI (Cheng et al., 2024) and assist people through chat (Hong et al., 2024). Both demonstrate the strong correlation between GUI understanding capability and specific tasks for GUI agents.

- We compare the benchmark results on GUI-WORLD with existing benchmarks (Xie et al., 2024; Qinghong Lin et al., 2024; Liu et al., 2024c) for operating on GUI as shown in Table 18, and find

Table 18: Strong Correlation Between Our Benchmark (GUI Understanding) and Other GUI Agent Benchmarks.

| Model | GUI-WORLD | VisualAgentBench | VideoGUI | OS-World |
|---|---|---|---|---|
| GPT-4o | **1** | **1** | **1** | 2 |
| GPT-4V | **2** | **2** | **2** | 1 |
| Gemini-1.5-Pro | **3** | **3** | **3** | 3 |
| Qwen-VL-Max | **4** | **4** | **4** | / |

Table 19: The Performance of Video-LLaVA.

| Scenario | MCQA | Description | Conversation | Dynamic | Static | Caption | Average |
|---|---|---|---|---|---|---|---|
| XR | 0.442 | 1.100 | 2.686 | 2.055 | 1.808 | 1.654 | 2.258 |
| Android | 0.513 | 1.162 | 2.952 | 1.858 | 1.673 | 1.763 | 2.259 |
| IOS | 0.497 | 1.143 | 2.966 | 1.992 | 1.680 | 1.654 | 2.319 |
| Multi | 0.459 | 1.106 | 2.863 | 2.069 | 1.781 | 1.772 | 2.329 |
| Website | 0.524 | 1.183 | 3.059 | 2.102 | 1.736 | 1.371 | 2.410 |
| Software | 0.529 | 1.241 | 2.942 | 1.958 | 1.657 | 1.519 | 2.290 |
| Average | 0.494 | 1.156 | 2.911 | 2.005 | 1.722 | 1.622 | 2.311 |

that the results generally match, i.e., the stronger the understanding ability, the stronger the agent performance.

- For the definition of chat helping humans, we select 180 videos from the benchmark, choosing 30 videos for each scenario. We ask 5 human annotators to pose the question they most wanted to ask after watching each video. We then use GUI-VID, both before and after fine-tuning, to answer these questions. The human annotators who ask the questions are then asked to indicate which answer is more helpful. The results are shown in Table 17, demonstrating that models trained in GUI understanding are more favored by people when acting as GUI agents.

**Performance of Newly Released Models in GUI-WORLD Test Set.** We evaluate two latest models, LLaVA-Next-Video-7B-DPO (Liu et al., 2024a) and Video-LLaVA (Lin et al., 2023a). We show their performance in Tables 20 and 19. Our model outperforms these in most tasks, except conversation, likely due to their use of DPO during training.

**Performance on MV-Bench.** We conducted evaluations on MVBench (Li et al., 2023c) across four models, as shown in Table 21. While our model showed some performance degradation compared to the original model, it still significantly outperformed the other two baseline models. We would like to note that the primary purpose of training GUI-VID was to demonstrate the effectiveness of our dataset, rather than achieving state-of-the-art performance. Therefore, we did not employ the strongest baseline model or optimal training methodologies, which may have contributed to some catastrophic forgetting. However, given that GUI-VID is specifically designed for GUI-related tasks, we believe this performance trade-off is acceptable within our research context.

**Performance on Mind2Web-Multimodal.** We evaluated our model on the Mind2Web-Multimodal dataset (Deng et al., 2024) to demonstrate the effectiveness of fine-tuning on GUI-WORLD. Mind2Web-Multimodal is a multiple-choice GUI benchmark that assesses GUI operation capabilities, where each sample comprises an image, a task description, and response options. Our evaluation included four experimental settings: zero-shot inference using both VideoChat2 and GUI-VID, as well as versions of these models fine-tuned on Mind2Web-Multimodal. We conducted the fine-tuning process for 3 epochs. As shown in Tables 22 and 23, fine-tuning on GUI-WORLD enhances the models' GUI understanding capabilities and improves their performance on operational benchmarks.

**Additional Experiment Results on GUI-WORLD.** For captioning tasks, Table 26 shows comprehensive experimental results among six scenarios. For scores of LLM-as-a-Judge in a specific task, see Tables 27, 28, 29, 30, and 31. For performance in fine-grain (application level), see Figure 16 for Gemini-Pro and Figure 17 for Qwen-VL-Max.

Table 20: LLaVA-Next-Video-7B-DPO Performance

| Scenarios | MCQA | Description | Conversation | Dynamic | Static | Caption | Average |
|---|---|---|---|---|---|---|---|
| XR | 0.596 | 1.867 | 3.123 | 2.580 | 2.147 | 1.987 | 2.709 |
| Android | 0.243 | 1.675 | 3.338 | 2.360 | 1.980 | 2.189 | 2.675 |
| IOS | 0.581 | 1.762 | 3.229 | 2.536 | 2.051 | 2.017 | 2.717 |
| Multi | 0.355 | 1.069 | 2.982 | 2.437 | 1.870 | 2.541 | 2.541 |
| Website | 0.484 | 1.729 | 3.123 | 2.422 | 1.854 | 2.004 | 2.588 |
| Software | 0.569 | 1.762 | 3.220 | 2.448 | 1.868 | 2.149 | 2.641 |
| Average | 0.471 | 1.644 | 3.169 | 2.464 | 1.961 | 2.148 | 2.645 |

Table 21: GUI-VID outperforms two mainstream Video LLMs, while slightly lagging behind VideoChat2 on MVBench.

| Category | VideoChat2 | GUI-VID | VideoLLaMA | VideoChatGPT |
|---|---|---|---|---|
| Action Prediction | 47.5 | 39.0 | 25.5 | 26.0 |
| Unexpected Action | 60.0 | 57.0 | 39.0 | 26.5 |
| Object Existence | 58.0 | 54.0 | 48.0 | 54.0 |
| Object Interaction | 71.5 | 51.0 | 40.5 | 28.0 |
| Object Shuffle | 41.0 | 30.5 | 38.0 | 40.0 |
| Moving Direction | 23.0 | 21.0 | 22.5 | 23.0 |
| Action Localization | 23.0 | 30.5 | 22.5 | 20.0 |
| Scene Transition | 88.0 | 65.5 | 43.0 | 31.0 |
| Action Count | 39.5 | 34.5 | 34.0 | 30.5 |
| Moving Count | 42.0 | 29.0 | 22.5 | 25.5 |
| Moving Attribute | 58.5 | 53.5 | 32.5 | 39.5 |
| State Change | 44.5 | 41.0 | 45.5 | 48.5 |
| Character Order | 36.5 | 39.0 | 32.5 | 29.0 |
| Egocentric Navigation | 35.0 | 34.0 | 40.0 | 33.0 |
| Episodic Reasoning | 38.5 | 37.0 | 30.0 | 29.5 |
| Average | 47.1 | 41.1 | 34.4 | 32.3 |

# E  PROMPTS

In this section, we provide detailed prompts for models and human annotators. Figure 19 shows the guideline of human annotation, and Figure 18 shows the prompt for leveraging LLMs to refine grammarly mistakes and polish sentence for human annotations. Figures 20, 21, and 22 present the prompt for Human-MLLM collaboration method to generate GUI-orientaed tasks. Figure 23 illustrate the prompt for benchmarking MLLMs, different GUI scenarios and different QA type has different prompt. Figure 24 presents the prompt for free-form QA using LLM-as-a-Judge. Figure 25 presents the prompt for multiple-choice QA.

Table 22: Performance of 4 settings in test domain subset of Mind2web-Multimodal.

| Category | ZeroShot | | Fine-tuned | |
|---|---|---|---|---|
| | VideoChat2 | GUI-VID | VideoChat2 | GUI-VID |
| Cooking | 21.28% | 23.40% | 23.40% | **26.24%** |
| Education | 17.61% | 16.61% | 24.92% | **28.24%** |
| Finance | 26.02% | 29.59% | 26.02% | **32.65%** |
| Government | 18.80% | 18.28% | 26.89% | **27.42%** |
| Health | 17.08% | 17.37% | 22.58% | **26.48%** |
| Home service | 19.08% | 20.14% | 21.55% | **28.27%** |
| Housing | 12.21% | 14.13% | 18.20% | **21.84%** |
| Job | 16.25% | 18.00% | 20.00% | **20.25%** |
| Moving | 14.36% | 17.44% | 20.00% | **24.62%** |
| Pet | 14.05% | 17.97% | **21.57%** | 21.24% |
| Shipping | 13.42% | 12.99% | **20.78%** | 20.34% |
| Social media | 22.15% | 24.37% | 22.78% | **29.43%** |
| Weather | 27.14% | 22.14% | **26.43%** | 20.00% |
| **Overall** | 17.53% | 18.59% | 22.37% | **25.14%** |

Table 23: Performance of 4 settings in test website subset of Mind2web-Multimodal.

| Category | ZeroShot | | Fine-tuned | |
|---|---|---|---|---|
| | VideoChat2 | GUI-VID | VideoChat2 | GUI-VID |
| Auto | 20.00% | 16.00% | **19.00%** | 18.00% |
| Department | 9.52% | 11.90% | 14.29% | **16.67%** |
| Digital | 19.73% | 19.73% | 15.65% | **20.41%** |
| Event | **34.65%** | 25.74% | 21.78% | 22.77% |
| General | 18.32% | 18.85% | 17.80% | **20.94%** |
| Music | 16.67% | 15.66% | 21.69% | **24.10%** |
| Other | 17.81% | 21.92% | 27.40% | **35.62%** |
| Restaurant | 10.70% | 14.44% | **17.65%** | 16.58% |
| Sports | 16.98% | 18.87% | 20.75% | **26.42%** |
| **Overall** | 17.94% | 17.96% | 18.84% | **21.20%** |

Table 24: Detailed Performance of GPT-4o using UVD+ViP Keyframe Identification Method.

| Scenario | MCQA | Description | Conversation | Dynamic | Static | Caption | Average |
|---|---|---|---|---|---|---|---|
| Software | **86.2%** | 3.297 | **4.282** | 3.354 | 3.478 | **4.112** | **3.749** |
| Website | 82.0% | 3.248 | 4.155 | 3.415 | **3.567** | 4.074 | 3.744 |
| XR | 84.2% | 2.980 | 3.775 | 3.034 | 3.122 | 3.587 | 3.347 |
| Multi | 82.1% | **3.391** | 4.165 | **3.466** | 3.404 | 3.868 | 3.659 |
| IOS | 86.0% | 3.157 | 4.017 | 3.353 | 3.492 | 4.050 | 3.648 |
| Mobile | 80.7% | 2.827 | 3.871 | 2.970 | 3.014 | 3.844 | 3.340 |
| Average | 83.5% | 3.150 | 4.044 | 3.265 | 3.346 | 3.923 | 3.581 |

Table 25: Detailed Performance of GPT-4o using UVD+R3M Keyframe Identification Method.

| Scenario | MCQA | Description | Conversation | Dynamic | Static | Caption | Average |
|---|---|---|---|---|---|---|---|
| Software | 85.8% | **3.290** | **4.273** | 3.352 | 3.458 | **4.134** | 3.741 |
| Website | 82.7% | 3.282 | 4.114 | 3.460 | **3.591** | 4.065 | 3.746 |
| XR | **87.7%** | 3.010 | 3.861 | 3.142 | 3.161 | 3.600 | 3.433 |
| Multi | 83.6% | 3.237 | 4.129 | **3.503** | 3.417 | 3.897 | 3.737 |
| IOS | 86.4% | 3.165 | 4.094 | 3.328 | 3.480 | 4.078 | 3.663 |
| Android | 80.6% | 2.835 | 3.876 | 2.968 | 3.072 | 3.865 | 3.353 |
| Average | 84.5% | 3.136 | 4.058 | 3.292 | 3.363 | 3.940 | **3.612** |

Table 26: Scores of Caption (Cap.) and Description (Des.) tasks in six GUI scenarios.

| Models | Setting | Software | | Website | | XR | | Multi | | IOS | | Android | | Avg. | |
|---|---|---|---|---|---|---|---|---|---|---|---|---|---|---|---|
| | | Cap. | Des. | Cap. | Des. | Cap. | Des. | Cap. | Des. | Cap. | Des. | Cap. | Des. | Cap. | Des. |
| Gemini-Pro-1.5 | R. | 3.659 | 2.837 | 3.613 | 2.860 | 2.995 | 2.590 | 3.276 | 2.470 | 3.678 | 2.936 | - | - | 3.444 | 2.739 |
| | E. | 3.350 | 2.468 | 3.159 | 2.422 | 2.837 | 2.279 | 2.824 | 2.109 | 3.394 | 2.519 | 3.185 | 2.312 | 3.125 | 2.351 |
| Qwen-VL-Max | R. | 2.381 | 1.758 | 2.326 | 1.681 | 2.172 | 1.772 | 2.035 | 1.463 | 2.513 | 1.662 | 2.141 | 1.565 | 2.261 | 1.650 |
| | E. | 2.459 | 1.693 | 2.317 | 1.599 | 2.167 | 1.638 | 2.190 | 1.438 | 2.189 | 1.615 | 2.002 | 1.429 | 2.221 | 1.569 |
| | H. | 2.474 | 1.711 | 2.457 | 1.698 | 2.383 | 1.777 | 1.910 | 1.346 | 2.577 | 1.795 | 2.474 | 1.711 | 2.360 | 1.665 |
| GPT-4V | R. | 3.579 | 2.676 | 3.612 | 2.699 | 2.975 | 2.525 | 3.281 | 2.661 | 3.757 | 2.775 | 3.655 | 2.755 | 3.479 | 2.682 |
| | E. | 3.141 | 2.301 | 3.293 | 2.380 | 2.471 | 2.085 | 3.063 | 2.324 | 3.624 | 2.611 | 3.201 | 2.312 | 3.132 | 2.335 |
| | H. | 3.352 | 2.509 | 3.702 | 2.750 | 3.050 | **3.556** | 3.524 | 2.673 | 3.670 | 2.588 | - | - | 3.460 | 2.614 |
| GPT-4o | H. | **4.048** | 3.028 | 4.067 | 3.233 | 3.398 | 2.729 | **3.869** | 3.111 | 4.014 | 2.993 | 4.071 | 3.095 | **3.911** | **3.869** |
| ChatUnivi | - | 1.587 | 1.240 | 1.569 | 1.254 | 1.417 | 1.148 | 1.575 | 1.267 | 1.480 | 1.146 | 1.778 | 1.249 | 1.568 | 1.217 |
| Minigpt4Video | - | 1.246 | 1.073 | 1.200 | 1.057 | 1.320 | 1.106 | 1.130 | 1.034 | 1.190 | 1.076 | 1.184 | 1.061 | 1.212 | 1.068 |
| VideoChat2 | - | 1.992 | 1.312 | 1.817 | 1.307 | 1.838 | 1.426 | 2.222 | 1.433 | 2.169 | 1.270 | 2.119 | 1.294 | 1.900 | 1.340 |
| GUI-VID | - | 3.562 | 2.085 | 3.655 | 2.167 | **3.747** | 2.153 | 3.370 | 1.742 | 3.566 | 2.071 | 2.662 | 1.248 | 3.427 | 1.911 |

Table 27: Detailed scores for each tasks in **Website** scenarios.

| Models | Setting | Static | Sequential | Prediction | Conversation1 | Conversation2 | Average |
|---|---|---|---|---|---|---|---|
| Gemini-Pro-1.5 | R. | 3.279 | 3.050 | 3.560 | 3.579 | 3.796 | 3.452 |
| | E. | 2.983 | 2.491 | 3.432 | 3.405 | 3.760 | 3.215 |
| Qwen-VL-Max | R. | 2.317 | 2.271 | 2.802 | 2.995 | 3.069 | 2.656 |
| | E. | 2.256 | 2.198 | 2.821 | 2.861 | 3.144 | 2.627 |
| | H. | 2.308 | 2.078 | 2.832 | 3.061 | 3.358 | 2.698 |
| GPT-4V | R. | 3.461 | 3.214 | 3.754 | 3.778 | 4.029 | 3.648 |
| | E. | 3.197 | 2.808 | 3.487 | 3.717 | 3.954 | 3.433 |
| | H. | **3.498** | 3.255 | 3.727 | 3.731 | 4.061 | 3.655 |
| | C.C. | 1.746 | 2.738 | 3.645 | 3.363 | 3.632 | 3.025 |
| | D.C. | 2.704 | 2.917 | 3.686 | 3.680 | 3.901 | 3.380 |
| | H.+D.C. | 3.313 | 3.221 | **3.852** | 3.850 | **4.171** | 3.682 |
| GPT-4o | H. | 3.443 | **3.373** | 3.672 | **4.086** | 4.122 | **3.740** |
| ChatUnivi | - | 1.701 | 1.668 | 2.524 | 2.514 | 3.338 | 2.349 |
| Minigpt4Video | - | 1.309 | 1.233 | 1.766 | 1.439 | 1.854 | 1.520 |
| VideoChat2 | - | 1.771 | 1.777 | 2.288 | 2.461 | 2.812 | 2.221 |
| GUI-VID | - | 2.406 | 2.341 | 3.544 | 3.135 | 3.355 | 2.957 |

Table 28: Detailed scores for each tasks in **XR** scenarios.

| Models | Setting | Static | Sequential | Prediction | Conversation1 | Conversation2 | Average |
|---|---|---|---|---|---|---|---|
| Gemini-Pro-1.5 | R. | 2.892 | 2.505 | 3.543 | 3.222 | 3.611 | 3.154 |
| | E. | 2.814 | 2.163 | 3.510 | 3.108 | 3.455 | 3.006 |
| Qwen-VL-Max | R. | 2.047 | 1.968 | 2.712 | 2.879 | 3.132 | 2.469 |
| | E. | 2.125 | 1.973 | 2.658 | 2.760 | 3.029 | 2.499 |
| | H. | 1.886 | 1.920 | 2.656 | 2.727 | 3.012 | 2.373 |
| GPT-4V | R. | **2.934** | 2.668 | 3.392 | 3.291 | 3.714 | 3.200 |
| | E. | 2.222 | 2.153 | 3.310 | 3.151 | 3.618 | 2.892 |
| | H. | 2.893 | 2.778 | 3.538 | **3.364** | 3.747 | **3.265** |
| | C.C. | 1.744 | 2.412 | 3.327 | 3.080 | 3.485 | 2.809 |
| | D.C. | 2.427 | 2.409 | 3.518 | 3.176 | **3.749** | 3.056 |
| | H.+D.C. | 2.775 | 2.635 | **3.580** | 3.235 | 3.734 | 3.191 |
| GPT-4o | H. | 2.871 | **2.745** | 3.370 | 3.596 | 3.836 | 3.285 |
| ChatUnivi | - | 1.660 | 1.420 | 2.205 | 2.250 | 3.270 | 2.161 |
| Minigpt4Video | - | 1.225 | 1.161 | 1.610 | 1.347 | 1.465 | 1.362 |
| VideoChat2 | - | 1.654 | 1.547 | 2.192 | 2.099 | 2.529 | 2.005 |
| GUI-VID | - | 2.444 | 2.147 | 3.347 | 2.836 | 3.036 | 2.764 |

Table 29: Detailed scores for each tasks in **Multi-windows** scenarios.

| Models | Setting | Static | Sequential | Prediction | Conversation1 | Conversation2 | Average |
|---|---|---|---|---|---|---|---|
| Gemini-Pro-1.5 | R. | 2.538 | 2.410 | 3.296 | 3.152 | 3.402 | 2.959 |
| | E. | 2.545 | 2.049 | 2.972 | 2.930 | 3.389 | 2.777 |
| Qwen-VL-Max | R. | 1.793 | 1.872 | 2.770 | 2.897 | 3.122 | 2.432 |
| | E. | 1.866 | 1.780 | 2.730 | 2.627 | 3.105 | 2.362 |
| | H. | 1.884 | 1.969 | 2.913 | 2.689 | 3.104 | 2.490 |
| GPT-4V | R. | **3.185** | 2.655 | 3.745 | 3.699 | 3.973 | 3.452 |
| | E. | 2.902 | 2.406 | 3.636 | 3.420 | 3.729 | 3.219 |
| | H. | 3.000 | 2.952 | 3.801 | 3.597 | 3.889 | 3.449 |
| | C.C. | 2.097 | 2.973 | 3.774 | 3.331 | 3.621 | 3.160 |
| | D.C. | 2.671 | 2.979 | 3.849 | 3.466 | 3.822 | 3.358 |
| | H.+D.C. | 3.037 | **3.162** | **4.079** | 3.748 | 4.036 | 3.617 |
| GPT-4o | H. | 3.108 | 3.106 | 3.829 | **4.043** | **4.188** | **3.654** |
| ChatUnivi | - | 1.658 | 1.623 | 2.514 | 2.384 | 3.199 | 2.275 |
| Minigpt4Video | - | 1.205 | 1.186 | 1.690 | 1.400 | 1.801 | 1.457 |
| VideoChat2 | - | 1.754 | 1.774 | 2.479 | 2.420 | 2.699 | 2.222 |
| GUI-VID | - | 2.485 | 2.067 | 3.537 | 2.954 | 3.247 | 2.861 |

Table 30: Detailed scores for each tasks in **IOS** scenarios.

| Models | Setting | Static | Sequential | Prediction | Conversation1 | Conversation2 | Average |
|---|---|---|---|---|---|---|---|
| Gemini-Pro-1.5 | R. | 3.076 | 2.637 | 3.370 | 3.366 | 3.615 | 3.213 |
| | E. | 2.852 | 2.356 | 3.137 | 3.126 | 3.566 | 3.007 |
| Qwen-VL-Max | R. | 2.438 | 2.244 | 2.923 | 3.102 | 3.273 | 2.779 |
| | E. | 2.303 | 2.150 | 2.614 | 3.145 | 3.264 | 2.659 |
| | H. | 1.884 | 1.969 | 2.913 | 2.689 | 3.104 | 2.490 |
| GPT-4V | R. | **3.364** | **3.080** | **3.684** | 3.766 | **4.184** | **3.614** |
| | E. | 3.209 | 2.774 | 3.545 | 3.611 | 4.006 | 3.427 |
| | H. | 3.107 | 2.830 | 3.631 | 3.680 | 4.011 | 3.453 |
| | C.C. | 1.788 | 2.291 | 3.511 | 3.212 | 3.542 | 2.868 |
| | D.C. | 2.751 | 2.732 | 3.654 | 3.642 | 3.842 | 3.324 |
| | H.+D.C. | 3.090 | 2.965 | 3.740 | 3.786 | 3.994 | 3.516 |
| GPT-4o | H. | 3.183 | 2.993 | 3.460 | **4.050** | 4.141 | 3.558 |
| ChatUnivi | - | 1.771 | 1.642 | 2.408 | 2.559 | 3.307 | 2.337 |
| Minigpt4Video | - | 1.291 | 1.219 | 1.698 | 1.556 | 1.737 | 1.501 |
| VideoChat2 | - | 1.955 | 1.803 | 2.145 | 2.315 | 2.626 | 2.169 |
| GUI-VID | - | 2.262 | 2.133 | 3.401 | 2.843 | 3.224 | 2.773 |

Table 31: Detailed scores for each tasks in **Android** scenarios.

| Models | Setting | Static | Sequential | Prediction | Conversation1 | Conversation2 | Average |
|---|---|---|---|---|---|---|---|
| Gemini-Pro-1.5 | E. | 2.703 | 2.460 | 3.157 | 3.642 | 3.881 | 3.168 |
| Qwen-VL-Max | R. | 1.887 | 1.804 | 2.398 | 2.823 | 3.056 | 2.309 |
| | E. | 1.785 | 1.630 | 2.311 | 2.605 | 3.233 | 2.277 |
| GPT-4V | R. | **3.116** | 3.047 | **3.477** | 3.924 | 4.008 | 3.515 |
| | E. | 2.705 | 2.470 | 3.175 | 3.647 | 3.885 | 3.176 |
| | C.C. | 2.092 | 2.243 | 3.139 | 3.443 | 3.782 | 2.939 |
| | D.C. | 3.015 | 2.890 | 3.357 | 3.883 | 3.990 | 3.427 |
| GPT-4o | H. | 3.057 | **3.220** | 3.373 | **3.981** | **4.186** | **3.561** |
| ChatUnivi | - | 1.835 | 1.654 | 2.317 | 2.712 | 3.433 | 2.390 |
| Minigpt4Video | - | 1.183 | 1.159 | 1.507 | 1.342 | 1.521 | 1.342 |
| VideoChat2 | - | 1.732 | 1.754 | 2.125 | 2.340 | 2.645 | 2.119 |
| GUI-VID | - | 2.010 | 1.928 | 3.053 | 2.755 | 3.105 | 2.572 |

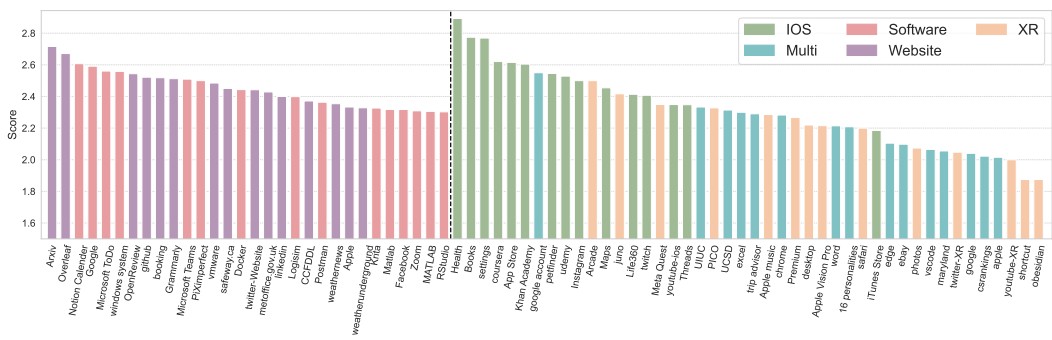

Figure 16: Fine-grained performance of Gemini-Pro-1.5 in each software and website.

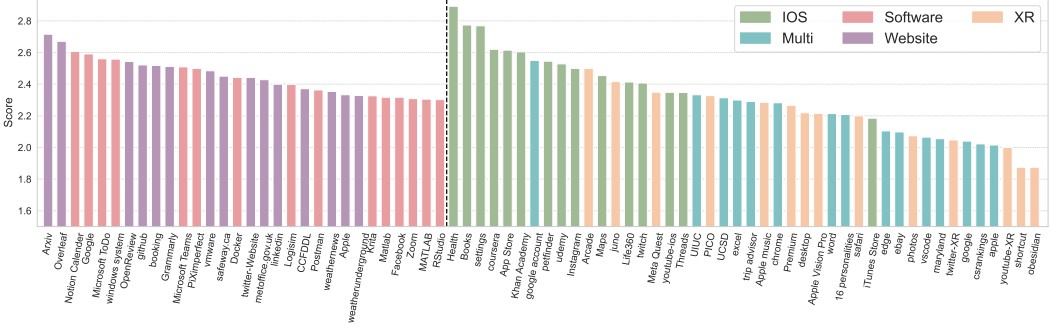

Figure 17: Fine-grained performance of Qwen-VL-Max in each software and website.

**Refining Human Annotation on Goal and Sub-goal**

```
As an expert in English, please refine the following English
instructions (or objectives) into a polished phrase or a
concise sentence.
Avoid including irrelevant content and provide the polished
output directly.
Here is the English sentence: {string}
```

Figure 18: Refining Human Annotation on Goal and Sub-goal.

---

**Guideline for Human Annotation**

**Main Interface**
1. Video List Panel (Left Panel): Displays a list of loaded video files. Each video file is shown with its name for identification.
2. Video Display Area (Center Panel): Shows the currently selected video for playback and annotation.
3. Control Settings (Right Panel):
Operating System: Select the operating system of the machine where the video was recorded.
Full Screen: Toggle full screen mode for the video display.
Multi-application?: Indicate if multiple applications in the video.
Application/Website: Enter the name of the application or website being used in the video.
User Goal: Enter the goal of the user performing the annotation.
4. Playback and Annotation Controls (Bottom Panel)
Annotate: Open a annotation window to add a new keyframe annotation.
Play: Starts or pauses the video playback.
Load Video: Allows you to load a single video file.
Load Video Folder: Allows to load multiple video files from a folder.
Previous Video / Next Video: Navigate through the loaded video files.
Save to JSON: Save the annotations in a JSON format.
**Annotation Window**
1. Mouse Action: Select a type of mouse action (e.g. click, drag).
2. Keyboard Action: Select the type of keyboard action (e.g., typing, key press).
3. Keyboard Operation Record: Enter details of the keyboard operation, if any.
4. sub-action Purpose: Describe the purpose of the action being annotated.
**How to Use**
**Loading Videos**
1. Load Multiple Videos
Click on the Load Video Folder button.
Select the folder containing your video files.
All video files in the folder will be loaded and listed in the Video List Panel.
**Playing Videos**
Select a video from the Video List Panel. Click the Play button to start or pause the video.
**Annotating Videos**
1. Start Annotation
Pause the video at the desired frame.
Click the Annotate button to open the annotation window.
2. Annotation Window
Select the Mouse Action Type and Keyboard Action Type from the dropdown menus.
If there is a keyboard action, enter the details in the Keyboard Operation Record field.
Describe the action's purpose in the Sub-action Purpose field.
Click OK to save the annotation.
**Saving Annotations**
Once all annotations are completed, click the Save to JSON button.

Figure 19: Guideline for Human Annotation.

**(Part 1) GPT-4V Generating GUI-oriented Tasks**

```
You are an AI visual assistant.  This is a video of a mobile GUI,
which I've divided into multiple frames and sent to you.  Please
provide a detailed description of what occurs throughout the entire
video, focusing on the changes in the GUI elements or scenes rather
than static aspects of a single frame.  The detailed description
should be placed under the key 'Description'.  Based on your
description, please design the following tasks:
Generate a precise caption for the video.  This caption should
encapsulate the main activities or changes observed throughout the
video sequence.  Place this caption under the key 'Caption'.
Create a free-form QA question related to the video's static GUI
content, along with its answer.  The question should delve into the
details or changes in the static GUI elements or scenes captured in
the video.  The QA task should be nested under the key Static QA,
with 'Question' and 'Answer' as subkeys.
Develop a multiple-choice QA question about the video, with four
options:  one correct answer and three incorrect or irrelevant
options.  This task should assess the understanding of specific
elements retieval or changes depicted in the video.  Structure
this task under the key MCQA, with 'Question' detailing the query,
'Options' listing the four choices including one correct answer, and
'Correct Answer' specifying the correct option, denoted, for example,
as {[[B]]}.
Here are some key information of the video to help you understand
the video comprehensively:
System:  {item['system']}
Application:  {item['app']}
Summary of the video:  {item['goal']}
Key Operation/Sub goal in the video:  {[i['sub_goal'] for i in
item['keyframes']]}
Notice:  Ensure that the questions you design for these tasks are
answerable and the answers can be deduced from the GUI video content.
The answerable question should be designed as difficult as possible.
The tasks should be unambiguous and the answers must be definitively
correct based on your understanding of the video content.  Only
include questions that have definite answers:  (1) one can see the
content in the image that the question asks about and can answer
confidently; (2) one can determine confidently from the image that
it is not in the image.  Do not ask any question that cannot be
answered confidently.
Each of these tasks should focus on the dynamic aspect of the GUI
elements or scenes.  Provide detailed answers when answering complex
questions.  For example, give detailed examples or reasoning steps
to make the content more convincing and well-organized.  The answers
should be in a tone that a visual AI assistant is seeing the image
and answering the question.
For the free-form QA tasks, please ensure that the answers are
as detailed and lengthy as possible, with no concern for length.
You can include multiple paragraphs if necessary to provide a
comprehensive and thorough response.  Please structure your
response using JSON format and specific keys mentioned in the task
requirements.
```

Figure 20: (Part 1) GPT-4V Generating GUI-oriented Tasks.

**(Part 2) GPT-4V Generating GUI-oriented Tasks.**

```
You are an AI visual assistant.  This is a video of a <Scene
Name> GUI, which I've divided into multiple frames and sent
to you.  Please provide a detailed description of what occurs
throughout the entire video, focusing on the changes in the
GUI elements or scenes rather than static aspects of a single
frame.  The detailed description should be placed under the
key 'Description'.  Based on your description, please design
the following tasks:
A Sequential QA task:  Design a question that requires
understanding the sequence of GUI element changes or scene
transformations in the video.  The question should be
free-form and necessitate the use of temporal information
from the sequential images.  The task should be structured
under the key Sequential-QA with subkeys 'Question' and
'Answer'.
A Next Stage Prediction task:  Formulate a question that asks
about the subsequent state or event following a certain frame
in the video.  The question should be designed in a free-form
manner and predict future GUI elements or scene changes,
structured under the key Prediction with subkeys 'Question'
and 'Answer'.
A two-round dialogue task:  Create a dialogue with two rounds
of interaction.  The first round includes a user instruction
and an assistant response, and the second round's user
instruction should be based on the response from the first
round.  Both rounds should be free-form and nested under the
key Conversation, with subkeys 'User 1', 'Assistant 1', 'User
2', and 'Assistant 2'.
A reasoning task:  Design a multi-choice QA task that
requires reasoning to identify the correct answer from four
options.  This task should test the reasoning ability to
infer or deduce information that is not explicitly provided.
It should be structured under the key Reasoning, with subkeys
'Question', 'Options', and 'Correct Answer'.
Here are some key information of the video to help you
understand the video comprehensively:
System: {item['system']}
Application: {item['app']}
Summary of the video: {item['goal']}
Key Operation/Sub goal in the video: {[i['sub_goal'] for i in
item['keyframes']]}
```

Figure 21: (Part 2) GPT-4V Generating GUI-oriented Tasks.

**(Part 3) GPT-4V Generating GUI-oriented Tasks.**

```
 Notice:  Ensure that the questions you design for these
tasks are answerable and the answers can be deduced from
the GUI video content.  The answerable question should be
designed as difficult as possible.  The tasks should be
unambiguous and the answers must be definitively correct
based on your understanding of the video content.  Only
include questions that have definite answers:  (1) one can
see the content in the image that the question asks about
and can answer confidently; (2) one can determine confidently
from the image that it is not in the image.  Do not ask any
question that cannot be answered confidently.
Each of these tasks should focus on the dynamic aspect
of the GUI elements or scenes, with each answerable task
as difficult as possible.  Provide detailed answers when
answering complex questions.  For example, give detailed
examples or reasoning steps to make the content more
convincing and well-organized.  The answers should be in
a tone that a visual AI assistant is seeing the image and
answering the question.
For the free-form QA tasks, please ensure that the answers
are as detailed and lengthy as possible, with no concern for
length.  You can include multiple paragraphs if necessary
to provide a comprehensive and thorough response.  Please
structure your response using JSON format and specific keys
mentioned in the task requirements.
```

Figure 22: (Part 3) GPT-4V Generating GUI-oriented Tasks.

**Prompts for Benchmarking MLLMs**

"XR": "You are an AI visual assistant. Here are sequential images of Mixed-Reality combining GUI interface and real world, which are selected from a GUI video.",
"software": "You are an AI visual assistant. Here are sequential GUI interface images of a specific software, which are selected from a GUI video.",
"website": "You are an AI visual assistant. Here are sequential GUI interface images of a desktop website, which are selected from a GUI video.",
"mobile": "You are an AI visual assistant. Here are sequential GUI mobile interface images, which are selected from a GUI video.",
"multi": "You are an AI visual assistant. Here are sequential GUI interface images of interaction among multiple softwares and websites, which are selected from a GUI video.",
"IOS": "You are an AI visual assistant. Here are sequential GUI IOS interface images, which are selected from a GUI video.",

"Sequential-QA": "This is a question about sequential information in sequential images.",
"Prediction": "This is a question about predicting the next action base on the previous actions in the sequential images.",
"Reasoning": "This is a multiple choice question with only one correct answer. This question may need multiple steps of reasoning according to the vision information in sequential images.",
"Description 1": "Please give me a detail description of these sequential images.",
"Description 2": "Offer a thorough analysis of these sequential images",
"Caption": "Please give me a concise caption of these sequential images.",
"static QA": "This is a question about static information such as text, icon, layout in these sequential images.",
"MCQA": "This is a multiple choice question with only one correct answer. This question may require sequential analysis ability to the vision information in these sequential images.",
"Conversation 1": "Act as an assistant to answer the user's question in these sequential images.",
"Conversation 2": "This is a multi-turn conversation task. You will be provide the first round conversation and act as an assistant to answer the user's question in the second round according to these sequential images."
Notice = "You can first provide an overall description of these sequential images, and then analyze the user's question according to the sequential images and description. Finally, give an answer based on this description and the image information. Please format your output in a JSON format, with key 'Description' for the description of these sequential images, key 'Analysis' for your analysis on the user's question and key 'Answer' for your answer to the User's question."

Figure 23: Prompts for Benchmarking MLLMs.

---

**Prompt for LLM-as-a-Judge: Judging Free-form and Conversational Tasks**

```
 You are an impartial judge.  I will provide you with a
question, a 'gold standard' answer, and a response that
needs evaluation.  Your task is to assess the quality of the
response in comparison to the 'gold standard' answer.  Please
adhere to the following guidelines:

1.  Start your evaluation by comparing the response to
the 'gold standard' answer.  Offer a brief explanation
highlighting similarities and differences, focusing on
relevance, accuracy, depth, and level of detail.
2.  Conclude your evaluation with a score from 1 to 5, where
1 indicates the response is mostly irrelevant to the 'gold
standard' answer, and 5 indicates it is very similar or
equivalent.
3.  Present your findings in JSON format, using 'Evaluation'
for your textual analysis and 'Score' for the numerical
assessment.
4.  Ensure objectivity in your evaluation.  Avoid biases and
strive for an even distribution of scores across the spectrum
of quality.  Your scoring must be as rigorous as possible and
adhere to the following rules:
- Overall, the higher the quality of the model's response,
the higher the score, with factual accuracy and meeting user
needs being the most critical dimensions.  These two factors
largely dictate the final composite score.
- If the model's response is irrelevant to the question,
contains fundamental factual errors, or generates harmful
content, the total score must be 1.
- If the model's response has no severe errors and is
essentially harmless, but of low quality and does not meet
user needs, the total score should be 2.
- If the model's response generally meets user requirements
but performs poorly in certain aspects with medium quality,
the total score should be 3.
- If the model's response is close in quality to the reference
answer and performs well in all dimensions, the total score
should be 4.
- Only when the model's response surpasses the reference
answer, fully addresses the user's problem and all needs,
and nearly achieves a perfect score in all dimensions, can
it receive a score between 5.
- As an example, the golden answer could receive a 4-5.

Here is the response for you to judge:
Question:  {question}
Golden Answer:  {golden_answer}
Response:  {response}
Now, directly output your response in json format.
```

Figure 24: Prompt for LLM-as-a-Judge: Judging Free-form and Conversational Tasks .

---

**Prompt for LLM-as-a-Judge: Judging Multiple-Choice QA Tasks**

You are a helpful assistant tasked with judging a Multiple
Choice Question Answering exercise.
I will provide a correct answer with only one option, and a
response that requires evaluation.
If the response matches the correct answer, simply output
"Yes"; If it does not, output "No".
Please avoid including any irrelevant information.
Here are some examples:

Example 1:
Question:  Based on the GUI video, why might the 'Loading'
animation continue without reaching the next stage?  A. The
user has not yet entered their login credentials.  B. There
is a system update being installed.  C. The server is taking
time to authenticate the login credentials.  D. The 'Log In'
button is malfunctioning.
Answer:  C
Response:  C. The server is taking time to authenticate the
login credentials.
Output:  Yes

Example 2:
Question:  If the user wants to resume the group video call
after checking messages, what action should they take?  A.
Turn their head to the right.  B. Close the messaging app
interface.  C. Say a voice command to switch applications.  D.
Turn their head to the left.
Answer:  A
Response:  B
Output:  No

Example 3:
Question:  What action does the user take to start playing
music in the video?  A. Closed the music player application B.
Moved the music player to a new position C. Clicked the play
button D. Adjusted the system volume
Answer:  [[B]]
Response:  C
Output:  No

Here is the question, answer, and response for you to judge:
Question:  {question}
Answer:  {answer}
Response:  {response}
Now, directly output "Yes" or "No".

Figure 25: Prompt for LLM-as-a-Judge: Judging Multiple-Choice QA Tasks.

# F  CASE STUDY

In this section, we provide detailed case studies for six GUI scenarios, each divided into two parts. The following examples show human-annotated frames and various tasks associated with them:

- **Android:** Figures 26 and 27.
- **IOS:** Figures 28 and 29.
- **Multiple-windows Interaction:** Figures 30 and 31.
- **Website:** Figures 34 and 35.
- **XR:** Figures 36 and 37.
- **Software:** Figures 32 and 33.

**(Part 1) Android**

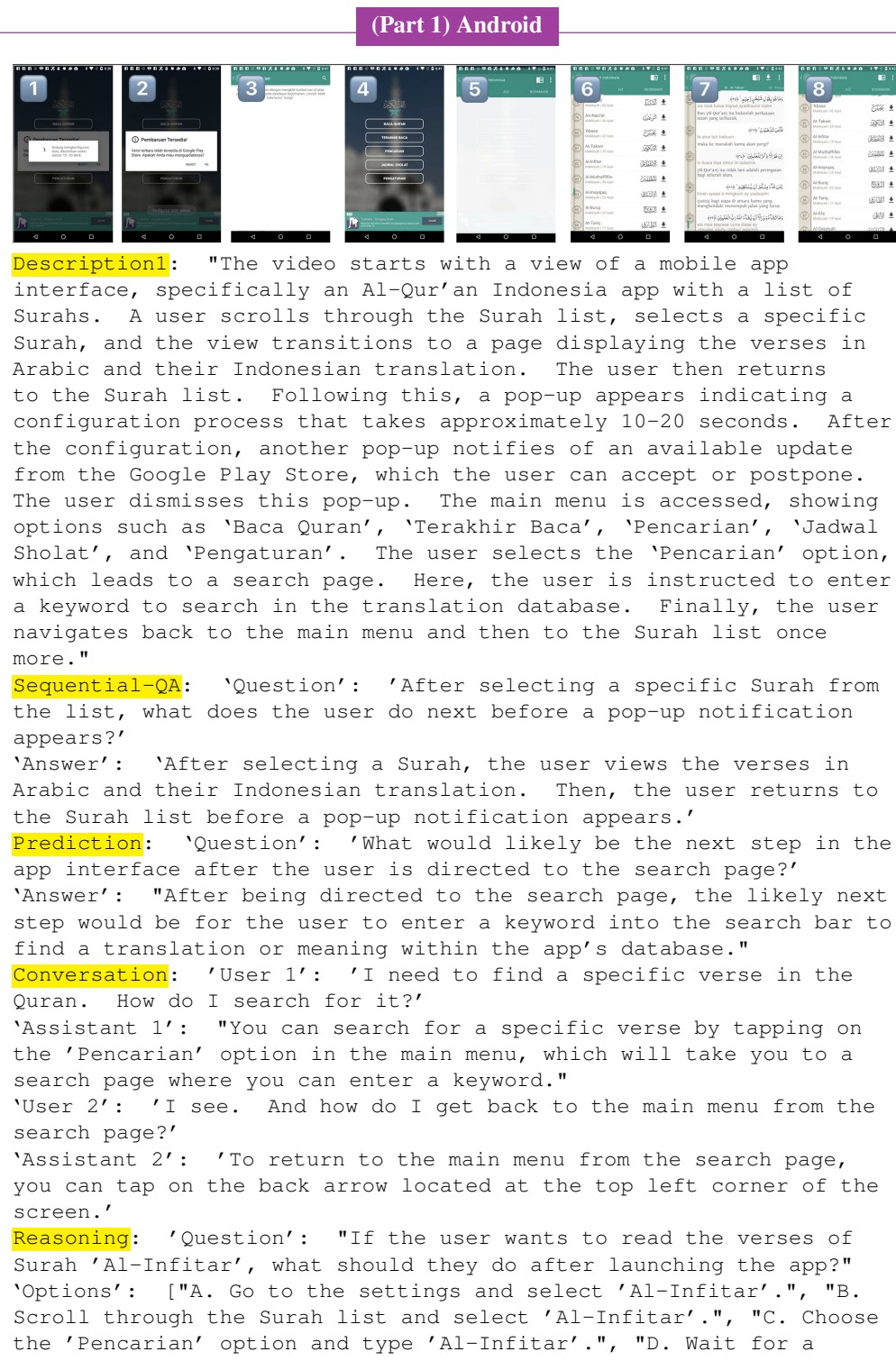

Description1: "The video starts with a view of a mobile app interface, specifically an Al-Qur'an Indonesia app with a list of Surahs. A user scrolls through the Surah list, selects a specific Surah, and the view transitions to a page displaying the verses in Arabic and their Indonesian translation. The user then returns to the Surah list. Following this, a pop-up appears indicating a configuration process that takes approximately 10-20 seconds. After the configuration, another pop-up notifies of an available update from the Google Play Store, which the user can accept or postpone. The user dismisses this pop-up. The main menu is accessed, showing options such as 'Baca Quran', 'Terakhir Baca', 'Pencarian', 'Jadwal Sholat', and 'Pengaturan'. The user selects the 'Pencarian' option, which leads to a search page. Here, the user is instructed to enter a keyword to search in the translation database. Finally, the user navigates back to the main menu and then to the Surah list once more."

Sequential-QA: 'Question': 'After selecting a specific Surah from the list, what does the user do next before a pop-up notification appears?'
'Answer': 'After selecting a Surah, the user views the verses in Arabic and their Indonesian translation. Then, the user returns to the Surah list before a pop-up notification appears.'

Prediction: 'Question': 'What would likely be the next step in the app interface after the user is directed to the search page?'
'Answer': "After being directed to the search page, the likely next step would be for the user to enter a keyword into the search bar to find a translation or meaning within the app's database."

Conversation: 'User 1': 'I need to find a specific verse in the Quran. How do I search for it?'
'Assistant 1': "You can search for a specific verse by tapping on the 'Pencarian' option in the main menu, which will take you to a search page where you can enter a keyword."
'User 2': 'I see. And how do I get back to the main menu from the search page?'
'Assistant 2': 'To return to the main menu from the search page, you can tap on the back arrow located at the top left corner of the screen.'

Reasoning: 'Question': "If the user wants to read the verses of Surah 'Al-Infitar', what should they do after launching the app?"
'Options': ["A. Go to the settings and select 'Al-Infitar'.", "B. Scroll through the Surah list and select 'Al-Infitar'.", "C. Choose the 'Pencarian' option and type 'Al-Infitar'.", "D. Wait for a pop-up and select 'Al-Infitar' from there."]
'Correct Answer': "B. Scroll through the Surah list and select 'Al-Infitar'."

Figure 26: Case study for Android (part 1).

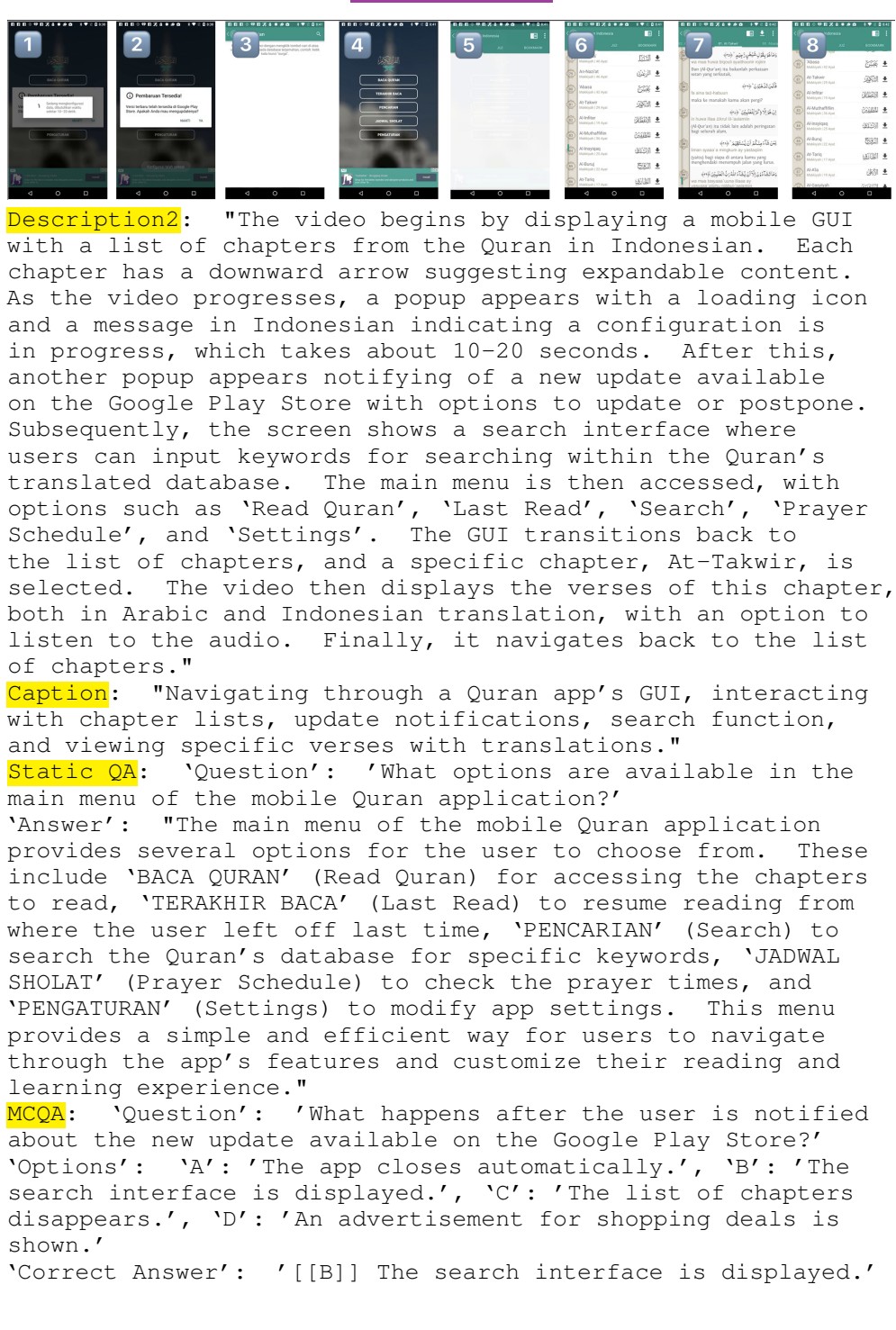

**(Part 2) Android**

Description2: "The video begins by displaying a mobile GUI with a list of chapters from the Quran in Indonesian. Each chapter has a downward arrow suggesting expandable content. As the video progresses, a popup appears with a loading icon and a message in Indonesian indicating a configuration is in progress, which takes about 10-20 seconds. After this, another popup appears notifying of a new update available on the Google Play Store with options to update or postpone. Subsequently, the screen shows a search interface where users can input keywords for searching within the Quran's translated database. The main menu is then accessed, with options such as 'Read Quran', 'Last Read', 'Search', 'Prayer Schedule', and 'Settings'. The GUI transitions back to the list of chapters, and a specific chapter, At-Takwir, is selected. The video then displays the verses of this chapter, both in Arabic and Indonesian translation, with an option to listen to the audio. Finally, it navigates back to the list of chapters."

Caption: "Navigating through a Quran app's GUI, interacting with chapter lists, update notifications, search function, and viewing specific verses with translations."

Static QA: 'Question': 'What options are available in the main menu of the mobile Quran application?'
'Answer': "The main menu of the mobile Quran application provides several options for the user to choose from. These include 'BACA QURAN' (Read Quran) for accessing the chapters to read, 'TERAKHIR BACA' (Last Read) to resume reading from where the user left off last time, 'PENCARIAN' (Search) to search the Quran's database for specific keywords, 'JADWAL SHOLAT' (Prayer Schedule) to check the prayer times, and 'PENGATURAN' (Settings) to modify app settings. This menu provides a simple and efficient way for users to navigate through the app's features and customize their reading and learning experience."

MCQA: 'Question': 'What happens after the user is notified about the new update available on the Google Play Store?'
'Options': 'A': 'The app closes automatically.', 'B': 'The search interface is displayed.', 'C': 'The list of chapters disappears.', 'D': 'An advertisement for shopping deals is shown.'
'Correct Answer': '[[B]] The search interface is displayed.'

Figure 27: Case study for Android (part 2).

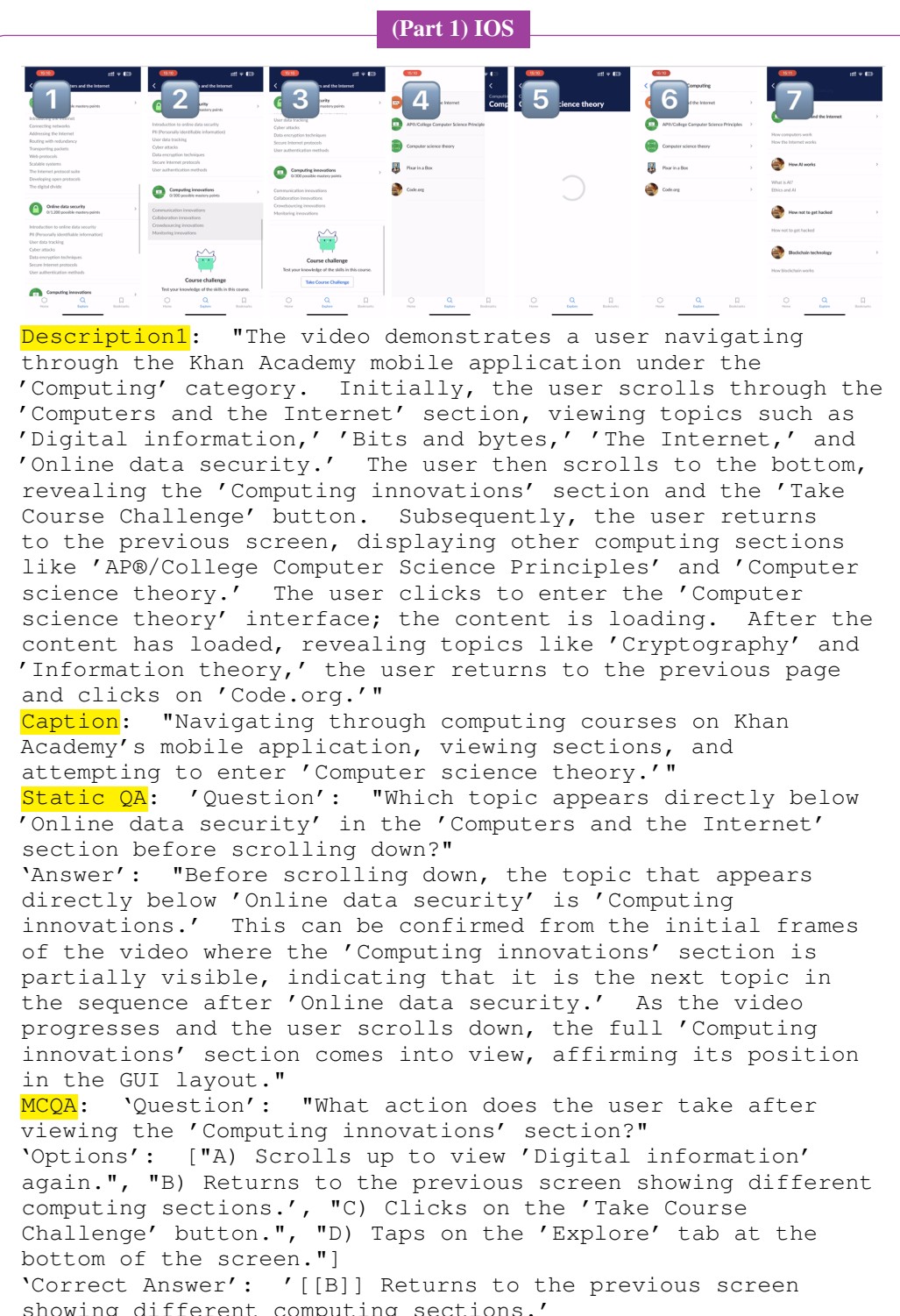

**(Part 1) IOS**

Description1: "The video demonstrates a user navigating through the Khan Academy mobile application under the 'Computing' category. Initially, the user scrolls through the 'Computers and the Internet' section, viewing topics such as 'Digital information,' 'Bits and bytes,' 'The Internet,' and 'Online data security.' The user then scrolls to the bottom, revealing the 'Computing innovations' section and the 'Take Course Challenge' button. Subsequently, the user returns to the previous screen, displaying other computing sections like 'AP®/College Computer Science Principles' and 'Computer science theory.' The user clicks to enter the 'Computer science theory' interface; the content is loading. After the content has loaded, revealing topics like 'Cryptography' and 'Information theory,' the user returns to the previous page and clicks on 'Code.org.'"

Caption: "Navigating through computing courses on Khan Academy's mobile application, viewing sections, and attempting to enter 'Computer science theory.'"

Static QA: 'Question': "Which topic appears directly below 'Online data security' in the 'Computers and the Internet' section before scrolling down?"
'Answer': "Before scrolling down, the topic that appears directly below 'Online data security' is 'Computing innovations.' This can be confirmed from the initial frames of the video where the 'Computing innovations' section is partially visible, indicating that it is the next topic in the sequence after 'Online data security.' As the video progresses and the user scrolls down, the full 'Computing innovations' section comes into view, affirming its position in the GUI layout."

MCQA: 'Question': "What action does the user take after viewing the 'Computing innovations' section?"
'Options': ["A) Scrolls up to view 'Digital information' again.", "B) Returns to the previous screen showing different computing sections.', "C) Clicks on the 'Take Course Challenge' button.", "D) Taps on the 'Explore' tab at the bottom of the screen."]
'Correct Answer': '[[B]] Returns to the previous screen showing different computing sections.'

Figure 28: Case study for IOS (part 1).

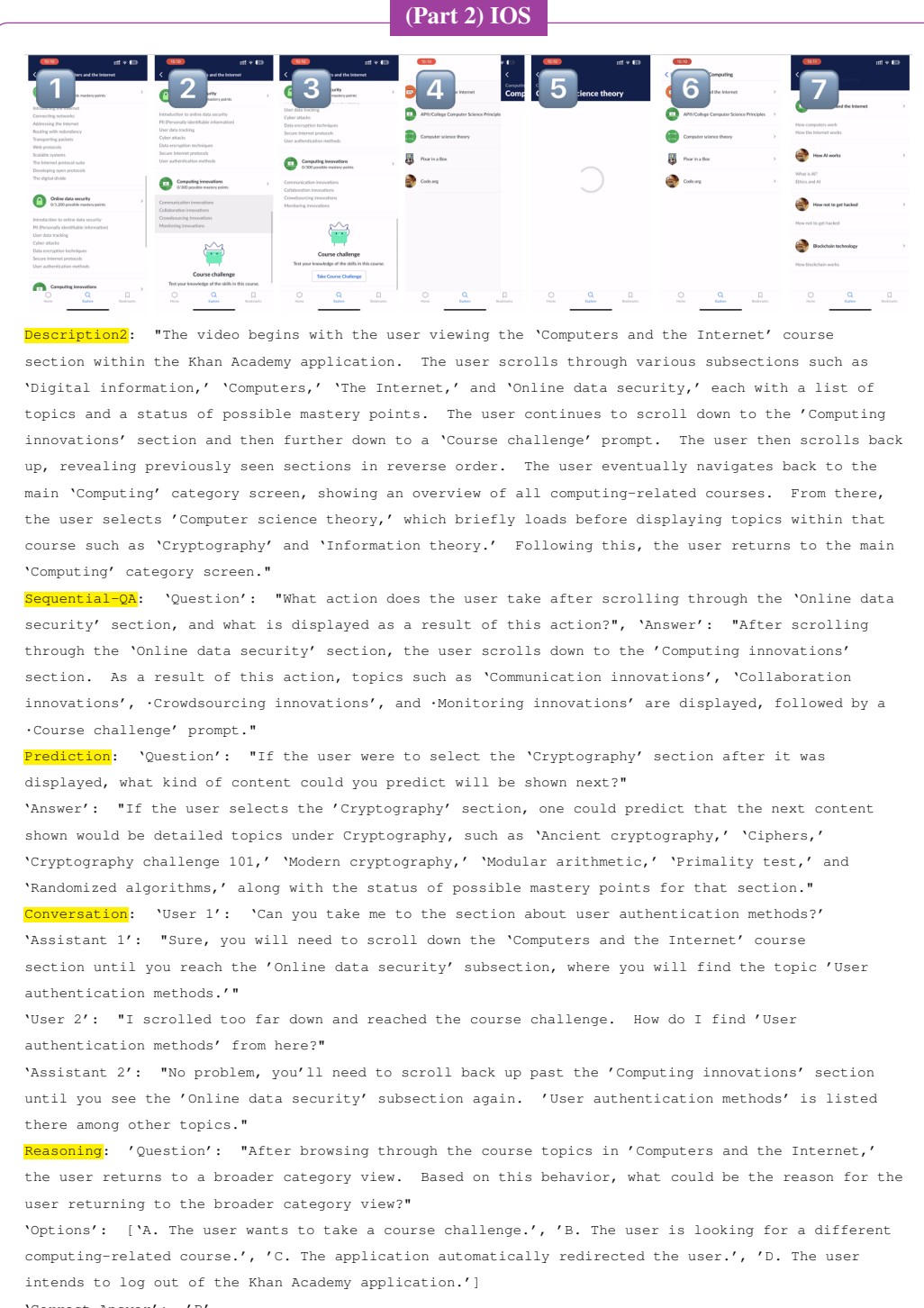

**(Part 2) IOS**

Description2: "The video begins with the user viewing the 'Computers and the Internet' course section within the Khan Academy application. The user scrolls through various subsections such as 'Digital information,' 'Computers,' 'The Internet,' and 'Online data security,' each with a list of topics and a status of possible mastery points. The user continues to scroll down to the 'Computing innovations' section and then further down to a 'Course challenge' prompt. The user then scrolls back up, revealing previously seen sections in reverse order. The user eventually navigates back to the main 'Computing' category screen, showing an overview of all computing-related courses. From there, the user selects 'Computer science theory,' which briefly loads before displaying topics within that course such as 'Cryptography' and 'Information theory.' Following this, the user returns to the main 'Computing' category screen."

Sequential-QA: 'Question': "What action does the user take after scrolling through the 'Online data security' section, and what is displayed as a result of this action?", 'Answer': "After scrolling through the 'Online data security' section, the user scrolls down to the 'Computing innovations' section. As a result of this action, topics such as 'Communication innovations', 'Collaboration innovations', ·Crowdsourcing innovations', and ·Monitoring innovations' are displayed, followed by a ·Course challenge' prompt."

Prediction: 'Question': "If the user were to select the 'Cryptography' section after it was displayed, what kind of content could you predict will be shown next?"
'Answer': "If the user selects the 'Cryptography' section, one could predict that the next content shown would be detailed topics under Cryptography, such as 'Ancient cryptography,' 'Ciphers,' 'Cryptography challenge 101,' 'Modern cryptography,' 'Modular arithmetic,' 'Primality test,' and 'Randomized algorithms,' along with the status of possible mastery points for that section."

Conversation: 'User 1': 'Can you take me to the section about user authentication methods?'
'Assistant 1': "Sure, you will need to scroll down the 'Computers and the Internet' course section until you reach the 'Online data security' subsection, where you will find the topic 'User authentication methods.'"
'User 2': "I scrolled too far down and reached the course challenge. How do I find 'User authentication methods' from here?"
'Assistant 2': "No problem, you'll need to scroll back up past the 'Computing innovations' section until you see the 'Online data security' subsection again. 'User authentication methods' is listed there among other topics."

Reasoning: 'Question': "After browsing through the course topics in 'Computers and the Internet,' the user returns to a broader category view. Based on this behavior, what could be the reason for the user returning to the broader category view?"
'Options': ['A. The user wants to take a course challenge.', 'B. The user is looking for a different computing-related course.', 'C. The application automatically redirected the user.', 'D. The user intends to log out of the Khan Academy application.']
'Correct Answer': 'B'

Figure 29: Case study for IOS (part 2).

**(Part 1) Multiple-Windows Interaction**

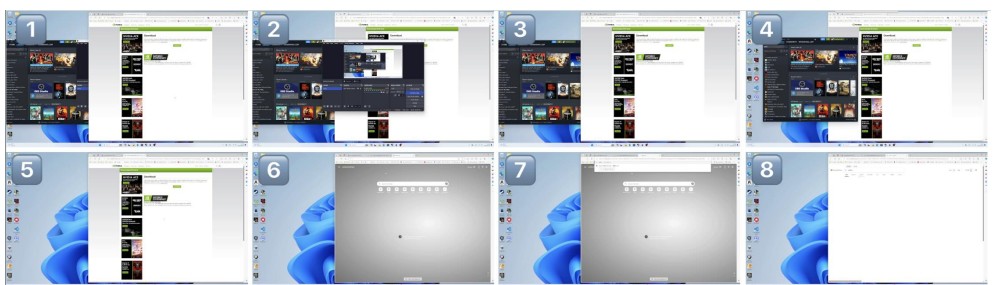

Description1: "The video begins with a Windows desktop
displaying multiple open applications, including Steam, OBS
Studio, and a web browser with NVIDIA's website loaded. The
user starts by clicking on the back page of the browser,
which partially obscures the OBS window. Then, the user
clicks on the OBS application, bringing it to the forefront.
The user minimizes OBS, followed by dragging the Steam window
to the center of the screen and minimizing it as well. A
new web page is opened in the Edge browser's navigation
bar, and the user types 'office' into the search bar. The
browser navigates to the Bing search interface, and 'office'
is successfully searched."
Caption: 'Navigating and Managing Multiple Applications on
Windows Including Steam, OBS Studio, and Edge Browser'
Static QA: 'Question': "Which web browser is used in the
video and which website is prominently featured before the
search for 'office'?"
'Answer': "The web browser used in the video is Microsoft
Edge. The prominently featured website before the search
for 'office' is NVIDIA's official website where the 'Download
Drivers' page is displayed."
MCQA: 'Question': 'What action is taken after the OBS
application is minimized?', 'Options': ['A. The Steam window
is closed.'
'B. The Steam window is moved to the center of the screen and
minimized.', 'C. The Edge browser is closed.', 'D. A file is
opened from the desktop.']
'Correct Answer': '[[B]] The Steam window is moved to the
center of the screen and minimized.'

Figure 30: Case study for multiple-windows interaction (part 1).

**(Part 2) Multiple-Windows Interaction**

'Description2': "The video shows a Windows desktop with several application windows open, including Steam, OBS, and Edge. Initially, the OBS window is partially covering the Edge browser. The user clicks on the Edge browser, bringing it to the foreground, and then minimizes the OBS window, clearing the view. Next, the user moves the Steam window to the center of the screen and minimizes it as well. Afterward, the user opens a new tab in Edge and enters 'office' into the search bar, which leads to a Bing search results page for 'office'. The video ends with the user on the Bing search page with 'office' results displayed."

Sequential-QA: 'Question': 'After moving the Steam window to the center, what did the user do next in the Edge browser?'
'Answer': "After moving the Steam window to the center, the user minimized the Steam application. Then, they opened a new tab in the Edge browser and entered 'office' into the search bar, which led them to the Bing search results page for 'office'."

Prediction: 'Question': "What would be the likely next action the user performs after searching for 'office' on Bing?"
'Answer': "The likely next action the user would perform after searching for 'office' on Bing would be to click on one of the search results to find more information or to visit the Microsoft Office website to purchase or learn more about the office suite."

Conversation: 'User 1': 'Can you minimize the OBS application for a better view of the browser?'
'Assistant 1': 'Certainly, the OBS application has been minimized, providing a clear view of the Edge browser.'
'User 2': 'Great, now can you search for Microsoft Office in the Edge browser?'
'Assistant 2': "Of course, a new tab has been opened in the Edge browser and 'office' has been entered into the search bar. The Bing search results for 'office' are now displayed."

Reasoning: 'Question': 'If the user needs to record gameplay footage next, which application should they interact with and what would be their first step?'
'Options': ["A. They should open the Steam application and click on the 'STORE' tab.", "B. They should open the Edge browser and search for 'game recording software'.", "C. They should reopen the OBS application and click on the 'Start Recording' button.", "D. They should access the Windows Start menu and search for the 'Camera' app."]
'Correct Answer': 'C'

Figure 31: Case study for multiple-windows interaction (part 2).

**(Part 1) Software**

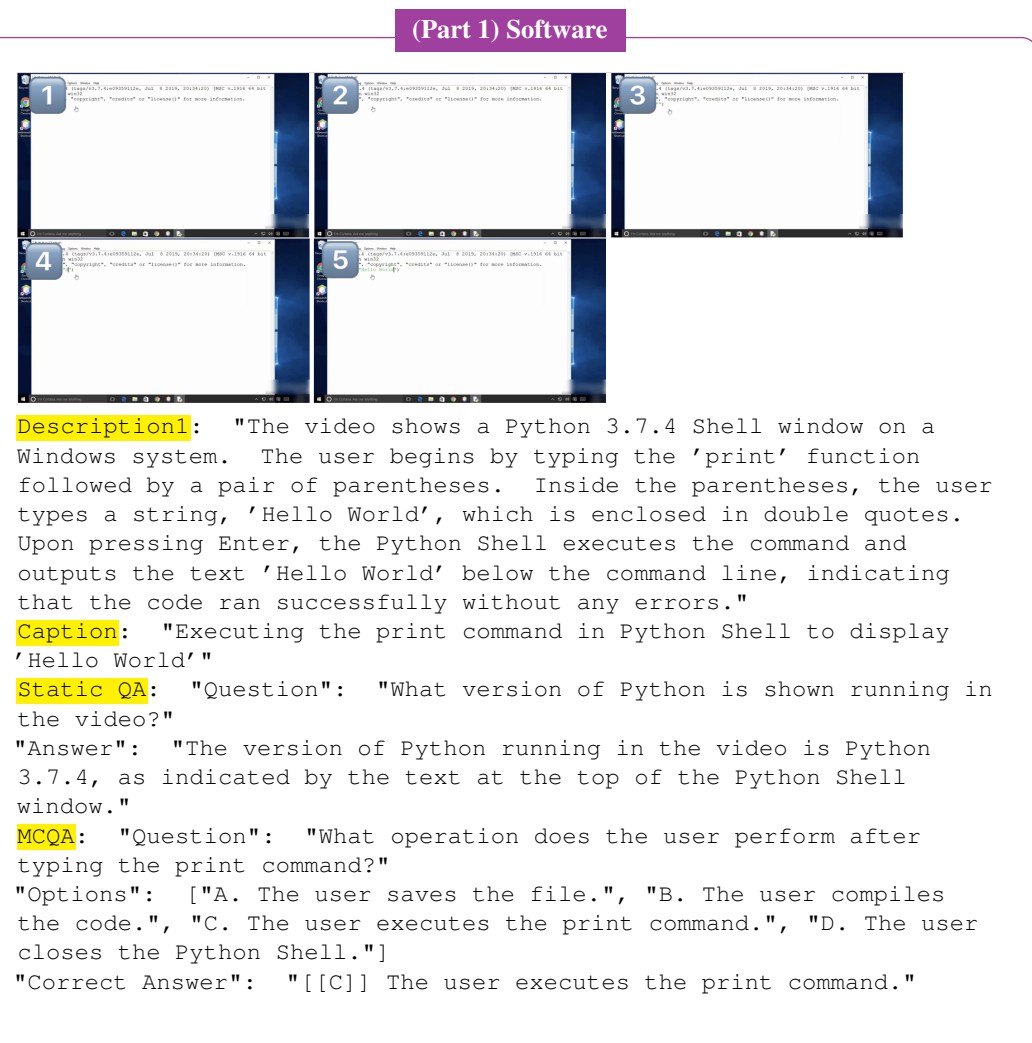

**Description1**: "The video shows a Python 3.7.4 Shell window on a Windows system. The user begins by typing the 'print' function followed by a pair of parentheses. Inside the parentheses, the user types a string, 'Hello World', which is enclosed in double quotes. Upon pressing Enter, the Python Shell executes the command and outputs the text 'Hello World' below the command line, indicating that the code ran successfully without any errors."

**Caption**: "Executing the print command in Python Shell to display 'Hello World'"

**Static QA**: "Question": "What version of Python is shown running in the video?"
"Answer": "The version of Python running in the video is Python 3.7.4, as indicated by the text at the top of the Python Shell window."

**MCQA**: "Question": "What operation does the user perform after typing the print command?"
"Options": ["A. The user saves the file.", "B. The user compiles the code.", "C. The user executes the print command.", "D. The user closes the Python Shell."]
"Correct Answer": "[[C]] The user executes the print command."

Figure 32: Case study for software (part 1).

**(Part 2) Software**

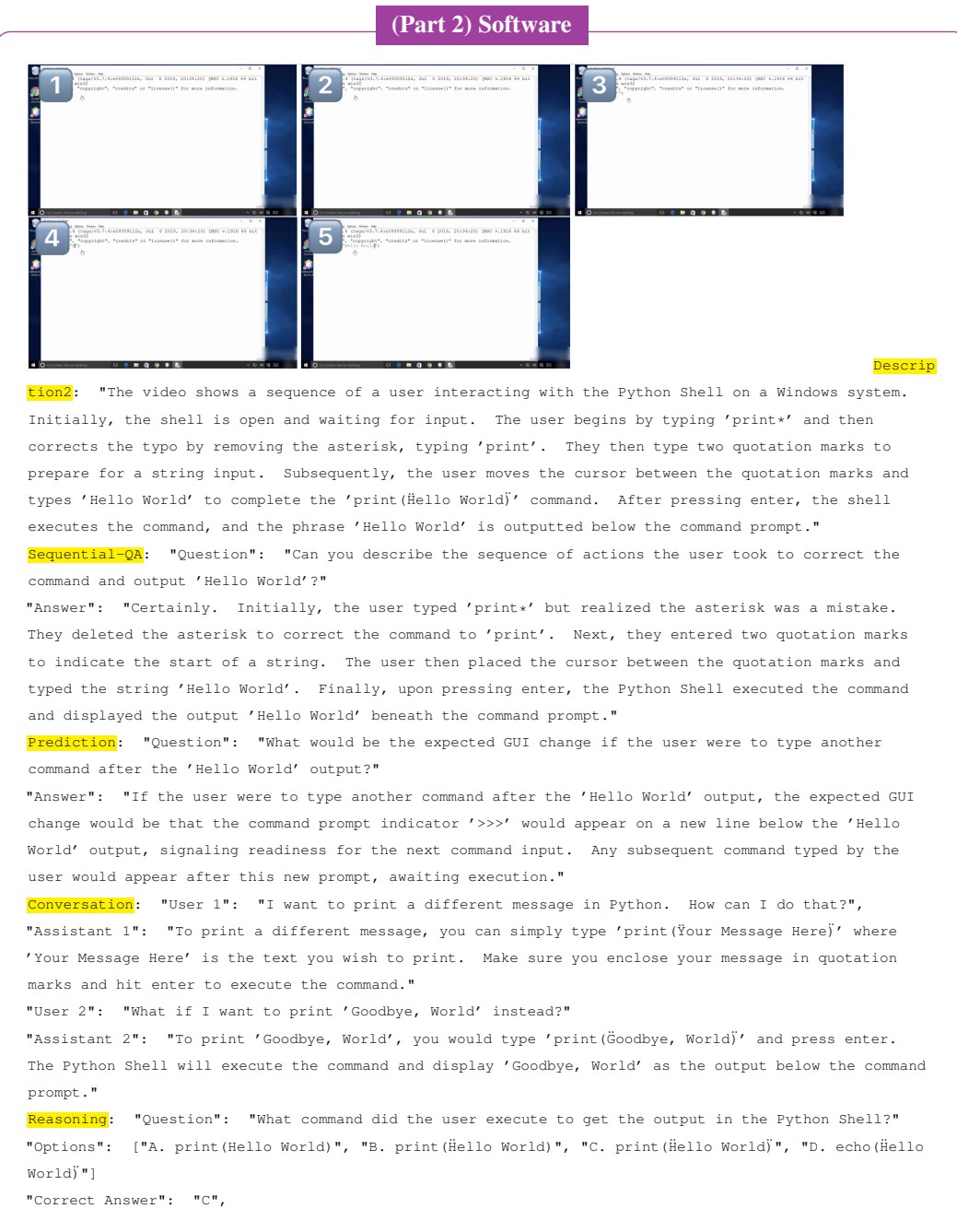

**Description2:** "The video shows a sequence of a user interacting with the Python Shell on a Windows system. Initially, the shell is open and waiting for input. The user begins by typing 'print*' and then corrects the typo by removing the asterisk, typing 'print'. They then type two quotation marks to prepare for a string input. Subsequently, the user moves the cursor between the quotation marks and types 'Hello World' to complete the 'print(Ḧello World)̈' command. After pressing enter, the shell executes the command, and the phrase 'Hello World' is outputted below the command prompt."

**Sequential-QA:** "Question": "Can you describe the sequence of actions the user took to correct the command and output 'Hello World'?"

"Answer": "Certainly. Initially, the user typed 'print*' but realized the asterisk was a mistake. They deleted the asterisk to correct the command to 'print'. Next, they entered two quotation marks to indicate the start of a string. The user then placed the cursor between the quotation marks and typed the string 'Hello World'. Finally, upon pressing enter, the Python Shell executed the command and displayed the output 'Hello World' beneath the command prompt."

**Prediction:** "Question": "What would be the expected GUI change if the user were to type another command after the 'Hello World' output?"

"Answer": "If the user were to type another command after the 'Hello World' output, the expected GUI change would be that the command prompt indicator '>>>' would appear on a new line below the 'Hello World' output, signaling readiness for the next command input. Any subsequent command typed by the user would appear after this new prompt, awaiting execution."

**Conversation:** "User 1": "I want to print a different message in Python. How can I do that?",

"Assistant 1": "To print a different message, you can simply type 'print(Ÿour Message Here)̈' where 'Your Message Here' is the text you wish to print. Make sure you enclose your message in quotation marks and hit enter to execute the command."

"User 2": "What if I want to print 'Goodbye, World' instead?"

"Assistant 2": "To print 'Goodbye, World', you would type 'print(G̈oodbye, World)̈' and press enter. The Python Shell will execute the command and display 'Goodbye, World' as the output below the command prompt."

**Reasoning:** "Question": "What command did the user execute to get the output in the Python Shell?"

"Options": ["A. print(Hello World)", "B. print(Ḧello World)", "C. print(Ḧello World)̈", "D. echo(Ḧello World)̈"]

"Correct Answer": "C",

Figure 33: Case study for software (part 2).

**(Part 1) Website**

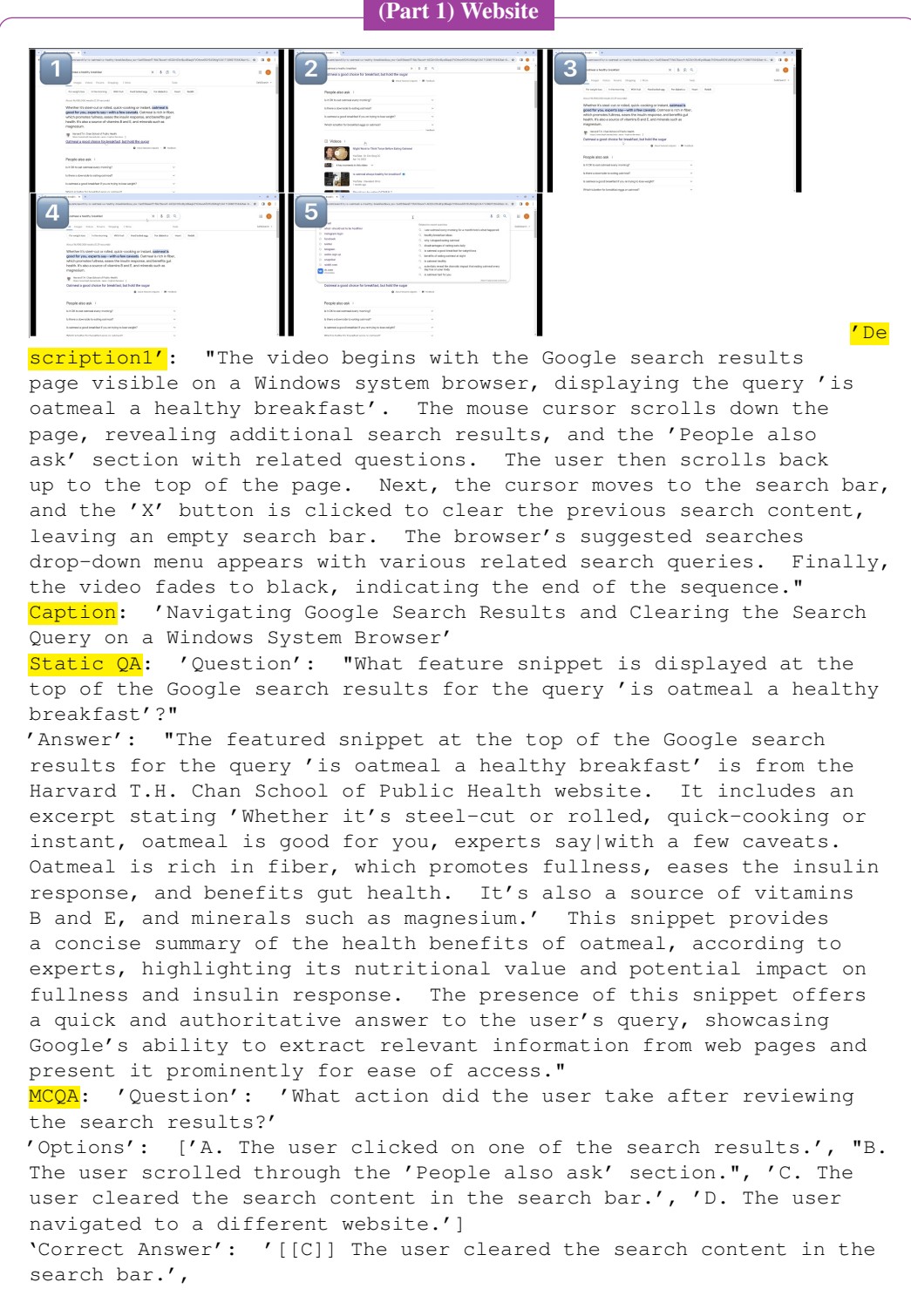

'Description1': "The video begins with the Google search results page visible on a Windows system browser, displaying the query 'is oatmeal a healthy breakfast'. The mouse cursor scrolls down the page, revealing additional search results, and the 'People also ask' section with related questions. The user then scrolls back up to the top of the page. Next, the cursor moves to the search bar, and the 'X' button is clicked to clear the previous search content, leaving an empty search bar. The browser's suggested searches drop-down menu appears with various related search queries. Finally, the video fades to black, indicating the end of the sequence."

Caption: 'Navigating Google Search Results and Clearing the Search Query on a Windows System Browser'

Static QA: 'Question': "What feature snippet is displayed at the top of the Google search results for the query 'is oatmeal a healthy breakfast'?"

'Answer': "The featured snippet at the top of the Google search results for the query 'is oatmeal a healthy breakfast' is from the Harvard T.H. Chan School of Public Health website. It includes an excerpt stating 'Whether it's steel-cut or rolled, quick-cooking or instant, oatmeal is good for you, experts say|with a few caveats. Oatmeal is rich in fiber, which promotes fullness, eases the insulin response, and benefits gut health. It's also a source of vitamins B and E, and minerals such as magnesium.' This snippet provides a concise summary of the health benefits of oatmeal, according to experts, highlighting its nutritional value and potential impact on fullness and insulin response. The presence of this snippet offers a quick and authoritative answer to the user's query, showcasing Google's ability to extract relevant information from web pages and present it prominently for ease of access."

MCQA: 'Question': 'What action did the user take after reviewing the search results?'

'Options': ['A. The user clicked on one of the search results.', "B. The user scrolled through the 'People also ask' section.", 'C. The user cleared the search content in the search bar.', 'D. The user navigated to a different website.']

'Correct Answer': '[[C]] The user cleared the search content in the search bar.',

Figure 34: Case study for website (part 1).

**(Part 2) Website**

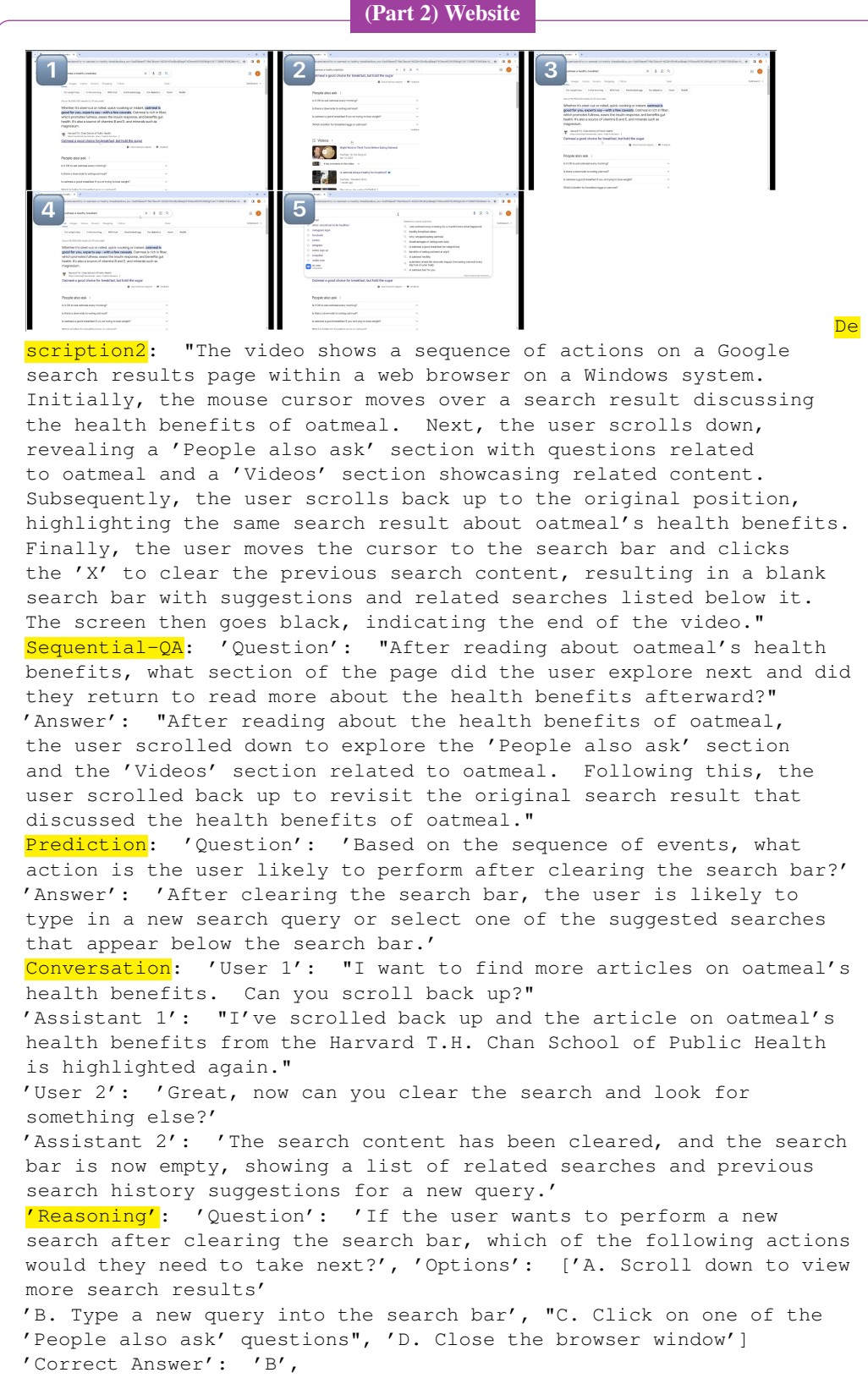

Description2: "The video shows a sequence of actions on a Google search results page within a web browser on a Windows system. Initially, the mouse cursor moves over a search result discussing the health benefits of oatmeal. Next, the user scrolls down, revealing a 'People also ask' section with questions related to oatmeal and a 'Videos' section showcasing related content. Subsequently, the user scrolls back up to the original position, highlighting the same search result about oatmeal's health benefits. Finally, the user moves the cursor to the search bar and clicks the 'X' to clear the previous search content, resulting in a blank search bar with suggestions and related searches listed below it. The screen then goes black, indicating the end of the video."

Sequential-QA: 'Question': "After reading about oatmeal's health benefits, what section of the page did the user explore next and did they return to read more about the health benefits afterward?" 'Answer': "After reading about the health benefits of oatmeal, the user scrolled down to explore the 'People also ask' section and the 'Videos' section related to oatmeal. Following this, the user scrolled back up to revisit the original search result that discussed the health benefits of oatmeal."

Prediction: 'Question': 'Based on the sequence of events, what action is the user likely to perform after clearing the search bar?' 'Answer': 'After clearing the search bar, the user is likely to type in a new search query or select one of the suggested searches that appear below the search bar.'

Conversation: 'User 1': "I want to find more articles on oatmeal's health benefits. Can you scroll back up?" 'Assistant 1': "I've scrolled back up and the article on oatmeal's health benefits from the Harvard T.H. Chan School of Public Health is highlighted again." 'User 2': 'Great, now can you clear the search and look for something else?' 'Assistant 2': 'The search content has been cleared, and the search bar is now empty, showing a list of related searches and previous search history suggestions for a new query.'

'Reasoning': 'Question': 'If the user wants to perform a new search after clearing the search bar, which of the following actions would they need to take next?', 'Options': ['A. Scroll down to view more search results' 'B. Type a new query into the search bar', "C. Click on one of the 'People also ask' questions", 'D. Close the browser window'] 'Correct Answer': 'B',

Figure 35: Case study for website (part 2).

**(Part 1) XR**

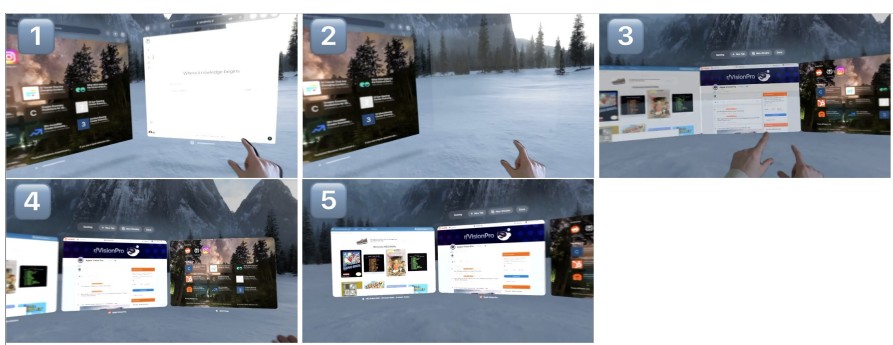

Description1: "The video showcases a user navigating through various pages within the Apple Vision Pro browser on a Windows system. Initially, the browser displays the start page with Favorites and Reading List. The user then turns their head to the right, which triggers the transition to view a webpage on the right side. Following this, the user pinches with both hands to exit the page and then pinches with both hands and fingers moving towards the middle to expand the browser's various pages. This reveals multiple open browser tabs side by side. The user continues to turn their head left and right to view different pages on each side. Lastly, the user selects and expands a specific tab to fill the screen, displaying its content."

Caption: 'Navigating through multiple browser pages using head movement and hand gestures in Apple Vision Pro on Windows'

Static QA: 'Question': "What is the main category listed under the Favorites section on the browser's start page?"
'Answer': "The main category listed under the Favorites section on the browser's start page is 'Perplexity', denoted by a unique icon, followed by other favorites like Instagram and various websites."

MCQA: 'Question': 'How does the user switch between different open tabs in the Apple Vision Pro browser?'
'Options': ['A. Using keyboard shortcuts', 'B. Turning their head left and right', 'C. Scrolling with a mouse', 'D. Typing the tab number']
'Correct Answer': '[[B]] Turning their head left and right'

Figure 36: Case study for XR (part 1).

**(Part 2) XR**

**Description2:** "The video starts with a full-screen view of a browser interface titled 'Apple Vision Pro' displaying various website thumbnails and bookmarks. The user then turns their head to the right, causing the right side of the browser to come into view. Next, the user looks at a dot at the bottom of the page and pinches with both hands, which causes the browser to exit the full-screen view and shrink to a smaller, windowed mode. The user then performs a pinching motion with both hands, bringing the fingers towards the middle, which causes the browser's various pages to expand, giving an overview of multiple open tabs. The user again turns their head to the right to view the right side page and then to the left to view the left side page. Throughout the video, the GUI elements such as tabs, the address bar, and website thumbnails respond dynamically to the user's head movements and hand gestures."

**Sequential-QA:** 'Question': 'What action did the user take immediately after turning their head to the right to view the right side page?'
'Answer': 'After turning their head to the right to view the right side page, the user looked at a dot at the bottom of the page and pinched with both hands, which caused the browser to exit full-screen mode and shrink to a smaller windowed interface.'

**Prediction:** 'Question': 'If the user pinches with both hands and moves their fingers towards the middle after the browser shrinks to a smaller window, what will happen next?'
'Answer': "If the user pinches with both hands and moves their fingers towards the middle after the browser shrinks, the action will cause the browser's various pages to expand, providing an overview of multiple open tabs."

**Conversation:** 'User 1': 'How do I view all the pages I have open in the browser?'
'Assistant 1': 'You can view all the open pages by pinching with both hands and moving your fingers towards the middle. This will expand the browser and show an overview of all the open tabs.'
'User 2': 'Great, and how can I exit this overview?'
'Assistant 2': 'To exit the overview of open pages, you can look at the dot at the bottom of the page and pinch with both hands. This will exit the overview and return you to the individual page view.'

**Reasoning:** 'Question': 'How can the user access the options to open a new tab or window from the current state?'
'Options': ['A. Turn their head to the left and select the plus sign.', 'B. Swipe left on the touchpad.', 'C. Turn their head to the right and select the 'Done' button.', 'D. Pinch with both hands to exit the current view and access the toolbar.']
'Correct Answer': 'D'

Figure 37: Case study for XR (part 2).

