# OpenReview forum: "GUI-World: A Video Benchmark and Dataset for Multimodal GUI-oriented Understanding"
_ICLR.cc/2025/Conference — ICLR 2025 Poster_

### Official Review · Reviewer_cJKW · 2024-11-01

**Soundness:** 2
**Presentation:** 2
**Contribution:** 3
**Rating:** 6
**Confidence:** 3

**Summary:**

This paper introduces gui-world, a GUI-oriented dataset designed for multimodal large language model (LLM)-based agents, encompassing six distinct GUI scenarios that include videos, human-annotated keyframes, detailed captions, and a variety of question-and-answer types. The scenarios span a range of multimodal GUI environments such as desktop operating systems, mobile platforms, websites, software applications, and extended reality (XR) platforms. To handle the dynamic, sequential tasks within these environments, the authors created a pipeline for human-annotated keyframes and captioning, along with diverse Q&A generated through human-LLM collaboration. Extensive experiments demonstrate that videoChat2, trained with a two-stage training architecture, surpasses multiple existing visionLLM and videoLLM models in performance. Lastly the authors provide a number of insights that can inspire future work.

**Strengths:**

- Strong motivation behind the dataset collection effort, underscoring the significance of the topic.
-  A comprehensive new dataset, developed with a unique construction pipeline, to evaluate and enhance GUI-oriented capabilities in MLLMs.
-  A method aimed at improving model perception of GUI elements.
-  Thorough experiments and detailed result analysis.

**Weaknesses:**

1) Unclear Novelty of the Model: The novelty of the model is not well established. Although VideoChat2 is used for experiments, it’s unclear which aspects of the model or pipeline method are innovative. The model seems to closely resemble the approach in Li et al. (2023b), and the construction method appears similar to that in Lai et al. (2024). Greater justification of the model’s novelty and a clear rationale for selecting VideoChat2 would be valuable. Additionally, training and comparing models other than VideoChat2 in the same way could strengthen claims of improved GUI perception abilities in Section 3.

2) Low Clarity and Unclear Descriptions:
2-1)Table 1: While Table 1 uses check marks to indicate inclusion, more detailed quantitative and qualitative comparisons across each category would enhance clarity. For instance, some existing datasets also include websites, as indicated in Table 1, but it’s unclear how the quantity or quality of the proposed dataset compares with those.
2-2)Table 3 Results: The results in Table 3 seem selectively presented, as data for "H" in Gemini-Pro-1.5 and "R" and "E" for GPT-4o are missing. Without these results, it’s challenging to assess whether the performance gains are mainly due to human annotations or if GPT-4o is inherently a strong model without them.
2-3)Score Analysis: The authors report accuracy using the LLM-as-a-Judge approach, but score variations appear minor. Applying statistical tests could clarify which scores in the table are significantly higher.

3) Low Presentation Quality: The paper’s readability and flow could be improved to enhance presentation quality.
3-1)	Figure 2: Currently, Figure 2 lacks contextual information. Moving it to the evaluation section, along with relevant text from line 96, could make it more informative.
3-2) Mixed Results, Implications, and Limitations in Section 4.2: Separating results, implications, and limitations in Section 4.2 could improve readability. Specifically, a distinct discussion of limitations and implications would help to inspire and guide future research more effectively.

**Questions:**

1) Could the authors provide further justification for the novelty of the model and training method?
2) Could the authors include the missing results in Table 1, specifically"H" for Gemini-Pro-1.5 and "R" and "E" for GPT-4o?
3) Could you report which scores in the results are statistically significantly higher than others?
4) It’s unclear in the paper whether the dataset will be made publicly available. If so, could the authors specify when it will be accessible?
5) Many references include only titles without complete publication details, such as conference or journal venues, page numbers, and publication years.
6) How were keyframes randomly selected for the "Random" condition? Table 3 presents results for Random selection, but it’s not clear which method was used. Was an established random selection method applied, or was a new tailored method developed for this study?
7) In Line 205, it’s noted that “they typically underperform…” How did the authors address this limitation? If GUI-Vid exhibits this issue, an example would be helpful to illustrate.
8) The data collection reportedly covers a wide range of software and websites, but the quantity of data collected isn’t specified. To assess balance in data collection, it would be useful to include the number of samples by software and websites, such as in Figures 9 and 10.

---

> ### Author Response · Authors · 2024-11-23
> **Rebuttal by Authors (1)**
>
> Thank you very much for your valuable feedback. We apologize for any confusion caused by certain details in the paper. We will address each of your concerns and provide explanations to help you better understand the contributions of this paper step by step:
>
> ---
>
> **Q1:** Could the authors provide further justification for the novelty of the model and training method?
>
> **A1:** Thank you for your critique. Our innovation in our training methodology is the two-step training pipeline. As shown in Figure 8, this approach—first training on video captions before progressing to complex reasoning or conversation tasks—proves more effective than training directly on mixed data. We would like to emphasize that our primary contribution lies in proposing a GUI-oriented dataset and benchmark. We utilized existing models for training primarily to validate the effectiveness of our dataset, rather than modifying model architectures to demonstrate novelty. We selected VideoChat2 because it achieved SOTA performance at the time of our research; LLaVA-OV was released in August.
>
> ---
>
> **Q2:** Could the authors include the missing results in Table 1, specifically"H" for Gemini-Pro-1.5 and "R" and "E" for GPT-4o?
>
> **A2:** Thank you for this valuable feedback. We have expanded our experimental evaluation to include both Gemini and GPT-4, with detailed results presented in **Tables 1, 2, and 3**. In response to your comments, we have substantially revised this section of the manuscript. The keyframe identifier is now framed as an exploratory experiment, with all main results utilizing human evaluation to simulate realistic scenarios of model perception and operation. We have also incorporated two additional baselines for keyframe selection, termed 'model-based' approaches, drawing from the embodied AI domain. For clarity, we have renamed the 'Extract' component to 'Program.' For a comprehensive view of our keyframe selection methodology comparisons, we direct you to *Tables 3* and *6* in our revised manuscript.
>
> **Table 1: Gemini-Pro-1.5 performance with human extracted keyframe.**
> | Scenario | Yes_Rate | Description | Conv1 | Conv2 | Dynamic | Static | Caption | Average |
> |--------|----------|-------------|-------|-------|---------|---------|----------|------|
> | Software | 0.829 | 3.138 | 3.828 | 3.945 | 3.127 | 2.906 | 3.128 | 3.385 |
> | Multi | 0.834 | 3.176 | 3.379 | 3.716 | 3.168 | 2.806 | 3.248 | 3.246 |
> | IOS | 0.803 | 3.299 | 3.751 | 3.938 | 3.198 | 3.253 | 3.629 | 3.467 |
> | Website | 0.792 | 3.235 | 3.608 | 3.744 | 3.283 | 3.146 | 3.394 | 3.412 |
> | XR | 0.833 | 3.048 | 3.231 | 3.516 | 2.971 | 2.853 | 3.25 | 3.108 |
> | Mobile | 0.785 | 2.312 | 3.642 | 3.881 | 2.809 | 2.703 | 3.185 | 3.168 |
> | Average | 0.813 | 3.035 | 3.573 | 3.790 | 3.093 | 2.945 | 3.306 | 3.298 |
>
> **Table 2: GPT-4o performance with randomly extracted keyframes.**
> | Scenario | MCQA | Description | Conv1 | Conv2 | Dynamic | Static | Caption | Free |
> |--------|----------|-------------|-------|-------|---------|---------|----------|------|
> | multi | 0.806 | 2.93 | 3.522 | 3.741 | 3.102 | 2.857 | 3.380 | 3.260 |
> | software | 0.864 | 2.995 | 3.904 | 3.983 | 3.086 | 2.900 | 3.537 | 3.388 |
> | website | 0.797 | 2.987 | 3.722 | 3.877 | 3.204 | 3.078 | 3.528 | 3.415 |
> | mobile | 0.900 | 3.062 | 3.733 | 3.809 | 3.108 | 2.995 | 3.452 | 3.347 |
> | IOS | 0.846 | 2.944 | 3.643 | 3.813 | 3.094 | 3.110 | 3.397 | 3.343 |
> | XR | 0.843 | 2.728 | 3.596 | 3.836 | 2.871 | 3.057 | 3.398 | 3.285 |
> | average | 0.843 | 2.941 | 3.687 | 3.843 | 3.077 | 3.000 | 3.449 | 3.340 |
>
> **Table 3: GPT-4o performance with programly (formerly "Extracted") extracted keyframes.**
> | Scenario | MCQA | Description | Conv1 | Conv2 | Dynamic | Static | Caption | Free |
> |--------|----------|-------------|-------|-------|---------|---------|----------|------|
> | software | 0.856 | 2.803 | 3.852 | 3.978 | 3.127 | 2.923 | 3.791 | 3.388 |
> | multi | 0.810 | 2.836 | 3.762 | 3.912 | 3.186 | 2.831 | 3.603 | 3.371 |
> | IOS | 0.793 | 2.721 | 3.773 | 3.867 | 2.947 | 2.912 | 3.732 | 3.284 |
> | website | 0.817 | 2.958 | 3.803 | 3.796 | 3.246 | 3.182 | 3.812 | 3.462 |
> | XR | 0.833 | 2.453 | 3.318 | 3.563 | 2.782 | 2.603 | 3.127 | 3.012 |
> | mobile | 0.885 | 2.812 | 3.702 | 3.907 | 3.021 | 2.779 | 3.793 | 3.283 |
> | average | 0.832 | 2.764 | 3.702 | 3.837 | 3.052 | 2.872 | 3.643 | 3.300 |

---

> > ### Author Response · Authors · 2024-11-23
> > **Rebuttal by Authors (2)**
> >
> > **Q3:** Could you report which scores in the results are statistically significantly higher than others?
> >
> > **A3:** In our experiments, we identified several statistically significant findings:
> >
> > 1. As shown in *Table 3*, commercial Image LLMs significantly outperform open-source Video LLMs, with GPT-4V demonstrating superior performance compared to all other commercial models tested in our study.
> >
> > 2. *Table 4* in our paper reveals that including vision input substantially outperforms approaches that replace visual input with keyframe captions, highlighting the crucial role of visual context in GUI tasks.
> >
> > 3.  In our task breakdown analysis, we found that models perform significantly worse on dynamic tasks compared to static tasks or basic caption tasks. Even the strongest commercial model, GPT-4V, only achieves a score of 3.117 on these dynamic tasks.
> >
> > ---
> >
> > **Q4:** It’s unclear in the paper whether the dataset will be made publicly available. If so, could the authors specify when it will be accessible?
> >
> > **A4:** We have open-sourced our dataset along with complete code for model training and inference. The dataset is released under the CC 4.0 license, demonstrating our commitment to contributing to the open-source community. Due to double-blind review requirements, we regret that we cannot provide the links here.
> >
> > ---
> >
> > **Q5:** Many references include only titles without complete publication details, such as conference or journal venues, page numbers, and publication years.
> >
> > **A5:** Thank you for your careful attention to detail. Upon review, we found that many references were using the 'misc' citation format. We have updated these to the 'article' citation format in the new manuscript. We greatly appreciate your observation!
> >
> > ---
> >
> > **Q6:** How were keyframes randomly selected for the "Random" condition? *Table 3* presents results for Random selection, but it’s not clear which method was used. Was an established random selection method applied, or was a new tailored method developed for this study?
> >
> > **A6:** Regarding the random setting, we employed uniform sampling to select 10 frames from each video, maintaining equal intervals between frames. We did not implement any other random selection methods. We have added these details to the experiment setup section. Additionally, we have updated all experiments to use human evaluation as the primary keyframe identifier to simulate scenarios where models perceive and operate as humans would. Thank you for bringing this to our attention!
> >
> > ---
> >
> > **Q7:** How did the authors address this limitation? If GUI-Vid exhibits this issue, an example would be helpful to illustrate.
> >
> > **A7:** We have identified two main challenges. First, as analyzed in our paper, the language backbone of VideoLLM was relatively weak, an issue that has been partially addressed with the recent release of LLaVA-OV. Second, there is a data scarcity problem in the GUI video domain. Although our dataset contains more than 12,000 samples, this volume is insufficient for pretraining and can only support fine-tuning. Recent research has shown that incorporating document data into pretraining significantly improves model performance on rich-text tasks. Following this insight, we believe that once sufficient GUI data becomes available for pretraining (specifically for image-text alignment), open-source GUI models will demonstrate substantial performance improvements.
> >
> > ---
> >
> > **Q8:** The data collection reportedly covers a wide range of software and websites, but the quantity of data collected isn’t specified. To assess balance in data collection, it would be useful to include the number of samples by software and websites, such as in *Figures 9* and *10*.
> >
> > **A8:** Thank you for your valuable suggestions. In the next version of our manuscript, we will enhance *Figures 9* and *10* by incorporating statistical analyses of the data samples and percentages. Given the complex graphical elements involved, we are carefully considering the most effective approach to visualize our data distribution.
> >
> > ---
> >
> > Thank you again for taking the time to review our paper. Your feedback has helped us improve both the clarity and quality of our work.

---

> > > ### Comment · Reviewer_cJKW · 2024-11-25
> > >
> > > Thank you for the answers.
> > > A5: Could the authors review the citations again? I noticed that many citations remain incomplete. Simply changing misc to article does not fully address the issue. If a work is from arXiv, it should be indicated as such in the paper. Also, some references are now published works and should no longer be listed as arXiv preprints (e.g., Mapping Natural Language Instructions to Mobile UI Action Sequences).
> > > A8: Thank you. I will review the updates to Figures 9 and 10, when they are ready.

---

> > > > ### Author Response · Authors · 2024-11-25
> > > > **Rebuttal by Authors**
> > > >
> > > > Thank you for your prompt response. We have addressed your concerns in the latest manuscript.
> > > >
> > > > ---
> > > > **A5:** Citation problem.
> > > >
> > > > **Re-A5:** Thank you for sharing the solution. We discovered that many of our previous BibTeX entries were imported directly from arXiv, which contained `eprint` fields but lacked journal information, resulting in abbreviated citations. We have now updated all citations with complete information from google scholar, including proper journal references where applicable. Thank you for this valuable feedback that helped improve our paper's citation problem.
> > > >
> > > > ---
> > > > **A8:** Figure Update.
> > > >
> > > > **Re-A8:** Thanks for your patient. We have created a pie chart to better illustrate our dataset distribution in ***Figure 10***. To improve readability, we merge apps representing less than 2% of the total into an "Others" category. While we acknowledge that visualizing such diverse data presents challenges, we believe this revised representation offers the most effective way to show the distribution of our dataset.

---

> > > > > ### Author Response · Authors · 2024-12-02
> > > > > **Thanks!**
> > > > >
> > > > > Thank you for raising your score! We sincerely appreciate your time and dedication in reviewing our work and are truly delighted by your strong endorsement of our research.

---

> > ### Comment · Reviewer_cJKW · 2024-11-25
> >
> > Thank you for your responses. I have noted the updates regarding the model novelty and believe these changes do not diminish the contribution of the paper.

---

### Official Review · Reviewer_HMZj · 2024-11-03

**Soundness:** 3
**Presentation:** 3
**Contribution:** 3
**Rating:** 8
**Confidence:** 4

**Summary:**

This paper argues that "existing works" for evaluating multi-modal language models' (MMLMs') ability to perform vision-dependent tasks with graphical user interfaces (GUIs) are limited in their evaluative characteristics. To remedy this problem, the authors introduce three key contributions: 1) GUI-World,  a comprehensive GUI dataset comprising 12,379 videos specifically designed to assess and improve GUI-oriented capabilities of MLLMs, spanning a range of categories and scenarios, including desktop, mobile, and eXtended Reality (XR), and representing the first GUI-oriented instruction-tuning dataset in video domain; 2) GUI-Vid, a GUI-oriented video LLM with enhanced capabilities to handle various and complex GUI tasks;, and 3) a series of experiments that demonstrate how existing MLLMs struggle with GUI-oriented tasks and suggest that improvements in vision perception, along with an increase in the number of keyframes and higher resolution, can boost performance in GUI-oriented tasks.

**Strengths:**

I'd like to express my thanks to the authors. The paper was an enjoyable and thorough read.

## Overview of Strengths
- Manuscript and appendix very well-written and thorough.
- Problem is important to several areas machine learning and adjacent areas of science (e.g., human-computer interaction)
- Multiple contributions (i.e., large dataset, model, and new knowledge from several experiments)
- Comprehensive and rigorous in experimental breadth.

### 1. Manuscript Quality
The manuscript is extremely well-written and thorough in its objective. The appendix is *incredibly* thorough, and I found it challenging to identify gaps in aspects of the paper that weren't communicated in some capacity or another. I have no qualms about the paper's writing quality, though there are areas where specific phrases and word choices could be revised.

### 2. Problem Significance
The paper argues that "existing works" for evaluating multi-modal language models' (MMLMs') ability to perform vision-dependent tasks with graphical user interfaces (GUIs) are limited in two ways: (1) an inability to handle dynamic environments that involve sequential operations and (2) significantly limited support for scenarios beyond web-based environments (e.g., web browsers). An alternative (and arguably simpler) framing of the tackled problem is that current evaluation datasets fail to evaluate MMLMs in a comprehensive and robust fashion. Generally speaking, the problem is one that is recognized as being significantly important to the broader ML community.

### 3. Scope of Contributions
The paper frames its contributions around a dataset, a model trained on the dataset, and a series of experiments with the trained model. Generally speaking, I believe that the work is significant and contributes new knowledge that would be valuable not only to ICLR's community, but certainly more broadly (e.g., to the field of human-computer interaction). The work's comprehensive nature is broad in several ways (e.g., data collection, task type, evaluation, etc.). The originality and significance of these contributions is somewhat of an open question, which I discuss under Weaknesses.

### 4. Experimental Rigor
The paper employs approaches and methods that are both well-motivated and well placed in the literature for ICLR's standards. Most of the paper's methodological novelty stems from dynamic and sequential GUI content, which I find to be generally well-grounded as they're supervised, validated, and refined by human annotators. I'd argue that the data collection process itself is quite rigorous in its approach and can be viewed as a standard baseline for collecting such data. There's sufficient detail to *mostly* replicate the methodology with some exceptions.

**Weaknesses:**

There are two high-level weaknesses with the manuscript:
- Novelty of contributions needs clearer articulation
- Details aren't sufficiently clear for reproducibility.

## 1. Novelty of Contributions
The paper's magnitude of contributions make it challenging to assess significance and contribution as clearly as I'd like to for several reasons.
1. The differentiation between contributions from existing literature and the trifecta of three primary contributions presented in this manuscript (i.e., dataset, model, and experiments) are simply not well articulated. The authors repeatedly use phrases such as "take the first step to". It's unclear if this is said with respect to this being the first step in accomplishing their goal within the scope of their paper or in a larger context (i.e. within the scope of the entire research community).
2. There are similar works that aren't included that are more narrow than GUI-World, but certainly have overlap with portions of the GUI-World dataset (e.g. [1]). It's unclear how GUI-World is substantially different for these areas of overlap.
There are a large family of prior works that predate MLLMs but fundamentally operate on the same basis of data collection + practical application (e.g., [2,3,4,5]). The authors should clarify their contributions in the scope of this prior work *and* integrate these clarifications into their paper.
3. The manuscript has a plethora of smaller contributions that come in the form of "small findings". For example, the notion that programmatic extraction of keyframes is disadvantageous for MMLMs is a *stellar* finding that's buried by the emphasis on other contributions more aligned to #1. I'd like to see the authors elevate their contribution clarity by surfacing these smaller findings at greater detail.
4. It's unclear if the data collection methodology itself (i.e. Steps 1 and 2 in the pipeline) should be viewed as a contribution.

[1] Chen, Wentong et al. “GUICourse: From General Vision Language Models to Versatile GUI Agents.” ArXiv abs/2406.11317 (2024): n. pag.
[2] Li, Gang, and Yang Li. "Spotlight: Mobile ui understanding using vision-language models with a focus." arXiv preprint arXiv:2209.14927 (2022).
[3] Wang, Bryan, et al. "Screen2words: Automatic mobile UI summarization with multimodal learning." The 34th Annual ACM Symposium on User Interface Software and Technology. 2021.
[4] Wang, Bryan, Gang Li, and Yang Li. "Enabling conversational interaction with mobile ui using large language models." Proceedings of the 2023 CHI Conference on Human Factors in Computing Systems. 2023.
[5] Li, Yang, et al. "Widget captioning: Generating natural language description for mobile user interface elements." arXiv preprint arXiv:2010.04295 (2020).

### 2. Detail Clarity and Reproducibility
The paper's claims hinge on methodological details that aren't well articulated. I view this as a problem that stems from the paper's scope being so incredibly large, but its a problem that must be remedied nonetheless. The following provides several pointers to areas where I needed more information to understand how the methodology should be reproduced:
1. The reliability of the LLM-as-a-Judge method is not detailed in any capacity. It is unclear how rigor and reliability were established with the method, if at all, for the scope of the authors' task.
2. The design of the process followed by student workers is not detailed. The paper states that students were provided with a "specific software task", but the set of "software tasks" is not explained in any capacity.
3. The manuscript's paragraph on YouTube Video data collection should specifically include portions of the content in A.3. that detail the search + collection process used by human collectors. Otherwise, readers would be misled to believe that the collection process was targeted at collecting videos of "specific software tasks" rather than collection tutorial videos without a specific task in mind.

**Questions:**

1. Provide explicit statements that indicate the significance of each of your contributions (e.g., "We are the first to do X, Y, and Z").
2. Clarify whether the data collection pipeline itself is a contribution.
3. Add appropriate citations for relevant work that you are building on *and* explicitly state how the methodology presented here is different.
4. Add additional commentary on the "smaller contributions" that as noted under "Weaknesses".
5. Incorporate all cited works mentioned in weaknesses to broaden the scope of cited prior work.
6. Include more information about the LLM-as-a-Judge method's reliability in your pipeline. (Note: To be clear, I am not looking for citations from other studies. I am seeking more data to understand that the LLMaaJ method works with your specific prompts.)
7. Replace "existing works" in the Introduction with text that is more explicit in its meaning.
8. Clearly describe and detail the process followed by student works including the specific set of tasks that were used.

**Details Of Ethics Concerns:**

Legal compliance. The authors introduce a dataset of videos that are sourced from YouTube and a family of social media platforms. It is unclear if there is an concern regarding copyright and/or Terms of Service.

Responsible research practice. The authors employ a data collection methodology that sources task-related video recording data from student workers. This is, by definition, technically considered human subjects data, and it is unclear if what approval processes the authors went through in order to collect the data (e.g., an IRB review)  and release the data (e.g., waivers signed by student workers).

---

> ### Author Response · Authors · 2024-11-23
> **Rebuttal by Authors (1)**
>
> Thank you very much for your valuable feedback. We apologize for any confusion caused by certain details in the paper. We will address each of your concerns and provide explanations to help you better understand the contributions of this paper step by step:
>
> ---
> **Q1:** Provide explicit statements that indicate the significance of each of your contributions.
>
> **A1:** We thank the reviewers for the opportunity to clarify the significance of our contributions. Below, we address the question regarding the impact and novelty of each contribution in our work:
>
> 1. **Development of GUI-WORLD Dataset**:
>
>     GUI-WORLD is the first dataset specifically designed for instruction-tuning in video-based GUI domains. It goes beyond static environments, addressing dynamic and sequential GUI tasks. Spanning 12,379 videos across six diverse scenarios (e.g., desktop, mobile, multi-window, and XR), it bridges a critical gap in GUI-oriented MLLM research, enabling the evaluation of dynamic tasks not covered in prior datasets like WebArena. This dataset can serve as both VideoLLM pretraining and instruction tuning for enhancing the GUI-related capability.
>
> 2. **Benchmarking Multimodal LLMs**:
>
>     Our comprehensive benchmarks evaluate nine state-of-the-art MLLMs, revealing their limitations in handling dynamic and sequential GUI tasks. These results emphasize that even advanced models struggle with GUI comprehension without enhancements such as keyframe selection or textual context integration. By providing this benchmark, we establish a baseline for future research in GUI-oriented MLLMs.
>
> 3. **Proposal and Implementation of GUI-Vid**:
>
>     GUI-Vid is the first fine-tuned video-based GUI agent tailored for complex GUI tasks. It achieves a 30% performance improvement over baseline models, surpassing open-source video LLMs and achieving results comparable to leading commercial vision LLMs like GPT-4V. GUI-Vid demonstrates significant improvements in dynamic and sequential GUI comprehension, showcasing the utility of GUI-WORLD.
>
> 4. **Insights into GUI Challenges**:
>
>     Our analysis highlights actionable factors—such as keyframe resolution, the integration of textual information, and improved vision perception—that directly affect model performance. For example, we demonstrate that models leveraging higher-resolution frames perform better, suggesting clear pathways for researchers to optimize future systems.
>
> 5. **Implications for Future Research**:
>
>     GUI-WORLD and our findings establish a foundation for further exploration of GUI-oriented MLLMs in emerging areas like multi-window and XR environments. By identifying existing challenges and proposing GUI-Vid as a prototype solution, our work invites the community to tackle these limitations and improve multimodal LLMs' real-world applicability.
>
> ---
>
> **Q2:** Clarify whether the data collection pipeline itself is a contribution.
>
> **A2:** At the time of our paper writing and submission, works like GUICourse and VideoGUI had not yet been published. Therefore, our claim of novelty is temporally valid in two aspects:
>
> 1. We were the first to develop a pipeline that crawls GUI tutorial videos from YouTube and annotates mouse and keyboard operations within them to create a GUI video dataset.
> 2. We developed specialized software tools and quality assessment metrics for recording and annotating human interactions with GUI videos. This systematic approach to data collection and validation represents a significant technical contribution.
>
> Our data collection pipeline thus constitutes a meaningful contribution to the field, as it introduced novel methodologies and tools for creating high-quality GUI interaction datasets. The timing of our submission predates similar works in this space, establishing our pipeline's originality in the research timeline.
>
> ---
>
> **Q3:** Add appropriate citations for relevant work that you are building on and explicitly state how the methodology presented here is different.
>
> **A3:** Thank you for bringing this to our attention. We have updated both our tables and the related work section to include citations and discussions of all the relevant works you mentioned. Our key differentiation can be summarized as follows: we are the first to extend GUI understanding tasks into the video domain, and we demonstrate the critical importance of visual dynamic context contained within videos for comprehending GUI-related tasks.
>
> ---
>
> **Q4:** Add additional commentary on the "smaller contributions" that as noted under "Weaknesses".
>
> **A4:** Thank you for your valuable suggestions! We have thoroughly restructured our manuscript by providing deeper insights through expanded analysis of model failure cases. Additionally, following **Reviewer iEU7**'s recommendation, we have reorganized the paper structure by presenting our proposed model after the benchmark section, which enhances the overall clarity and flow of the manuscript.

---

> > ### Author Response · Authors · 2024-11-23
> > **Rebuttal by Authors (2)**
> >
> > **Q5:** Incorporate all cited works mentioned in weaknesses to broaden the scope of cited prior work.
> >
> > **A5:** Thank you for your suggestions. The papers you referenced are highly relevant, and we have incorporated them all into *Table 1* and the related work section of our latest manuscript.
> >
> > ---
> >
> > **Q6:** Include more information about the LLM-as-a-Judge method's reliability in your pipeline.
> >
> > **A6:** In Appendix C.3 and Table 16, we demonstrate the correlation between LLM-as-a-Judge scores and human evaluations. For this analysis, we selected 1,000 samples (consisting of video, question, answer, and ground truth) for human annotation to validate the reliability of the LLM-as-a-Judge approach. Our results show that GPT-4 achieves a Pearson correlation coefficient of 0.856 with human judgments, indicating that GPT-4 can effectively serve as a reliable substitute for human evaluation. This strong correlation substantiates the reliability of our LLM-as-a-Judge.
> >
> > ---
> > **Q7:** Replace "existing works" in the Introduction with text that is more explicit in its meaning.
> >
> > **A7:** Thanks for your advice. In our latest version of the manuscript, we have revised this section accordingly.
> >
> > ---
> >
> > **Q8:** Clearly describe and detail the process followed by student works including the specific set of tasks that were used.
> >
> > **A8:** Thank you for your suggestion. We have updated these details in the dataset construction section of the Appendix in our latest manuscript. In this step, we assigned specific software applications and websites to each student, recording their mouse/keyboard interactions and corresponding GUI recordings as they explored and used these tools. While we did not maintain individual records of which student worker was assigned to which specific website or software, all applications and websites involved in our study are illustrated in Figures 9 and 10 of Appendix A.
> >
> > ---
> >
> > **Q9:** Copyright and privacy of student worker related problems.
> >
> > **A9:** Thank you for your careful attention to this matter. Our dataset is not under the MIT License but rather uses the CC4.0 license, specifically to address YouTube video copyright concerns. Our dataset and models are freely available for academic use and modification with proper attribution. Regarding commercial use, we currently do not intend to commercialize or permit commercial applications of our work, which aligns with YouTube's open-source policies.
> > Concerning the student workers, the dataset publication and release were conducted with explicit consent from all student workers, who are co-authors and volunteers of this paper. To protect their privacy during screen recordings that might involve personal information, we instructed them to use fictitious personal information throughout their interactions.
> >
> > ---
> >
> > Thank you again for taking the time to review our paper. Your feedback has helped us improve both the clarity and quality of our work.

---

> > > ### Author Response · Authors · 2024-12-02
> > > **Rebuttal by Authors: Additional Experiment on Mind2web-Multimodal**
> > >
> > > **Dear Reviewer HMZj**,
> > >
> > > Thank you for your time and effort in reviewing our paper. We add additional experiments to prove the correlation between GUI understanding and GUI operating capability as mentioned by other reviewers.
> > >
> > > We evaluated the correlation between GUI understanding and GUI operating on Mind2Web-Multimodal **[1]** to demonstrate the effectiveness of fine-tuning on GUI-World based on your advice. Mind2Web-Multimodal is a multiple-choice GUI benchmark that assesses GUI operation capabilities, where each sample comprises an image, a task description, and response options. Our evaluation included four experimental settings: zero-shot inference using both VideoChat2 and GUI-Vid, as well as versions of these models fine-tuned on Mind2Web-Multimodal training split. We conducted the fine-tuning for 3 epochs, following the experimental settings detailed in our paper. As shown in **Tables 1** and **2**, fine-tuned GUI-Vid achieve higher accuracy, demonstrating understanding capabilities improves their performance on operational benchmarks.
> > >
> > > **[1]** Mind2Web: Towards a Generalist Agent for the Web
> > >
> > > **Table 1: Performance of 4 settings in test domain subset of Mind2web-Multimodal.**
> > > | Category | VideoChat2 ZeroShot | GUI-Vid ZeroShot | VideoChat2 finetune | GUI-Vid finetune |
> > > |----------|-------------------:|------------------:|-------------------:|------------------:|
> > > | Cooking | 21.28% | 23.40% | 23.40% | **26.24%** |
> > > | Education | 17.61% | 16.61% | 24.92% | **28.24%** |
> > > | Finance | 26.02% | 29.59% | 26.02% | **32.65%** |
> > > | Government | 18.80% | 18.28% | 26.89% | **27.42%** |
> > > | Health | 17.08% | 17.37% | 22.58% | **26.48%** |
> > > | Home service | 19.08% | 20.14% | 21.55% | **28.27%** |
> > > | Housing | 12.21% | 14.13% | 18.20% | **21.84%** |
> > > | Job | 16.25% | 18.00% | 20.00% | **20.25%** |
> > > | Moving | 14.36% | 17.44% | 20.00% | **24.62%** |
> > > | Pet | 14.05% | 17.97% | **21.57%** | 21.24% |
> > > | Shipping | 13.42% | 12.99% | **20.78%** | 20.34% |
> > > | Social media | 22.15% | 24.37% | 22.78% | **29.43%** |
> > > | Weather | 27.14% | 22.14% | **26.43%** | 20.00% |
> > > | Overall | 17.53% | 18.59% | 22.37% | **25.14%** |
> > >
> > > **Table 2: Performance of 4 settings in test website subset of Mind2web-Multimodal.**
> > > | Category | VideoChat2 ZeroShot | GUI-Vid ZeroShot | VideoChat2 finetune | GUI-Vid finetune |
> > > |----------|-------------------:|------------------:|-------------------:|------------------:|
> > > | Auto | 20.00% | 16.00% | **19.00%** | 18.00% |
> > > | Department | 9.52% | 11.90% | 14.29% | **16.67%** |
> > > | Digital | 19.73% | 19.73% | 15.65% | **20.41%** |
> > > | Event | **34.65%** | 25.74% | 21.78% | 22.77% |
> > > | General | 18.32% | 18.85% | 17.80% | **20.94%** |
> > > | Music | 16.67% | 15.66% | 21.69% | **24.10%** |
> > > | Other | 17.81% | 21.92% | 27.40% | **35.62%** |
> > > | Restaurant | 10.70% | 14.44% | **17.65%** | 16.58% |
> > > | Sports | 16.98% | 18.87% | 20.75% | **26.42%** |
> > > | Overall | 17.94% | 17.96% | 18.84% | **21.20%** |
> > >
> > > ---
> > >
> > > Thank you for your invaluable assistance and support. Given the constraints of time, we wish to ensure that our responses have effectively addressed any concerns you may have had. If there are still lingering issues, please feel free to inform us. We eagerly anticipate your additional feedback.
> > >
> > > Once again, we appreciate your time and effort in reviewing our paper.

---

> > > > ### Author Response · Authors · 2024-12-02
> > > > **Follow-up by Authors**
> > > >
> > > > Dear reviewer, we want to follow up regarding our response to your review. If there are any additional concerns or further clarifications needed, we’d be more than happy to provide additional information. We look forward to hearing from you.

---

> > > > > ### Author Response · Authors · 2024-12-04
> > > > > **Thanks for raising the score!**
> > > > >
> > > > > Thank you for raising the score! It means a lot to us. Once again, thank you for taking the time to review and discuss our paper with us! Your efforts have made our paper more solid and better.

---

### Official Review · Reviewer_iEU7 · 2024-11-04

**Soundness:** 2
**Presentation:** 1
**Contribution:** 3
**Rating:** 6
**Confidence:** 4

**Summary:**

This paper presents a new dataset called GUI-World for GUI understanding tasks. The datasets consists primarily of ~12k videos, with extracted keyframes corresponding to various actions across multiple platforms and scenarios including various software packages, websites, and operating systems. The dataset also supplies human+llm generated captions for UI actions and multiple choice questions for a variety of UI understanding tasks. One main contribution of this dataset over existing datasets is a greater focus on "dynamic environments" i.e. tasks that require understanding of a sequence of views of a user interfaces.

The paper also presents GUI-vid. A version of the *VideoChat2* model from Li et al. that has been finetuned on GUI-World and demonstrates improved performance on the eval set of GUI-world that the original model and is preferred by humans in side by side comparisons. The authors note that however GUI-Vid does not outperform pre-trained frontier Vision LLMs such as GPT.

**Strengths:**

1. The scope of the dataset collected, both in terms of number of videos but also the annotations and task generation presented alongside. Relatively few UI understanding datasets are video based and this additional focus on the dynamic aspect of UI transitions will likely be useful to others working on this problem.
2. The evaluation of a number of contemporary multimodal large models on this dataset and insight into performance gaps.
3. The authors also go an extra step to demonstrate that GUI-world can be used to improve UI understanding capabilities via a fine-tuning experiment, though I do find this experiment somewhat limited and will comment on that below.

Overall I lean towards accept and am open to raising my score but have a few concerns regarding presentation of the dataset and results that are detailed below.

**Weaknesses:**

- **Uneven presentation of results/tasks:** The dataset can be partitioned along a number of axes, but that partitioning is not very consistent throughout the paper. The leaves certain results 'missing' given the design considerations and experimental setup presented.
	- Here are some different categorizations of tasks/queries I saw
		- Prediction, Reasoning, Captioning (Fig 3)
		- Static, Sequential, Prediction, Conversation, Reasoning, (Figure 5)
		- Free form, MCQA, Conversation (Table 2)
		- Detailed and Summarized Captioning, Static GUI content, Dynamic and Sequential GUI content (L253-L263)
		- Caption, Complex Tasks, Conversation (Table 5)
	- While i do think it is useful to have a number of orthogonal categorizations of queries in the dataset, many of these all of these are not well described/defined and having so many different ways of describing splits made it hard to follow and map these back onto results. It would be helpful if the authors had a table with definition for the major categorizations they would like readers to know about in this dataset.
	- The categorizations used should hopefully be mirrored in the metadata associated with the videos. It would be helpful for the authors to include an example metadata record (or at least the schema) for one or more videos in an appendix.
	- A gap this work is trying to address is that of dynamic environments/tasks. Yet I find the presentation of the performance difference between static vs dynamic content a bit unclear. It is presented in Table 5, but its not clear to me why "Complex Tasks" is the only thing where there is a split between Static and Dynamic (its also not very clear what "Pred" is as a category not what Complex Tasks includes). Why does Table 5 not have results for MCQA tasks? Could the authors shed a bit more light on their choices for how to organize these results?
- **The generation process for QA pairs in the dataset was not clear to me**. Could the authors briefly elaborate on Section 2.3 and  L1119-L1127 in appendix A.3. and give a step-by-step list of the process?
- **Mismatch between tabular results and conclusions for keyframe selection experiment:** It is possible I missed something so the authors should feel free to clarify, but I don't think the statement in L426-L431 *"Across both basic tasks such as captioning and more complex tasks like prediction and reasoning, significant variations are evident among keyframe selection methods. As shown in Table 22 and Table 24, GPT-4V and Gemini significantly benefit from using human-selected keyframes..."* match the results presented.
	- As far as I can tell neither Table 22, Table 24, Table 2, nor Table 5 present Human selected keyframes with Gemini-Pro 1.5 as stated in the text. What is this referencing?
	- The random keyframe condition seems best for gemini-pro in all the tables referenced. And also for Qwen-VL-Max and GPT-4V in Table 2 and Table 21. And for GPT4V in table 24.
	- Why refer to table 22 and table 24 rather than Table 2 (or 21)?
	- Am I missing something?
- Because of the flow of the paper the impression I took away from section 3 is that GUI-vid was finetuned only on GUI-world, however from Table 12 it looks like around half of the data for stage 1 finetuning came from MetaGUI and OmniAct. It would be better to move some of this information about training datasets used up into the main body of the paper.
- No details about the evaluation protocol/setup whose results are detailed in Table 8 are given. This result is likely quite interesting to readers in addition to just the benchmark scores. It would be helpful if the authors provided some information to contextualize what prefers means in this context and how it is measured.

**Suggestions for improvement**
- I think the paper would be easier to follow if the dataset and the results of evaluation against existing LLMs was presented first (completely) and then the fine-tuning experiment and results of that second.
- The x-axis of bar charts in Figure 7 and Figure 8 should start from zero, or alternatively 1 (as that is the lower bound of score. The current axis makes some difference look bigger than they are, for example comparing XR to other categories in Figure 7 leads one to think that the performance difference is bigger than what one expects after looking at Table 3, only after realizing the x-axis starts at 2.4 does one understand why the gap appears so large.
- The main takeaway I got from the finetuning experiment is that finetuning on the datasaet leads to improved performance on the eval split of this datasets, which is fine but not necessarily surprising. It would likely be more helpful if you evaluated GUI-Vid on one of the existing other benchmarks to show that it generalizes somewhat. I should say that this aspect is not particularly determinative of my score as I think the main strength of the paper lies in the other two contributions.
- I don't think there is much point in referencing GUI control via code generation as is done in L484, its hard to see the relevance of that to this work.

**Questions:**

- Are any parts of this dataset derived from existing datasets? The caption for Table 2 says *"For Android, we select videos from Rico"* (L162), but this wasn't really mentioned in the  data collection portion of the paper. So I just want to clarify my understanding?
- What is the "Textual information" that is referred to in L461-462 that references Table 5. What data in the table should we be looking at to understand the conclusion in this section.
- I couldn't find Table 4 referenced anywhere in the text. Two of the conditions in this have no visual input and its not clear why. I'm not sure what the purpose of this table is? Could the authors say a bit more about what readers should take away.
-  For conversational queries, what task are the models asked to do? How are they prompted to complete this task? The example in Figure 5 doesn't make it very clear what to me what the model is supposed to do when presented with this as a query, in that example, is the model supposed to predict the User's next question?
- L115-L1117 say that an LLM is used to reduce errors made by human annotators. What kind of errors were you seeing that needed fixing?

**Details Of Ethics Concerns:**

The dataset does contain content from tutorial videos downloaded from YouTube. My flag is not because I necessarily think this is a problem, more that I'm not necessarily qualified to assess in this regard.

---

> ### Author Response · Authors · 2024-11-23
> **Rebuttal by Authors (1)**
>
> Thank you very much for your valuable feedback. During this interval, we tried our best to address all your concerns and revised our paper based on your advice. We will address each of your concerns and provide explanations to help you better our latest update of this paper step by step:
>
> ---
>
> **Q1.1:** Uneven presentation of results/tasks in categorizations.
>
> **A1.1:** Thank you for your suggestion. We will reorganize our evaluation framework in the main text into three categories: **Caption, Complex Tasks, and Conversation.** The Caption category includes both simple and detailed captions, testing the model's basic and comprehensive understanding of video content. Complex Tasks encompasses Static analysis, Dynamic analysis, and Reasoning, evaluating the model's ability to interpret GUI elements and make logical inferences based on video content. The Conversation category examines the model's capability to provide helpful responses and maintain coherent multi-turn dialogues with users - a crucial skill for any chatbot.
> This three-tier framework effectively covers fundamental comprehension, reasoning capabilities, and knowledge expression skills.
>
> ---
>
> **Q1.2:** Add a table with definition for the major categorizations.
>
> **A1.2:** Thank you for your suggestion. Given the diverse presentation formats in our dataset, we will include a table to clearly illustrate the categorization method used for each experimental section. Our modified classification in our latest manuscript follows two frameworks: when categorized by scenario, we divide tasks into MCQA and free-form; when categorized by task type, we separate them into caption, complex, and conversation tasks. We did not include MCQA in our final scope as our research primarily focuses on free-form questions.
>
> We provide a clear summary of all tables in our experiments in **Table 1**. You can also view this in *Table 10* in our revised manuscript.
>
> **Table 1: Summary of our table, objective and category (view axis).**
> | Table | Objective | Category |
> |--------|------------|--------------|
> | Table 3 | Comparative analysis of model performance across six GUI scenarios | Scenario-specific |
> | Table 4 | Impact of textual information incorporation on GUI understanding | Scenario-specific |
> | Table 5 | Fine-grained evaluation of free-form responses in software tasks | Task-specific |
> | Table 6 | Assessment of different keyframe selection strategies | Task-specific |
> | Table 7 | Analysis of vision input modalities and quality effects | Task-specific |
> | Table 8 | Comprehensive evaluation of GUI-Vid and its components | Scenario-specific |
>
> ---
>
> **Q1.3:** Include an example metadata record (or at least the schema) for one or more videos.
>
> **A1.3:** Thank you for your suggestion. We will create and upload demonstration videos to our project homepage, which will help the community better understand our dataset construction methodology.
>
> ---
>
> **Q1.4:** Why does Table 5 not have results for MCQA tasks? Could the authors shed a bit more light on their choices for how to organize these results?
>
> **A1.4:** Thank you for your suggestion. We need to address an error in Table 5 where we incorrectly labeled Sequential Task as Dynamic task. To clarify, Dynamic tasks actually encompass both Sequential and Prediction tasks, and we will average these two columns to present the correct Dynamic task results. Regarding the MCQA omission, the table specifically focuses on breaking down free-form questions to calculate their average performance. While this was our intended approach, we acknowledge that we failed to clearly communicate this rationale. We apologize for this lack of clarity and have already corrected this explanation in our latest manuscript.
>
> ---
>
> **Q2:** Could the authors briefly elaborate on human-MLLM collaboration for generating QA pairs?
>
> **A2:** Thank you for your question. Our process consists of three main phases. First, we gather comprehensive video information including keyframes, actions performed at each keyframe, and annotators' descriptions of action purposes (e.g., "scrolling down to view more content"). We also input global video context to GPT-4V, such as the operating platform (Mac, Windows) and an overall operation summary (e.g., "opening a browser, searching for neurips, visiting the official website, and checking submission deadlines"). We prompt GPT-4V to generate diverse QA pairs, captions, and conversation questions in batches, specifically instructing it to avoid repetitive questions to ensure variety.
> In the second phase, human annotators review and correct any inaccurate information in GPT-4V's generated content based on the video context. Annotators also enhance overly simple questions from GPT-4V to increase complexity.
> Finally, we conducted a quality assessment by sampling 1,000 data points for annotation. Our statistical analysis shows a 98% human satisfaction rate, confirming the high quality of our dataset.

---

> > ### Author Response · Authors · 2024-11-23
> > **Rebuttal by Authors (2)**
> >
> > **Q3:** Mismatch between tabular results and conclusions for keyframe selection experiment.
> >
> > **A3:** Thank you for your observation. We would like to clarify that both Random and Human frames performed well, while the traditional programmatic keyframe extraction yielded poor results. Specifically, in Random setting, we input 10 frames to the model, whereas in the Human setting, we averaged only 7.463 frames (as shown in *Table 2*), demonstrating the efficiency advantage of human frame selection. To better present our findings, we have restructured this part as a study examining the impact of different keyframe identification methods in *Table 6* in the revised manuscript.
> >
> > ---
> >
> > **Q4:** Remove data from MetaGUI and OmniAct for training.
> >
> > **A4:** As shown in **Table 2**, after excluding OmniAct and MetaGUI from the training data, the model showed performance decreases in certain areas while demonstrating notable improvements in others. Specifically, we observed significant enhancements in MCQA tasks and XR-related capabilities. Conversely, models trained with MetaGUI and OmniAct exhibited superior performance on general free-form questions. These experimental results validate that models trained exclusively on GUI-World can achieve highly competitive performance, thereby confirming the exceptional quality of our dataset.
> >
> > **Table 2: Performance of GUI-Vid when removing data from MetaGUI and OmniAct.**
> > | Model | VideoChat2 | GUI-Vid | GUI-Vid (w.o. other data) |
> > |-------|------------|---------|---------------------------|
> > | Software_MCQA | 45.50% | **59.90%** | 59.70% |
> > | Software_Free | 2.144 | **2.847** | 2.811 |
> > | Website_MCQA | 42.60% | 54.10% | **57.30%** |
> > | Website_Free | 2.221 | **2.957** | 2.844 |
> > | XR_MCQA | 44.00% | 55.60% | **59.62%** |
> > | XR_Free | 2.005 | 2.764 | **2.853** |
> > | Multi_MCQA | 40.40% | **52.90%** | 50.00% |
> > | Multi_Free | 2.222 | **2.861** | 2.799 |
> > | IOS_MCQA | 40.20% | 51.80% | **55.31%** |
> > | IOS_Free | 2.169 | **2.773** | 2.769 |
> > | Android_MCQA | 44.70% | 53.40% | **57.67%** |
> > | Android_Free | 2.119 | **2.572** | 2.547 |
> > | Ave_MC | 42.90% | 54.60% | **56.60%** |
> > | Ave_Free | 2.147 | **2.796** | 2.767 |
> >
> > ---
> >
> > **Q5:** No details about the evaluation protocol/setup in *Table 8*.
> >
> > **A5:** Thank you for your advice. In the previous version, due to space limits, we put the experiment setup of this experiment in Appendix D. In our revised manuscript, we have moved these to Section 4.2. This evaluation focuses on the quality of generated content, where we selected 30 videos from each scenario, totaling 180 videos. Five annotators were asked to formulate questions in natural language that they found most pertinent for each video. The responses were then generated using both VideoChat2 and GUI-Vid, with annotators selecting the superior response between the two.
> >
> > ---
> >
> > **Q6:** Inconsistency in the x-axis of bar charts in Figure 7 and Figure 8 make results ambiguous.
> >
> > **A6:** Thank you for your valuable suggestion! We will align both graphs by setting their X-axes to start at 1.5, which will eliminate visual inconsistencies when comparing the two figures.
> >
> > ---
> >
> > **Q7:** Evaluate GUI-Vid on one of the existing benchmarks to show that it generalizes somewhat.
> >
> > **A7:** Thank you for raising this concern. We conducted evaluations on MVBench **[1]** across four models. While our model showed some performance degradation compared to the original model, it still significantly outperformed the other two baseline models. We would like to note that the primary purpose of training GUI-Vid was to demonstrate the effectiveness of our dataset, rather than achieving state-of-the-art performance. Therefore, we did not employ the strongest baseline model or optimal training methodologies, which may have contributed to some catastrophic forgetting. However, given that GUI-Vid is specifically designed for GUI-related tasks, we believe this performance trade-off is acceptable within our research context.
> >
> > **Table 3: GUI-Vid outperform two mainstream VideoLLMs, while slightly lack behind VideoChat2 on MVBench. **
> >
> > | Metric | VideoChat2 | GUI-Vid | VideoLLaMA | VideoChatGPT |
> > |--------|------------|---------|------------|--------------|
> > | Action Prediction | 47.5 | 39.0 | 25.5 | 26.0 |
> > | Unexpected Action | 60.0 | 57.0 | 39.0 | 26.5 |
> > | Object Existence | 58.0 | 54.0 | 48.0 | 54.0 |
> > | Object Interaction | 71.5 | 51.0 | 40.5 | 28.0 |
> > | Object Shuffle | 41.0 | 30.5 | 38.0 | 40.0 |
> > | Moving Direction | 23.0 | 21.0 | 22.5 | 23.0 |
> > | Action Localization | 23.0 | 30.5 | 22.5 | 20.0 |
> > | Scene Transition | 88.0 | 65.5 | 43.0 | 31.0 |
> > | Action Count | 39.5 | 34.5 | 34.0 | 30.5 |
> > | Moving Count | 42.0 | 29.0 | 22.5 | 25.5 |
> > | Moving Attribute | 58.5 | 53.5 | 32.5 | 39.5 |
> > | State Change | 44.5 | 41.0 | 45.5 | 48.5 |
> > | Character Order | 36.5 | 39.0 | 32.5 | 29.0 |
> > | Egocentric Navigation | 35.0 | 34.0 | 40.0 | 33.0 |
> > | Episodic Reasoning | 38.5 | 37.0 | 30.0 | 29.5 |
> > | **Average** | **47.1** | **41.1** | **34.4** | **32.3** |

---

> > > ### Author Response · Authors · 2024-11-23
> > > **Rebuttal by Authors (3)**
> > >
> > > **Q8:** I don't think there is much point in referencing GUI control via code generation as is done in L484, its hard to see the relevance of that to this work.
> > >
> > > **A8:** Thank you for this consideration. We acknowledge that GUI understanding represents just one component of a complete GUI Agent system, and that code generation capabilities are also crucial. Our statement at L484 serves to explicitly clarify the scope of our work: our primary contribution focuses on advancing GUI understanding through our dataset, rather than developing code generation capabilities for GUI operation. We made this deliberate choice to maintain a clear focus on the fundamental understanding aspect, which we believe is a critical foundation for any GUI Agent system.
> > >
> > > ---
> > >
> > > **Q9:** Are any parts of this dataset derived from existing datasets?
> > >
> > > **A9:** Thanks for your reminding. For the Android dataset, we indeed derive from Rico. We will add this information in our Section 2 for clarification.
> > >
> > > ---
> > >
> > > **Q10:** What is the "Textual information" that is referred to in L461-462 that references Table 5. What data in the table should we be looking at to understand the conclusion in this section.
> > >
> > > **A10:** The textual information in our study consists of detailed frame-by-frame captions generated by GPT-4V. This experiment was designed to evaluate whether providing such comprehensive textual descriptions could enhance the performance of both Image-based LLMs and Video-based LLMs. We conducted this analysis to understand how additional contextual information influences model performance.
> > >
> > > ---
> > >
> > > **Q11:** *Table 4* is not referred. Two of the conditions in this have no visual input and its not clear why. Could the authors say a bit more about what readers should take away.
> > >
> > > **A11:**  We apologize for the incorrect table reference in our manuscript - the results are presented in *Table 4*, not *Table 5* as previously stated. *Table 4* presents our ablation study that addresses two critical questions in the context of GUI-oriented tasks:
> > >
> > > 1. Given recent discussions suggesting that some MLLMs can perform well without visual input **[2]**, is vision input truly essential for GUI-based tasks?
> > > 2. Can additional textual context, in the form of GPT-4V-generated captions for each keyframe, further enhance MLLM performance on these tasks?
> > >
> > > Our findings reveal two key insights:
> > >
> > > - Vision input plays a crucial role in GUI-oriented tasks, demonstrating that visual understanding is indeed fundamental for effective GUI interaction
> > > - Supplementing the models with detailed frame-by-frame captions leads to improved performance, suggesting that rich textual descriptions can complement visual understanding in GUI-related tasks
> > >
> > > **[2]** Are We on the Right Way for Evaluating Large Vision-Language Models?
> > >
> > > ---
> > >
> > > **Q12:** For conversational queries, what task are the models asked to do? How are they prompted to complete this task?
> > >
> > > **A12:** In this task, we prompted GPT to generate two-turn conversations to train MLLMs in answering questions based on both conversational context and video content. Specifically, we designed this task to evaluate and enhance MLLMs' ability to comprehend and respond to questions while considering both the video content and the preceding dialogue context. Unlike traditional approaches, we did not explicitly constrain GPT-4V to generate conversations around specific themes. Instead, we only required that the dialogues be naturally relevant to the GUI video content, with the second question building upon and referencing the answer to the first question.
> > >
> > > ---
> > >
> > > **Q13:** L115-L1117 say that an LLM is used to reduce errors made by human annotators. What kind of errors were you seeing that needed fixing?
> > >
> > > **A13:** During the human annotation process for keyframe goals and captions, we recognized that the raw annotations might contain grammatical errors or suboptimal sentence structures. To address this quality concern, we employed an LLM to refine these annotations, enhancing their fluency, correcting typographical errors, and improving overall readability while preserving the original semantic content.
> > >
> > > ---
> > >
> > > **Q14:** Ethics Concerns of Youtube Videos.
> > >
> > > **A14:** Thank you for your careful attention to this licensing concern. Our dataset is released under the CC4.0 license rather than the MIT License, specifically to address YouTube video copyright considerations. Under this license, our dataset and models are freely available for academic use and modification, with proper attribution required. Regarding commercial use, we currently do not intend to commercialize the dataset ourselves nor permit commercial usage by others, which aligns with YouTube's terms of service and licensing requirements.
> > >
> > > ---
> > >
> > > Thank you again for taking the time to review our paper. Your detailed and valuable feedback has helped us improve both the clarity and quality of our work a lot.

---

> > > > ### Comment · Reviewer_iEU7 · 2024-11-23
> > > >
> > > > A14: From a quick search it seems that there are two licenses used on Youtube, the "Standard YouTube license" and "Creative Commons". Are the videos in the dataset creative commons licensed? It occurs to me that it might be useful to include metadata about the license of the original video in the metadata to facilitate downstream use by others (who can assess usage constraints in their particular context).

---

> > > ### Comment · Reviewer_iEU7 · 2024-11-23
> > >
> > > - A3: Random keyframe selection: Reading the updated draft and your response to reviewer cJKW about this method, I would not call this random. It seems better described as 'sequential' or 'spaced' or something similar. I had initially thought this was a truly random selection of frames. I still think its a useful result to know that sequential selection can outperform other methods.
> > > - A7: Thank you for running this experiment, could you say more about why you picked MVBench given the focus of this dataset on GUI tasks? My current interpretation of this is fine-tuning on GUI-world makes the base model worse at MVBench, but does MVBench contain a relevant set of tasks that you think GUI-world should help performance on? Why not one of the datasets you mention in the introduction (Table 1)?

---

> > ### Comment · Reviewer_iEU7 · 2024-11-23
> >
> > I really thank the authors for their detailed responses and updates to the paper! I have a few follow comments/questions below.
> >
> > A1.1/1.2: Thanks for attempting to address the issue of categorization, I recognize its challenging in a dataset that varies along a number of axes. However I should say that I don't think the addition of Table 10 to be helpful, it introduces another categorization scheme (Task vs Scenario—what is the difference) and from a flow perspective, its not that helpful to have a table in the appendix explaining the tables in the main paper.
> >
> > A1.3: That isn't what I meant, I meant you should include an example metadata record (text) in the paper/appendix. That would help readers (myself included) understand what annotations/categories we can use to partition the data if we want to use a portion of this dataset (vs categorizations that are just used in your analysis).
> >
> > A1.4: I still don't understand the results in Table 5 in relation to the text in the latest revision. In particular, L371-L373 says _"In the fine-grained tasks depicted in Table 5, GPT-4V and GPT-4o excel with static GUI content and prediction tasks over image sequences but struggle with providing detailed descriptions for entire videos and dynamic content."_ But when I look at the GPT-4o and GPT-4v rows in that table, the "Dyn" scores are higher than the "Static" scores. Which numbers should I be looking at? L377 says _"GUI-vid demonstrates proficiency in sequential tasks  but falls short in both captioning and static content"_, Which tasks are the sequential ones? The "caption/concise" seems to be its highest scoring task.

---

> ### Author Response · Authors · 2024-11-23
> **Rebuttal by Authors to Official Comment by Reviewer iEU7**
>
> Thank you very much for your prompt reply.  We will address each of your concerns and provide explanations to help you better our latest update of this paper step by step:
>
> ---
> **A1.1/1.2:** Thanks for attempting to address the issue of categorization, I recognize its challenging in a dataset that varies along a number of axes. However I should say that I don't think the addition of *Table 10* to be helpful, it introduces another categorization scheme (Task vs Scenario—what is the difference) and from a flow perspective, its not that helpful to have a table in the appendix explaining the tables in the main paper.
>
> **Re-A1.1/1.2:** `Task` and `Scenario` is two primary axis that we consider most important to show the evaluation results. The task-specific analysis shows performance across different capabilities such as image captioning, complex QA, and conversation. The scenario-specific analysis evaluates performance across various application domains such as XR, iOS, software, *etc.* We also update the caption of *Table 10* to avoid ambiguity.
> Due to space constraints in the main paper, *Table 10* has to be placed in the appendix. We plan to integrate this table into the main content in the camera-ready version. Thanks for your advice!
>
> ---
>
> **A1.3:** That isn't what I meant, I meant you should include an example metadata record (text) in the paper/appendix.
>
> **Re-A1.3:** Got it. The following metadata record for one sample has been included in *Figure 9* in the appendix. Thanks for your advice!
>
> ```json
> {
>     "system": "Windows",
>     "app": [
>         "edge, bing, steam"
>     ],
>     "region": "partial",
>     "goal": "View the submission interface for the dataset and benchmark track of nips 2024.",
>     "keyframes": [
>         {
>             "frame": 32,
>             "sub_goal": "Click to start downloading, restart downloading lethal company.",
>             "mouse": "click",
>             "keyboard": "none",
>             "keyboardOperation": ""
>         },
>         {
>             "frame": 176,
>             "sub_goal": "Click on edge, edge returns to the top of the screen.",
>             "mouse": "click",
>             "keyboard": "none",
>             "keyboardOperation": ""
>         },
>         {
>             "frame": 781,
>             "sub_goal": "Click on the hyperlink for dataset and benchmark, preparing to jump.",
>             "mouse": "click",
>             "keyboard": "none",
>             "keyboardOperation": ""
>         },
>         {
>             "frame": 839,
>             "sub_goal": "Jump to openreview, loading.",
>             "mouse": "click",
>             "keyboard": "none",
>             "keyboardOperation": ""
>         },
>         {
>             "frame": 1079,
>             "sub_goal": "The webpage loaded the submission interface for dataset and benchmark track.",
>             "mouse": "none",
>             "keyboard": "none",
>             "keyboardOperation": ""
>         },
>         {
>             "frame": 1131,
>             "sub_goal": "Place the mouse on \"add a submission\"",
>             "mouse": "hover",
>             "keyboard": "none",
>             "keyboardOperation": ""
>         }
>     ],
> }
> ```
> ---
> **A1.4:** Inconsistency between table results and textual analysis.
>
> **Re-A1.4:** Sorry for misleading you in L371-373. After we combine Prediction and Sequential, we forget to revise the result analysis part. We’ve now revised the result analysis that match to table results.
>
> ---
>
> **A3:** Misleading name of `Random` keyframe selection.
>
> **Re-A3:** Sorry for the misleading setting name. To avoid confusion with the existing 'sequential' task, we have replaced the term `Random` with `Linspace`, as it better reflects our methodology of uniform sampling (inspired by `np.linspace()` function).
>
> ---
>
> **A7:** Thank you for running this experiment, could you say more about why you picked MVBench given the focus of this dataset on GUI tasks? My current interpretation of this is fine-tuning on GUI-world makes the base model worse at MVBench, but does MVBench contain a relevant set of tasks that you think GUI-world should help performance on? Why not one of the datasets you mention in the introduction (*Table 1*)?
>
> **Re-A7:** Sorry we misunderstand your point, our formerly intent is to show the general performance of our model in other video VQA task to show that out model is still usable for other video tasks. We’re running experiment on Mind2Web to further prove our dataset enhance GUI understanding.
>
> ---
>
> **A14:** Copyright of Youtube video and advice on containing link for video.
>
> **Re-A14:** Thank you for raising this point. All videos in the dataset are protected under Creative Commons licenses. For videos sourced from YouTube, direct reference links will be provided upon dataset release.
>
> ---
> Thank you once again for taking the time to review our revised manuscript. Your detailed and valuable feedback has helped us improve both the clarity and quality of our work much better.

---

> > ### Author Response · Authors · 2024-12-02
> > **Rebuttal by Authors: Additional Experiment on Mind2web-Multimodal**
> >
> > **Dear Reviewer iEU7**,
> >
> > Thank you for your patience as we completed our additional experiments on GUI operating benchmarks.
> >
> > We evaluated the correlation between GUI understanding and GUI operating on Mind2Web-Multimodal **[1]** to demonstrate the effectiveness of fine-tuning on GUI-World based on your advice. Mind2Web-Multimodal is a multiple-choice GUI benchmark that assesses GUI operation capabilities, where each sample comprises an image, a task description, and response options. Our evaluation included four experimental settings: zero-shot inference using both VideoChat2 and GUI-Vid, as well as versions of these models fine-tuned on Mind2Web-Multimodal training split. We conducted the fine-tuning for 3 epochs, following the experimental settings detailed in our paper. As shown in **Tables 1** and **2**, fine-tuned GUI-Vid achieve higher accuracy, demonstrating understanding capabilities improves their performance on operational benchmarks.
> >
> > **[1]** Mind2Web: Towards a Generalist Agent for the Web
> >
> > **Table 1: Performance of 4 settings in test domain subset of Mind2web-Multimodal.**
> > | Category | VideoChat2 ZeroShot | GUI-Vid ZeroShot | VideoChat2 finetune | GUI-Vid finetune |
> > |----------|-------------------:|------------------:|-------------------:|------------------:|
> > | Cooking | 21.28% | 23.40% | 23.40% | **26.24%** |
> > | Education | 17.61% | 16.61% | 24.92% | **28.24%** |
> > | Finance | 26.02% | 29.59% | 26.02% | **32.65%** |
> > | Government | 18.80% | 18.28% | 26.89% | **27.42%** |
> > | Health | 17.08% | 17.37% | 22.58% | **26.48%** |
> > | Home service | 19.08% | 20.14% | 21.55% | **28.27%** |
> > | Housing | 12.21% | 14.13% | 18.20% | **21.84%** |
> > | Job | 16.25% | 18.00% | 20.00% | **20.25%** |
> > | Moving | 14.36% | 17.44% | 20.00% | **24.62%** |
> > | Pet | 14.05% | 17.97% | **21.57%** | 21.24% |
> > | Shipping | 13.42% | 12.99% | **20.78%** | 20.34% |
> > | Social media | 22.15% | 24.37% | 22.78% | **29.43%** |
> > | Weather | 27.14% | 22.14% | **26.43%** | 20.00% |
> > | Overall | 17.53% | 18.59% | 22.37% | **25.14%** |
> >
> > **Table 2: Performance of 4 settings in test website subset of Mind2web-Multimodal.**
> > | Category | VideoChat2 ZeroShot | GUI-Vid ZeroShot | VideoChat2 finetune | GUI-Vid finetune |
> > |----------|-------------------:|------------------:|-------------------:|------------------:|
> > | Auto | 20.00% | 16.00% | **19.00%** | 18.00% |
> > | Department | 9.52% | 11.90% | 14.29% | **16.67%** |
> > | Digital | 19.73% | 19.73% | 15.65% | **20.41%** |
> > | Event | **34.65%** | 25.74% | 21.78% | 22.77% |
> > | General | 18.32% | 18.85% | 17.80% | **20.94%** |
> > | Music | 16.67% | 15.66% | 21.69% | **24.10%** |
> > | Other | 17.81% | 21.92% | 27.40% | **35.62%** |
> > | Restaurant | 10.70% | 14.44% | **17.65%** | 16.58% |
> > | Sports | 16.98% | 18.87% | 20.75% | **26.42%** |
> > | Overall | 17.94% | 17.96% | 18.84% | **21.20%** |
> >
> > ---
> >
> > Thank you for your invaluable assistance and support. Given the constraints of time, we wish to ensure that our responses have effectively addressed any concerns you may have had. If there are still lingering issues, please feel free to inform us. We eagerly anticipate your additional feedback and hope that, if all your primary concerns have been resolved, you may reconsider raising your score.
> >
> > Once again, we appreciate your time and effort in reviewing our paper.

---

> > > ### Author Response · Authors · 2024-12-02
> > > **Follow-up by Authors**
> > >
> > > Dear reviewer, we want to follow up regarding our response to your review. If there are any additional concerns or further clarifications needed, we’d be more than happy to provide additional information. We look forward to hearing from you.

---

### Official Review · Reviewer_qzkL · 2024-11-04

**Soundness:** 2
**Presentation:** 3
**Contribution:** 3
**Rating:** 5
**Confidence:** 5

**Summary:**

The paper presents GUI-World, a dataset designed to enhance understanding with dynamic GUIs. It includes 12,379 GUI videos spanning six key scenarios (e.g., desktop, websites, mobile) and diverse tasks. The authors train and test a video model GUI-Vid on GUI-World, showing moderate performance gains and highlighting the challenges in dynamic GUI understanding.

**Strengths:**

This paper is novel in video understanding for GUIs, an area with limited research so far.

The dataset is large-scale and spans a wide range of GUI scenarios.

**Weaknesses:**

The main weakness is that the paper does not clearly demonstrate how much this dataset contributes to actual GUI agent tasks. From the examples provided, it seems more beneficial for GUI understanding rather than addressing high-level agent tasks.

**Questions:**

Can you show more experiments to address the issue in Weakness? Is it possible to demonstrate the capability of the model on traditional GUI grounding and agent benchmarks?

---

> ### Author Response · Authors · 2024-11-23
> **Rebuttal by Authors**
>
> Thank you very much for your valuable feedback. We apologize for any confusion caused by certain details in the paper. We will address each of your concerns and provide explanations to help you better understand the contributions of this paper step by step:
>
> ---
>
> **Q1:** The paper does not clearly demonstrate how much this dataset contributes to actual GUI agent tasks.
>
> **A1:** Thank you for your valuable feedback. We would like to first clarify the concept of GUI agents as discussed in our paper. In the field of GUI agents, there are two primary categories: **(1)** GUI assistants that help users interact with interfaces, as demonstrated in **[1]**, and **(2)** agents that execute GUI operations through coding. Our paper specifically focuses on the first category.
>
> We would like to emphasize our key contributions: We have made significant progress in the domain of video-based GUI agents by introducing two pioneering contributions:
>
> 1. The first video GUI caption & instruction & multi-round conversation dataset across six GUI scenarios
> 2. A finetuned GUI VideoLLM that achieves SOTA in GUI understanding.
>
> Furthermore, we have validated the effectiveness of our datasets by achieving state-of-the-art performance on open-source VideoLLM implementations. This accomplishment demonstrates the practical value and robustness of our approach.
>
> Our ultimate goal is to empower the open-source community to make meaningful contributions in the GUI domain and help bridge the gap with closed-source developments.
>
> ---
>
> **Q2:** Can you show more experiments to address the issue in Weakness? Is it possible to demonstrate the capability of the model on traditional GUI grounding and agent benchmarks?
>
> **A2:** As explicitly stated in Section 4.2 and Appendix A1, our agent is not designed for GUI grounding or code writing/execution tasks. The VideoLLM framework we employed has not been fine-tuned for code generation tasks and functions primarily as a GUI chat assistant. Moreover, our dataset does not include code generation tasks for fine-tuning. Consequently, our model does not possess operational task capabilities - a limitation we clearly acknowledged in Section 4.2 (updated to Section 5.2) of our paper:
>
> "However, in our explorative experiments, GUI-Vid still fails in GUI operating tasks via code generation like GPT-4 performing in Cradle **[2]**, which is likely due to the baseline LLM's weaker performance and the challenges of code generation instruction fine-tuning."
>
> We would like to emphasize that research value in the GUI agent domain extends beyond grounding and operational capabilities. Currently, open-source models still demonstrate limited understanding of GUI content, and this understanding is fundamental to developing agents capable of system operation. This precisely highlights the contribution of our dataset and benchmark, whose objective is to advance open-source VideoLLM development in the GUI domain to catch up on commercial models.
>
> ---
> We appreciate your critical feedback and thank you for your thorough review of our paper.
>
> **[1]** CogAgent: A Visual Language Model for GUI Agents
>
> **[2]** Cradle: Empowering Foundation Agents Towards General Computer Control

---

> ### Comment · Reviewer_qzkL · 2024-11-23
> **Response to Authors' Reply**
>
> Thank you for your detailed reply. However, I may find myself unable to fully agree with all the points raised.
>
> The authors define GUI Agents as two categories:
> (1) GUI assistants that help users interact with interfaces, as demonstrated in **CogAgent**,
> (2) agents that execute GUI operations through coding.
>
> Unfortunately, this definition seems to present two issues:
>
> 1. Based on the paper and the reply, the authors appear to emphasize that a chatbot with strong GUI understanding capabilities is itself an agent. This interpretation diverges from the broader and more widely accepted definition of GUI agents within the field.
> 2. The authors split agents into "GUI assistants" and "agents that execute GUI operations through coding." However, GUI agents do not necessarily require coding capabilities. Even without the ability to write code (e.g., using PyAutoGUI or Playwright), LMs can still perform agent tasks via zero-shot or few-shot learning on benchmarks such as Mind2Web or AndroidControl through multi-choice question (MCQ) formats. (Similarly for grounding tasks through SoM prompting.) (And even by generating plans and actions in natural language with a human evaluation. [1])
>
> Even the example provided in the response, CogAgent, undergoes large-scale pretraining followed by fine-tuning on web and Android agent tasks, and it has been **comprehensively evaluated on benchmarks like AITW and Mind2Web**. Moreover, **CogAgent is capable of coding PyAutoGUI** (see its results on OSWorld)
>
> While I understand that improving GUI understanding has the potential to enhance a model's performance on agent tasks, the paper's title, "A Dataset for Agents," implies a stronger focus on how this dataset can directly or indirectly improve agent task performance. As a researcher in this field, I would be more interested in learning how to use this dataset to effectively boost agent task capabilities.
>
> A common paradigm involves two training stages: (1) training on GUI understanding or related data, followed by (2) fine-tuning on downstream agent tasks. The CogAgent mentioned in the reply itself is already a good example. There are also many recent ones  (for example, [2]).
>
> Given the current the discussion, I would recommend revising the title to something like "for GUI Understanding", "GUI Assistance" to more accurately reflect the dataset's focus and applications.
>
> [1] Zheng et al. "Gpt-4v (ision) is a generalist web agent, if grounded." ICML 2024.
> [2] Liu, et al. "Harnessing Webpage UIs for Text-Rich Visual Understanding." arXiv

---

> > ### Author Response · Authors · 2024-11-23
> > **Rebuttal by Authors**
> >
> > Thank you for your detailed illustration of your concern in our paper and previous rebuttal. We acknowledge that our original title somewhat overclaimed the scope of our work. Our dataset and experiments indeed focus primarily on GUI understanding rather than being specifically designed for agent tasks. We apologize for misinterpreting your comments in our previous rebuttal and for any confusion regarding the concept of GUI agents.
> >
> > Based on your suggestions, we propose to revise the title to `GUI-World: A Video Dataset for Multimodal GUI-oriented Understanding,` which better reflects our focus on understanding aspects. This revision aligns more accurately with our core contributions.
> >
> > We appreciate your guidance and support in helping us improve the clarity and accuracy of our work.

---

> > > ### Author Response · Authors · 2024-12-02
> > > **Rebuttal by Authors: Additional Experiment on Mind2web-Multimodal**
> > >
> > > **Dear Reviewer qzKL**,
> > >
> > > Thank you for your patience as we completed our additional experiments on GUI operating benchmarks.
> > >
> > > We evaluated the correlation between GUI understanding and GUI operating on Mind2Web-Multimodal **[1]** to demonstrate the effectiveness of fine-tuning on GUI-World based on your advice. Mind2Web-Multimodal is a multiple-choice GUI benchmark that assesses GUI operation capabilities, where each sample comprises an image, a task description, and response options. Our evaluation included four experimental settings: zero-shot inference using both VideoChat2 and GUI-Vid, as well as versions of these models fine-tuned on Mind2Web-Multimodal training split. We conducted the fine-tuning for 3 epochs, following the experimental settings detailed in our paper. As shown in **Tables 1** and **2**, fine-tuned GUI-Vid achieves higher accuracy, demonstrating understanding capabilities improves their performance on operational benchmarks.
> > >
> > > **[1]** Mind2Web: Towards a Generalist Agent for the Web
> > >
> > > **Table 1: Performance of 4 settings in test domain subset of Mind2web-Multimodal.**
> > > | Category | VideoChat2 ZeroShot | GUI-Vid ZeroShot | VideoChat2 finetune | GUI-Vid finetune |
> > > |----------|-------------------:|------------------:|-------------------:|------------------:|
> > > | Cooking | 21.28% | 23.40% | 23.40% | **26.24%** |
> > > | Education | 17.61% | 16.61% | 24.92% | **28.24%** |
> > > | Finance | 26.02% | 29.59% | 26.02% | **32.65%** |
> > > | Government | 18.80% | 18.28% | 26.89% | **27.42%** |
> > > | Health | 17.08% | 17.37% | 22.58% | **26.48%** |
> > > | Home service | 19.08% | 20.14% | 21.55% | **28.27%** |
> > > | Housing | 12.21% | 14.13% | 18.20% | **21.84%** |
> > > | Job | 16.25% | 18.00% | 20.00% | **20.25%** |
> > > | Moving | 14.36% | 17.44% | 20.00% | **24.62%** |
> > > | Pet | 14.05% | 17.97% | **21.57%** | 21.24% |
> > > | Shipping | 13.42% | 12.99% | **20.78%** | 20.34% |
> > > | Social media | 22.15% | 24.37% | 22.78% | **29.43%** |
> > > | Weather | 27.14% | 22.14% | **26.43%** | 20.00% |
> > > | Overall | 17.53% | 18.59% | 22.37% | **25.14%** |
> > >
> > > **Table 2: Performance of 4 settings in test website subset of Mind2web-Multimodal.**
> > > | Category | VideoChat2 ZeroShot | GUI-Vid ZeroShot | VideoChat2 finetune | GUI-Vid finetune |
> > > |----------|-------------------:|------------------:|-------------------:|------------------:|
> > > | Auto | 20.00% | 16.00% | **19.00%** | 18.00% |
> > > | Department | 9.52% | 11.90% | 14.29% | **16.67%** |
> > > | Digital | 19.73% | 19.73% | 15.65% | **20.41%** |
> > > | Event | **34.65%** | 25.74% | 21.78% | 22.77% |
> > > | General | 18.32% | 18.85% | 17.80% | **20.94%** |
> > > | Music | 16.67% | 15.66% | 21.69% | **24.10%** |
> > > | Other | 17.81% | 21.92% | 27.40% | **35.62%** |
> > > | Restaurant | 10.70% | 14.44% | **17.65%** | 16.58% |
> > > | Sports | 16.98% | 18.87% | 20.75% | **26.42%** |
> > > | Overall | 17.94% | 17.96% | 18.84% | **21.20%** |
> > >
> > > ---
> > >
> > > Once again, thank you for your patience and thorough review of our paper. If you have any further questions, please don't hesitate to ask.

---

> ### Author Response · Authors · 2024-12-02
> **Follow-up by Authors**
>
> Dear reviewer, we want to follow up regarding our response to your review. If there are any additional concerns or further clarifications needed, we’d be more than happy to provide additional information. We look forward to hearing from you.

---

### Author Response · Authors · 2024-11-24
**Paper revision summary**

Dear ACs and Reviewers:

We sincerely thank all reviewers for their detailed and constructive feedback. We have carefully addressed the concerns raised and made substantial improvements to our manuscript. Below are the major updates in our revised version:

## Key Revisions Addressing Reviewers' Concerns:

1. **Paper Title and Focus:** Renamed the paper to "GUI-World: A Video Dataset for Multimodal GUI-oriented Understanding" to better reflect our core contribution and reduce ambiguity.

2. **Structural Improvements:** Reorganized the paper structure by separating model training into a dedicated section, allowing for a clearer presentation of our contributions across dataset creation, evaluation methodology, and model development.

3. **Enhanced Experimental Validation:** Expanded the keyframe selection analysis into a comprehensive ablation study
Incorporated additional baseline methods to evaluate optimal keyframe selection strategies for GUI applications.

4. **Task Categorization Refinement:** Restructured our task taxonomy into three main categories: (1) Caption generation (for pertaining) (2) Complex task (static and dynamic) and multi-round conversation (for instruction tuning). We also combined `sequential` and `prediction` tasks under a unified `dynamic` tasks.

## Additional Enhancements:

1. **Better Documentation:** Added detailed annotation metadata documentation in Appendix A to facilitate partial dataset usage.

2. **Enhance Experimental Clarity:** Provided comprehensive documentation in *Table 10* (Appendix) detailing the view axis and objectives for each experimental evaluation.

3. **Cover More Related Works:** Expanded the related work section to include recommended references, strengthening the paper's theoretical foundation and comprehensiveness.

We believe these revisions have substantially improved the paper's clarity and overall contribution to the field. We welcome any additional feedback or questions you may have.

---

### Author Response · Authors · 2024-12-03
**Thank you for your time and effort!**

Dear reviewers, as the rebuttal period for ICLR 2025 is ending today, we would like to follow up on our previous response to your comments. We would greatly appreciate it if you could take a moment to review our comments and additional experiments. Thank you for taking the time to review and discuss our paper with us! Your efforts have made our manuscript more solid and better.

---

### Author Response · Authors · 2024-12-04
**Global Response from Authors**

Dear Area Chair and Reviewers,

We extend our deepest gratitude for the professional guidance and valuable advice you have provided during the past period. Based on your feedback, we have conducted a comprehensive and thorough revision of our paper, especially addressing the concerns and suggestions you raised. We hope that this revision adequately resolves any doubts you may have.

During the rebuttal period, we conducted revision and additional experiments as follows:

1. Paper Title and Focus **(Reviewer qzkL)**: Renamed the paper to "GUI-World: A Video Dataset for Multimodal GUI-oriented Understanding" to better reflect our core contribution and reduce ambiguity.

2. Structural Improvements **(Reviewer iEU7)**: Reorganized the paper structure by separating model training into a dedicated section, allowing for a clearer presentation of our contributions across dataset creation, evaluation methodology, and model development.

3. Task Categorization Refinement **(Reviewer iEU7 & cJKW)**: Restructured our task taxonomy into three main categories: (1) Caption generation (for pertaining) (2) Complex task (static and dynamic) and multi-round conversation (for instruction tuning). We also combined sequential and prediction tasks under unified dynamic tasks.

4. Better Documentation **(Reviewer iEU7)**: Added detailed annotation metadata documentation in Appendix A to facilitate partial dataset usage.

5. Enhance Experimental Clarity **(Reviewer iEU7)**: Provided comprehensive documentation in Table 10 (Appendix) detailing the view axis and objectives for each experimental evaluation.

6. Cover More Related Works **(Reviewer cJKW)**: Expanded the related work section to include recommended references, strengthening the paper's theoretical foundation and comprehensiveness.

Additional Enhancements:

1. Other Baselines for Human Keyframe Extraction **(Reviewer cJKW)**:
We have expanded the keyframe selection analysis into a comprehensive ablation study, incorporating additional baseline methods and models. This enhanced analysis evaluates optimal keyframe selection strategies for GUI applications, addressing Reviewer cJKW's concerns.

2. Experiments Demonstrating GUI Understanding Boosting GUI Operating Tasks **(Reviewers qzkL, iEU7, & cJKW)**:
We conducted additional experiments evaluating VideoChat2 and GUI-Vid on the Mind2Web-Multimodal dataset in both zero-shot and fine-tuned settings. The results demonstrate that models trained on GUI-World outperform others on Mind2Web, validating that enhanced GUI understanding improves performance in GUI operating tasks and addressing the reviewers' concerns.

3. Removing Other GUI Datasets to Highlight the Efficacy of GUI-World **(Reviewer iEU7)**:
We have removed MobileGUI and OmniAct from the GUI-Vid training set to isolate and validate the efficacy of our GUI-World dataset as requested by Reviewer iEU7.

During the rebuttal period, **Reviewer cJKW** and **HMZj** raised their scores in support of our work. Furthermore, we have thoroughly addressed **Reviewer iEU7** and **qzkL**'s concerns by revising the paper and including additional experiments. We sincerely appreciate all reviewers' valuable time and expertise in reviewing our work.

---

### Meta-Review · Area_Chair_DLKU · 2024-12-20

**Metareview:**

The reviewers generally liked the paper and argued for the acceptance of the paper. While Reviewer qzkL seems less positive about the paper, the review is short and there is no response to the rebuttal. As a result, the meta review is mostly based on other reviewers' comments, and discussions with the authors. The reviewers liked the dataset contributed by the work, and think the dataset is the first of its kind to focus on video understanding for GUIs, an area with limited prior research. The dataset is large-scale and covers a wide range of GUI scenarios and tasks, making it a valuable resource for the research community. In addition, the work showed that fine-tuning on GUI-World can improve the performance of existing video models on GUI understanding tasks. The reviewers also pointed out several weaknesses of the work, including limited focus on agent tasks and the misleading framing / title of the work (understanding versus grounding). Some reviewers also found the presentation of results and task categorizations to be unclear and inconsistent. The reviewers also raised concerns about the reproducibility of the data collection and annotation process. The authors clarified these aspects in the rebuttal and the discussion, and revised the manuscript accordingly.

Overall, despite some initial concerns about clarity and the scope of the paper, the authors have addressed these issues through revisions and detailed responses to the reviewers. The dataset and benchmark are valuable contributions to the field of GUI understanding, and the paper provides insights into the challenges and potential solutions for developing GUI-oriented multimodal LLMs.

**Additional Comments On Reviewer Discussion:**

The authors wrote an effective rebuttal, and here are a few key points from the discussion:
The authors clarified the scope of their work and revised the title to better reflect the focus on GUI understanding.
Addressed concerns about the clarity of presentation and organization of results.
Provided more details about the data collection and annotation process to improve reproducibility.
Conducted additional experiments to further demonstrate the value of their dataset.

---

### Decision · Program_Chairs · 2025-01-22

Accept (Poster)